

# Global Sea Level Budget 1993-Present
WCRP Global Sea Level Budget Group*
*A full list of authors and their affiliations appears at the end of the paper
Corresponding author: Anny Cazenave, LEGOS, 18 Avenue Edouard Belin, 31401 Toulouse,
Cedex 9, France;  anny.cazenave@legos.obs-mip.fr





## Abstract

Global mean sea level is an integral of changes occurring in the climate system in response to unforced climate variability as well as natural and anthropogenic forcing factors. Its temporal evolution allows detecting changes (e.g., acceleration) in one or more components. Study of the sea level budget provides constraints on missing or poorly known contributions, such as the unsurveyed deep ocean or the still uncertain land water component. In the context of the World Climate Research Programme Grand Challenge entitled "Regional Sea Level and Coastal Impacts", an international effort involving the sea level community worldwide has been recently initiated with the objective of assessing the various data sets used to estimate components of the sea level budget during the altimetry era (1993 to present). These data sets are based on the combination of a broad range of space-based and in situ observations, model estimates and algorithms. Evaluating their quality, quantifying uncertainties and identifying sources of discrepancies between component estimates is extremely useful for various applications in climate research. This effort involves several tens of scientists from about fifty research teams/institutions worldwide (www.wcrp-climate.org/grand-challenges/gc-sea-level). The results presented in this paper are a synthesis of the first assessment performed during 2017-2018. We present estimates of the altimetry-based global mean sea level (average rate of 3.1 +/- 0.3 mm/yr and acceleration of 0.1 mm/yr$^2$ over 1993-present), as well as of the different components of the sea level budget (http://doi.org/10.17882/54854). We further examine closure of the sea level budget, comparing the observed global mean sea level with the sum of components. Ocean thermal expansion, glaciers, Greenland and Antarctica contribute by 42%, 21%, 15% and 8% to the global mean sea level over the 1993-present. We also study the sea level budget over 2005-present, using GRACE-based ocean mass estimates instead of sum of individual mass components. Results show closure of the sea level budget within 0.3 mm/yr. Substantial uncertainty remains for the land water storage component, as shown in examining individual mass contributions to sea level.



## 1. Introduction

Global warming has already several visible consequences, in particular increase of the Earth's mean surface temperature and ocean heat content (Rhein et al., 2013, Stocker et al., 2013), melting of sea ice, loss of mass of glaciers (Gardner et al., 2013), and ice mass loss from the Greenland and Antarctica ice sheets (Rignot et al., 2011, Shepherd et al., 2012). On average over the last 50 years, about 93% of heat excess accumulated in the climate system because of greenhouse gas emissions has been stored in the ocean, and the remaining 7% has been warming the atmosphere and continents, and melting sea and land ice (von Schuckmann et al., 2016). Because of ocean warming and land ice mass loss, sea level rises. After about 3000 years of nearly constant evolution since the end of the last deglaciation (e.g., Lambeck et al., 2010, Kemp et al., 2011, Kopp et al. 2014), direct observations from in situ tide gauges available since the mid-to-late 19$^{th}$ century show that the 20$^{th}$ century global mean sea level has been rising at a rate of 1.2 mm/yr to 1.9 mm/yr (Church and White, 2011, Jevrejeva et al., 2014a, Hay et al., 2015, Dangendorf et al., 2017). Since the early 1990s, this rate, now measured by high-precision altimeter satellites, has increased to ~3 mm/yr on average (Legeais et al., 2018, Nerem et al., 2018).

Accurate assessment of present-day global mean sea level variations and its components (ocean thermal expansion, ice sheet mass loss, glaciers mass change, changes in land water storage, etc.) is important for many reasons. The global mean sea level is an integral of changes occurring in the Earth's climate system in response to unforced climate variability as well as natural and anthropogenic forcing factors e.g., net contribution of ocean warming, land ice mass loss, and changes in water storage in continental river basins. Temporal changes of the components are directly reflected in the global mean sea level curve. If accurate enough, study of the sea level budget provides constraints on missing or poorly known contributions, e.g., the deep ocean, marginal seas and shelf areas, and polar regions undersampled by current observing systems, or still uncertain changes in water storage on land due to human activities (e.g. ground water depletion in aquifers). Global mean sea level corrected for ocean mass change in principle allows one to independently estimate temporal changes in total ocean heat content, from which the Earth's energy imbalance can be deduced (von Schuckmann et al., 2016). The sea level and/or ocean mass budget approach can also be used to constrain models of Glacial Isostatic Adjustment (GIA). The GIA phenomenon has significant impact on the interpretation of GRACE-based space gravimetry data over the oceans (for ocean mass change) and over Antarctica (for ice sheet mass balance). However,



there is still incomplete consensus on best estimates, a result of uncertainties in deglaciation
models and mantle viscosity structure. Finally, observed changes of the global mean sea level
and its components are fundamental for validating climate models used for projections.
In the context of the Grand Challenge entitled "Regional Sea Level and Coastal Impacts" of
the World Climate Research Programme (WCRP), an international effort involving the sea
level community worldwide has been recently initiated with the objective of assessing the sea
level budget during the altimetry era (1993 to present). To estimate the different components
of the sea level budget, different data sets are used. These are based on the combination of a
broad range of space-based and in situ observations. Evaluating their quality, quantifying their
uncertainties, and identifying the sources of discrepancies between component estimates,
including the altimetry-based sea level time series, are extremely useful for various
applications in climate research.
Several previous studies have addressed the sea level budget over different time spans and
using different data sets (e.g., Cazenave et al., 2009, Leuliette and Willis, 2010, Church and
White, 2011, Chambers et al., 2017, Dieng et al., 2017, Chen et al., 2017, Nerem et al., 2018).
Assessments of the published literature have also been performed in past IPCC
(Intergovernmental Panel on Climate Change) reports (e.g., Church et al., 2013). Building on
these previous works, here we intend to provide a collective update of the global mean sea
level budget, involving the many groups worlwide interested in present-day sea level rise and
its components. We focus on observations rather than model-based estimates and consider the
high-precision altimetry era starting in 1993 that includes the period since the mid-2000s
where new observing systems, like the Argo float project (Roemmich et al., 2012) and the
GRACE space gravimetry mission (Tapley et al., 2004) that provide improved data sets of
high value for such a study. Only the global mean budget is considered here. Regional budget
will be the focus of a future assessment.
Section 2 describes for each component of the sea level budget equation the different data sets
used to estimate the corresponding contribution to sea level, discusses associated errors and
provides trend estimates for the two periods. Section 3 addresses the mass and sea level
budgets over the study periods. A discussion is provided in Section 4, followed by a
conclusion.

## 2. Methods and Data




In this section, we briefly present the global mean sea level budget (sub section 2.1), then
provide, for each term of the budget equation, an assessment of the most up-to-date published
results. Multiple organisations and research groups routinely generate the basic measurements
as well as the derived data sets and products used to study the sea level budget. Sub sections
2.2 to 2.7 summarize the measurements and methodologies used to derive observed sea level,
as well as steric and mass components. In most cases, we focus on observations but in some
instances (e.g., for GIA corrections applied to the data), model-based estimates are the only
available information.

## 2.1 Sea level budget equation


Global mean sea level (GMSL) change as a function of time t is usually expressed by the sea
level budget equation:
$$\text{GMSL}(t) = \text{GMSL}(t)_{steric} + \text{GMSL}(t)_{ocean\ mass} \qquad (1)$$
where $\text{GMSL}(t)_{steric}$ refers to the contributions of ocean thermal expansion and salinity to sea
level change, and $\text{GMSL}(t)_{oceanmass}$ refers to the change in mass of the oceans. Due to water
conservation in the climate system, the ocean mass term (also noted as $M(t)_{ocean}$) can further
be expressed as:

$$M(t)_{ocean} + M(t)_{glaciers} + M(t)_{Greenland} + M(t)_{Antarctica} + M(t)_{TWS} + M(t)_{WV} + M(t)_{Snow}$$
$$+\ \text{uncertainty} = 0 \qquad (2)$$

where $M(t)_{glaciers}$, $M(t)_{Greenland}$, $M(t)_{Antarctica}$, $M(t)_{TWS}$, $M(t)_{WV}$, $M(t)_{Snow}$ represent temporal
changes in mass of glaciers, Greenland and Antarctica ice sheets, terrestrial water storage
(TWS), atmospheric water vapor (WV), and snow mass changes. The uncertainty is a result of
uncertainties in all of the estimates and potentially missing mass terms, for example,
permafrost melting.

From equation (2), we deduce:

$$\text{GMSL}(t)_{ocean\ mass} = -\ [M(t)_{glaciers} + M(t)_{Greenland} + M(t)_{Antarctica} + M(t)_{TWS} + M(t)_{WV} + M(t)_{Snow}$$
$$+\ \text{missing mass terms}] \qquad (3)$$



In the next subsections, we successively discuss the different terms of the budget (equations 1
and 2) and how they are estimated from observations. We do not consider the atmospheric
water vapor and snow components, assumed to be small. Two periods are considered: (1)
1993-present (i.e. the entire altimetry era), and (2) 2005-present (i.e. the period covered by
both Argo and GRACE).

## 161 2.2 Altimetry-based global mean sea level over 1993-present

The launch of the TOPEX/Poseidon (T/P) altimeter satellite in 1992 led to a new paradigm for
measuring sea level from space, providing for the first time precise and globally distributed
sea level measurements at 10-day intervals. At the time of the launch of T/P, the
measurements were not expected to have sufficient accuracy for measuring GMSL changes.
However, as the radial orbit error decreased from ~10 cm at launch to ~1 cm presently, and
other instrumental and geophysical corrections applied to altimetry system improved (e.g.,
Stammer and Cazenave, 2017), several groups regularly provided an altimetry-based GMSL
time series (e.g., Nerem et al. 2010, Church et al. 2011, Ablain et al., 2015, Legeais et al.,
2018). The initial T/P GMSL time series was extended with the launch of Jason-1 (2001),
Jason-2 (2008) and Jason-3 (2016). By design, each of these missions has an overlap period
with the previous one in order to inter-compare the sea level measurements and estimate
instrument biases (e.g., Nerem et al., 2010; Ablain et al., 2015). This has allowed the
construction of an uninterrupted GMSL time series that is currently 25-year long.

### 176 2.2.1 Global mean sea level datasets

Six groups (AVISO/CNES, SL_cci/ESA, University of Colorado, CSIRO, NASA/GSFC,
NOAA) provide altimetry-based GMSL time series. All of them use 1-Hz altimetry
measurements derived from T/P, Jason-1, Jason-2 and Jason-3 as reference missions. These
missions provide the most accurate long-term stability at global and regional scales (Ablain et
al. 2009, 2017a), and are all on the same historical T/P ground track. This allows computation
of a long-term record of the GMSL from 1993 to present. In addition, complementary
missions (ERS-1, ERS-2, Envisat, Geosat Follow-on, CryoSat-2, SARAL/AltiKa and
Sentinel-3A) provide increased spatial resolution and coverage of high latitude ocean areas, >
66°N/S latitude (e.g. the European Space Agency/ESA Climate Change Initiative/CCI sea
level data set; Legeais et al. 2018).



The above groups adopt different approaches when processing satellite altimetry data. The
most important differences concern the geophysical corrections needed to account for various
physical phenomena such as atmospheric propagation delays, sea state bias, ocean tides, and
the ocean response to atmospheric dynamics. Other differences come from data editing,
methods to spatially average individual measurements during orbital cycles and link between
successive missions (Masters et al. 2012; Henry et al. 2014).
Overall, the quality of the different GMSL time series is similar. Long-term trends agree well
to within 6% of the signal, approximately 0.2 mm/yr (see Figure 1) within the GMSL trend
uncertainty range (~ 0.3 mm/yr; see next section). The largest differences are observed at
interannual time scales and during the first years (before 1999; see below). Here we use an
ensemble mean GMSL based on averaging all individual GMSL time series.


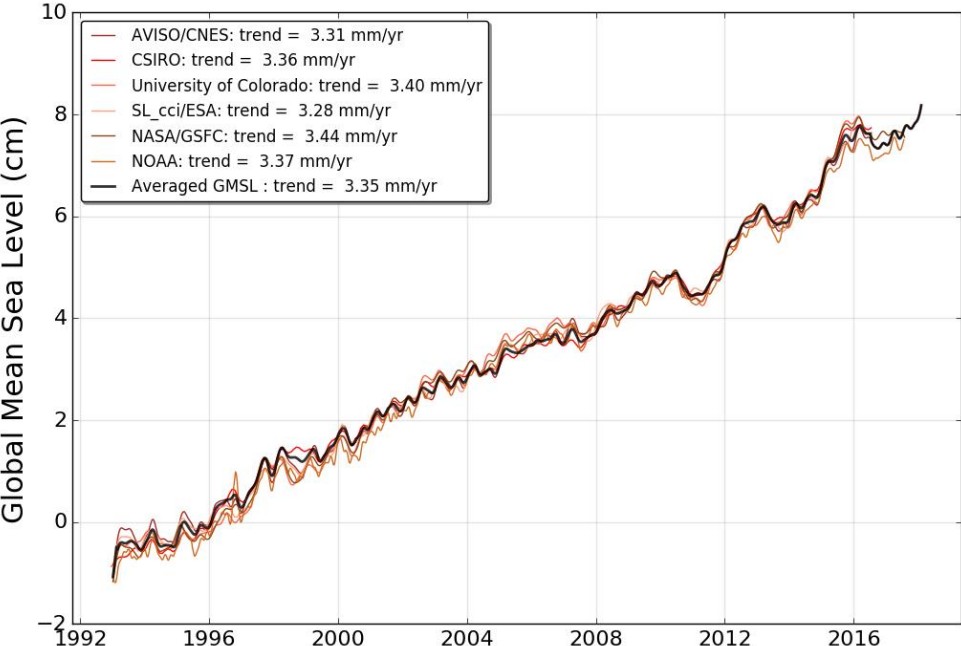



*Figure 1: Evolution of GMSL time series from 6 different groups (AVISO/CNES, SL_cci/ESA,*
*University of Colorado, CSIRO, NASA/GSFC, NOAA) products. Annual signals are removed*
*and 6-month smoothing applied. All GMSL time series are centered in 1993 with zero mean.*
*A GIA correction of -0.3 mm/yr has been subtracted to each data set.*



### 2.2.2 Global mean sea level uncertainties and TOPEX-A drift

Based on an assessment of all sources or uncertainties affecting the altimetric system (Ablain et al. 2017), the GMSL trend uncertainty (90% confidence interval) is estimated as ~0.4 mm/yr over the whole altimetry era (1993-2017). The main contribution to the uncertainty is the wet tropospheric correction with a drift uncertainty in the range of 0.2-0.3 mm/yr (Legeais et al. 2018) over a 10-year period. To a lesser extent, the orbit error (Couhert et al. 2015; Escudier et al., 2017) and the altimeter parameters' (range, sigma-0 and significant wave height/SWH) instability (Ablain et al., 2012) also contribute to the GMSL trend uncertainty, at the level of 0.1 mm/yr. Furthermore, imperfect links between successive altimetry missions lead to another trend uncertainty of about 0.15 mm/yr over the 1993-2017 period (Zawadzki and Ablain, 2016).

Uncertainties are higher during the first decade (1993-2002) where T/P measurements display larger errors at climatic scales. For instance, the orbit solutions are much more uncertain due to gravity field solutions calculated without GRACE data. Furthermore, the switch from TOPEX-A to TOPEX-B in February 1999 (with no overlap between the two instrumental observations) leads to an error of ~ 3 mm in the GMSL time series (Escudier et al., 2017).

However, the most significant error that affects the first 6 years (January-1993 to February 1999) of the T/P GMSL measurements is due to an instrumental drift of the TOPEX-A altimeter, not included in the formal uncertainty estimates discussed above. This effect on the GMSL time series was recently highlighted via comparisons with tide gauges (Valladeau et al. 2012; Watson et al. 2015; Chen et al. 2017; Ablain et al. 2017), via a sea level budget approach (i.e., comparison with the sum of mass and steric components; Dieng et al., 2017) and by comparing with Poseidon-1 measurements (Zawadsky, personal communication). In a recent study, Beckley et al. (2017) asserted that the corresponding error on the 1993-1998 GMSL resulted from incorrect onboard calibration parameters.

All three approaches conclude that during the period January 1993 to February 1999, the altimetry-based GMSL was overestimated. TOPEX-A drift correction was estimated close to 1.5 mm/yr (in terms of sea level trend) with an uncertainty of ±0.5 to ±1.0 mm/yr (Watson et al. 2015; Chen et al. 2017; Dieng et al. 2017). Beckley et al. (2017) proposed to not apply the suspect onboard calibration correction on TOPEX-A measurements. The impact of this approach is similar to the TOPEX-A drift correction estimated by Dieng et al. (2017) and Ablain et al. (2017b). In the latter study, accurate comparison between TOPEX A-based GMSL and tide gauge measurements leads to a drift correction to about -1.0 mm/yr between



January 1993 and July 1995, and +3.0 mm/yr between August 1995 and February 1999, with
an uncertainty of 1.0 mm/yr (with a 68% confidence level, see Table 1).

| TOPEX-A drift correction | to be subtracted from the first 6-years (Jan. 1993 to Feb. 1999) of the uncorrected GMSL record |
|---|---|
| Watson et al. (2015) | 1.5 +/- 0.5 mm/yr over Jan.1993/ Feb.1999 |
| Chen et al. (2017); Dieng et al. (2017) | 1.5 +/- 0.5 mm/yr over Jan.1993/ Feb.1999 |
| Beckley et al. (2017) | No onboard calibration applied |
| Ablain et al. (2017b) | -1.0 +/- 1.0 mm/yr over Jan.1993/ Jul.1995  +3.0 +/-1.0 mm/yr over Aug.1995- Feb.1999 |

*Table 1. TOPEX-A GMSL drift corrections proposed by different studies*

**2.2.3 Global Mean Sea Level variations**

The ensemble mean GMSL rate after correcting for the TOPEX-A drift (for all of the
proposed corrections) amounts to 3.1 mm/yr over 1993-2017 (Figure 2). This corresponds to a
mean sea level rise of about 7.5 cm over the whole altimetry period. More importantly, the
GMSL curve shows a net acceleration, estimated at 0.08 mm/yr² (Chen et al. 2017; Dieng et
al. 2017) and 0.084 +/- 0.025 mm/yr² (Nerem et al., 2018) (Note Watson et al. found a smaller
acceleration after correcting for the instrumental bias over a shorter period up to the end of
2014.). GMSL trends calculated over 10-year moving windows illustrate this acceleration
(Figure 3). GMSL trends are close to 2.5 mm/yr over 1993-2002 and 3.0 mm/yr over 1996-
2005. After a slightly smaller trend over 2002-2011, the 2008-2017 trend reaches 4.2 mm/yr.
Uncertainties (90% confidence interval) associated to these 10-year trends regularly decrease
through time from 1.3 mm/yr over 1993-2002 (corresponding to T/P data) to 0.65 mm/yr for
2008-2017 (corresponding to Jason-2 and Jason-3 data).
Removing the trend from the GMSL time series highlights inter-annual variations. Their
magnitudes depend on the period (+3 mm in 1998-1999, -5 mm in 2011-2012, and +10 mm in
2015-2016) and are well correlated in time with El Niño and La Niña events (Nerem et al.
2010; Cazenave et al. 2014, Nerem et al., 2018). However, substantial differences (of 1-3



mm) exist between the six detrended GMSL time series. This issue needs further
investigation.


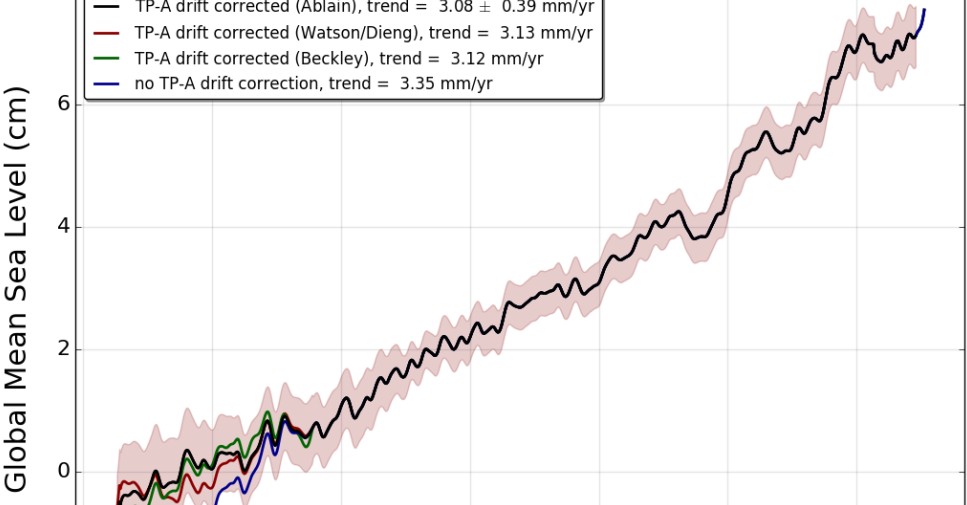



*Figure 2: Evolution of ensemble mean GMSL time series (average of the 6 GMSL products*
*from AVISO/CNES, SL_cci/ESA, University of Colorado, CSIRO, NASA/GSFC, and NOAA).*
*On the black, red and green curves, the TOPEX-A drift correction is applied respectively*
*based on (Ablain et al, 2017b), (Watson et al. 2015; Dieng et al. 2017) and Beckley et al.,*
*2017). Annual signal removed and 6-month smoothing applied; GIA correction also applied.*
*Uncertainties (90% confidence interval) of correlated errors over a 1-year period are*
*superimposed for each individual measurement (shaded area).*








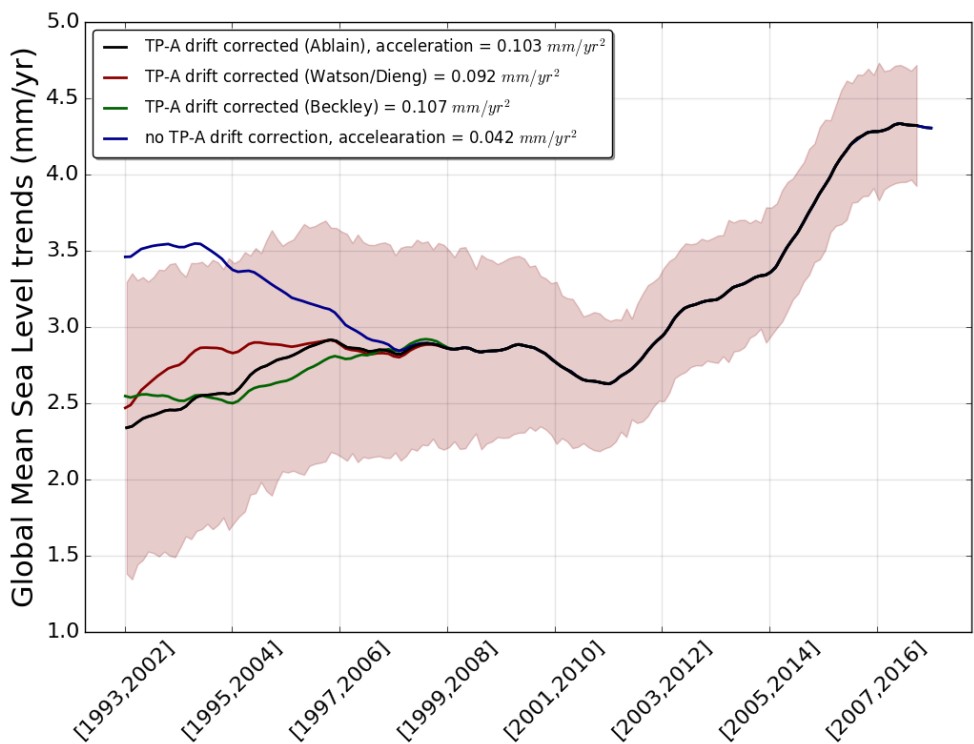


*Figure 3: Ensemble mean GMSL trends calculated over 10-year moving windows. On the black, red and green curves, the TOPEX-A drift correction is applied respectively based on (Ablain et al, 2017b), and Beckley et al., 2017). Uncorrected GMSL trends are shown by the blue curve. The shaded area represents trend uncertainty over 10-year periods (90% confidence interval).*


For the sea level budget assessment (section 3), we will use the ensemble mean GMSL time
series corrected for the TOPEX A drift using the Ablain et al. (2017b) correction.

**_2.2.4. Comparison with tide gauges_**

Prior to 1992 global sea level rise estimates rely on the tide gauge measurements, and it is
worth mentioning past attempts to produce global sea level reconstructions utilizing these
measurements (e.g. Gornitz et al. 1982; Bartnett 1984; Douglas 1991, 1997, 2001). Here we
focus on global sea level reconstructions that overlap with satellite altimetry data over a
substantial common time span. Some of these reconstructions rely on tide gauge data only



(Jevrejeva et al. 2006, 2014; Merrifield et al. 2009; Wenzel and Schroter 2010; Ray and
Douglas 2011; Hamlington et al. 2011, Spada and Galassi 2012; Thompson and Merrifield
2014; Dangendorf et al. 2017; Frederikse et al. 2017). In addition, there are reconstructions
that jointly use satellite altimetry and tide gauge records (Church and White 2006, 2011) and
reconstructions which combines tide gauge records with ocean models (Meyssignac et al.
2011) or physics-based and model-derived geometries of the contributing processes (Hay et
al. 2015).
For the period since 1993, with most of the world coastlines densely sampled, the rates of sea
level rise from all tide gauge based reconstructions and estimates from satellite altimetry
agree within their specific uncertainties, e.g., rates of $3.0 \pm 0.7$ mm· yr$^{-1}$ (Hay et al. 2015); 2.8
$\pm 0.5$ mm· yr$^{-1}$ (Church and White 2011; Rhein et al. 2013); $3.1 \pm 0.6$ mm·yr$^{-1}$ (Jevrejeva et
al, 2014); $3.1 \pm 1.4$ mm·yr$^{-1}$ (Dangendorf et al. 2017) and the estimate from satellite altimetry
3.2 $\pm 0.4$ mm· yr$^{-1}$ (Nerem et al. 2010; Rhein et al. 2013). However, classical tide gauge-
based reconstructions still tend to overestimate the inter-annual to decadal variability of
global mean sea level (e.g. Calafat et al., 2012; Dangendorf et al. 2015; Natarov et al. 2017)
compared to global mean sea level from satellite altimetry, due to limited and uneven spatial
sampling of the global ocean afforded by the tide gauge network. Sea level rise being non
uniform, spatial variability of sea-level measured at tide gauges is evidenced by 2D
reconstruction methods. The most widely used approach is the use of empirical orthogonal
functions (EOF) calibrated with the satellite altimetry data (e.g. Church and White, 2004).
Alternatively, Choblet et al. (2014) implemented a Bayesian inference method based on a
Voronoi tessellation of the Earth's surface to reconstruct sea level during the twentieth
century. Considerable uncertainties remain however in long term assessments due to poorly
sampled ocean basins such as the South Atlantic, or regions which are significantly influenced
by open-ocean circulation (e.g. Subtropical North Atlantic) (Frederikse et al. 2017).
Uncertainties involved in specifying vertical land motion corrections at tide gauges also
impact tide gauge reconstructions (Jevrejeva et al. 2014; Woppelmann and Marcos 2016;
Hamlington et al. 2016).  Frederikse et al. (2017) recently also demonstrated that both global
mean sea level reconstructed from tide gauges and the sum of steric and mass contributors
show a good agreement with altimetry estimates for the overlapping period 1993-2014.

**2.3 Steric sea level**



331 Steric sea level variations result from temperature (T) and salinity (S) related density changes

332 of sea water associated to volume expansion and contraction. These are refered as

333 thermosteric and halosteric components. Despite clear detection of regional salinity changes

334 and the dominance of the salinity effect on density changes at high latitudes (Rhein et al.,

335 2013), the halosteric contribution to present-day global mean steric sea level rise is negligible,

336 as the ocean's total salt content is essentially constant over multidecadal timescales (Gregory

337 and Lowe, 2000). Hence in this study, we essentially consider the thermosteric sea level

338 component.

339 Averaged over the $20^{th}$ century, ocean thermal expansion associated with ocean warming has

340 been the largest contribution to global mean sea level rise (Church et al., 2013). This remains

341 true for the altimetry period starting in the year 1993 (e.g., Chen et al. 2017; Dieng et al.,

342 2017, Nerem et al., 2018). But sum of all land ice components (glaciers, Greenland and

343 Antarctica) during this period, total land ice mass loss now dominates the sea level budget

344 (see section 3).

345 Until the mid-2000s, the majority of ocean temperature data have been retrieved from

346 shipboard measurements. These include vertical temperature profiles along research cruise

347 tracks from the surface sometimes all the way down to the bottom layer (e.g. Purkey and

348 Johnson, 2010) and upper-ocean broadscale measurements from ships of opportunity

349 (Abraham et al., 2013). These upper-ocean in situ temperature measurements however are

350 limited to the upper 700 m depth due to common use of expandable bathy thermographs

351 (XBTs). Although the coverage has been improved through time, large regions characterized

352 by difficult meteorological conditions remained undersampled, in particular the southern

353 hemisphere oceans and the Arctic area.

354

355 ***2.3.1 Thermosteric data sets***

356 Over the altimetry era, several research groups have produced gridded time series of

357 temperature data for different depth levels, based on XBTs (with additional data from

358 mechanical bathythermographs -MBTs- and conductivity-temperature-depth (CTD) devices

359 and moorings) and Argo float measurements. The temperature data have further been used to

360 provide thermosteric sea level products. These differ because of different strategies adopted

361 for data editing, temporal and spatial data gaps filling, mapping methods, baseline

362 climatology and instrument bias corrections (in particular the time-to-depth correction for

363 XBT data, Boyer et al., 2016).

364 The global ocean in situ observing system has been dramatically improved through the



implementation of the international Argo program of autonoumous floats, delivering a unique
inside of the interior ocean from the surface down to 2000 m depth of the ice-free global
ocean (Roemmich et al., 2012, Riser et al., 2016). More than 80% of intitally planned full
deployment of Argo float program was achieved during the year 2005, with virtually global
coverage of the ice-free ocean by the start of 2006. At present, more than 3800 floats provide
systematic T/S data, with quasi (60°S-60°N latitude) global coverage down to 2000 m depth.
A full overview on in situ ocean temperature measurements is given for example in Abraham
et al. (2013).
In this section, we consider a set of 11 direct (in situ) estimates, publically available over the
entire altimetry era, to review global mean thermosteric sea level rise and, ultimately, to
construct an ensemble mean timeseries.  These data sets are:
1. CORA = Coriolis Ocean database for ReAnalysis, Copernicus Service, France
marine.copernicus.eu/, product name :
INSITU_GLO_TS_OA_REP_OBSERVATIONS_013_002_b
2. CSIRO (RSOI) = Commonwealth Scientific and Industrial Research
Organisation/Reduced-Space Optimal Interpolation, Australia
3. ACECRC/IMAS-UTAS = Antarctic Climate and Ecosystem Cooperative Research
Centre/Institute for Marine and Antarctic Studies-University of Tasmania, Australia
http://www.cmar.csiro.au/sealevel/thermal_expansion_ocean_heat_timeseries.html
4. ICCES = International Center for Climate and Environment Sciences, Institute of
Atmospheric Physics, China
http://ddl.escience.cn/f/PKFR
5. ICDC = Integrated Climate Data Center, Universit of Hamburg, Germany
6. IPRC = International Pacific Research Center, University of Hawaii, USA
http://apdrc.soest.hawaii.edu/projects/Argo/data/gridded/On_standard_levels/index-
1.html
7. JAMSTEC = Japan Agency for Marine-Earth Science and Technology, Japan
ftp://ftp2.jamstec.go.jp/pub/argo/MOAA_GPV/Glb_PRS/OI/
8. MRI/JMA = Meteorological Resarch Institute/Japan Meteorological Agency, Japan
https://climate.mri-jma.go.jp/~ishii/.wcrp/
9. NCEI/NOAA = National Centers for Environmental Information/National Oceanic
and Atmospheric Adinistration, USA
10. SIO = Scripps Institution of Oceanography, USA
Deep/abyssal: https://cchdo.ucsd.edu/
11. SIO = Scripps Institution of Oceanography, USA
Deep/abyssal: https://cchdo.ucsd.edu/ (for the abyssal ocean)

Their characteristics are presented in Table 2.






| Product/Institution | Period | Depth-integration (m) | | | | Temporal resolution / Latitudinal range | Reference |
|---|---|---|---|---|---|---|---|
| | | 0-700 | 700-2000 | 0-2000 | ≥2000 | | |
| 1 CORA | 1993-2016* | Y | Y | Y | - - - | Monthly 60°S-60°N | http://marine.copernicus.eu/services-portfolio/access-to-products/ |
| 2 CSIRO (RSOI) | 2004-2017 | Y/E (0-300) | Y/E | Y/E | - - - | Monthly 65°S-65°N | Roemmich et al. (2015); Wijffels et al. (2016) |
| 3 CSIRO/ACE CRC/ IMAS-UTAS | 1970-2017 | Y/E (0-300) | - - - | - - - | - - - | Yearly (3-yr run. mean) 65°S-65°N | Domingues et al. (2008); Church et al. (2011) |
| 4 ICCES | 1970-2016 | Y/E (0-300) | Y/E | Y/E | - - - | Yearly 89°S-89°N | Cheng and Zhu (2016); Cheng et al. (2017) |
| 5 ICDC | 1993-2016* | Y (1993) | - - - | Y (2005) | - - - | Monthly | Gouretzki and Koltermann (2007) |
| 6 IPRC | 2005-2016* | - - - | - - - | Y | - - - | Monthly | http://apdrc.soest.hawaii.edu/projects/argo |
| 7 JAMSTEC | 2005-2016* | - - - | - - - | Y | - - - | Monthly | Hosoda et al. (2008) |
| 8 MRI/JMA | 1970-2016 (rel. to 1961-1990 averages) | Y/E (0-300) | Y/E | Y/E | - - - | Yearly 89°S-89°N | Ishii et al. (2017) |
| 9 NCEI/NOAA | 1970-2016 | Y/E | Y/E | Y/E | - - - | Yearly 89°S-89°N | Antonov et al. (2005) |
| 10 SIO | 2005-2016* | - - - | - - - | Y | - - - | Monthly | Roemmich and Gilson (2009) |
| 11 SIO (Deep/abyssal) | 1990-2010 (as of 01/2018) | - - - | - - - | - - - | Y/E | Linear trend 89°S-89°N, as an aggregation of 32 | Purkey and Johnson (2010) |

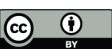



| | | | | | | deep ocean basins | |
|---|---|---|---|---|---|---|---|

*Table 2: In situ datasets and relevant information from originators and/or contributors.*


### 2.3.2 Individual estimates


All in situ estimates compiled in this study show a steady rise in global mean thermosteric sea
level, independent of depth-integration and decadal/multidecadal periods (Figure 4 and 5, left
panels). As the deep/abyssal ocean estimate only illustrates the updated version of the linear
trend from Purkey and Johnson (2010) for 1990-2010 extrapolated to 2016, it does not have
any variability superimposed.
Interannual to decadal variability during the Altimeter era (since 1993) is similar for both 0-
700 m and 700-2000 m, with larger amplitude in the upper ocean (Figure 4 and 5, right
panels). For the 0-700 m, there is an apparent change in amplitude before/after the Argo era
(since 2005), mostly due to a maximum (2-4 mm) around 2001-2004, except for one estimate.
Higher amplitude and larger spread in variability between estimates before the Argo era is a
symptom of the much sparser in situ coverage of the global ocean. Interannual variability over
the Argo era (Figures 4 and 5, right panels) is mainly modulated by El Niño Southern
Oscillation (ENSO) phases in the upper 500 m ocean, particularly for the Pacific, the largest
ocean basin (Roemmich et al., 2011; Johnson and Birnbaum, 2017).
In terms of depth contribution, on average, the upper 300 m explains the same percentage
(almost 70%) of the 0-700 m linear rate over both altimetry and Argo eras, but the
contribution from the 0-700 m to 0-2000 m varies: about 75% for 1993-2016 and 65% for
2005-2016. Thus, the 700-2000 m contribution m increases by 10% during the Argo decade,
when the number of observations within 700-2000 m has significantly increased.





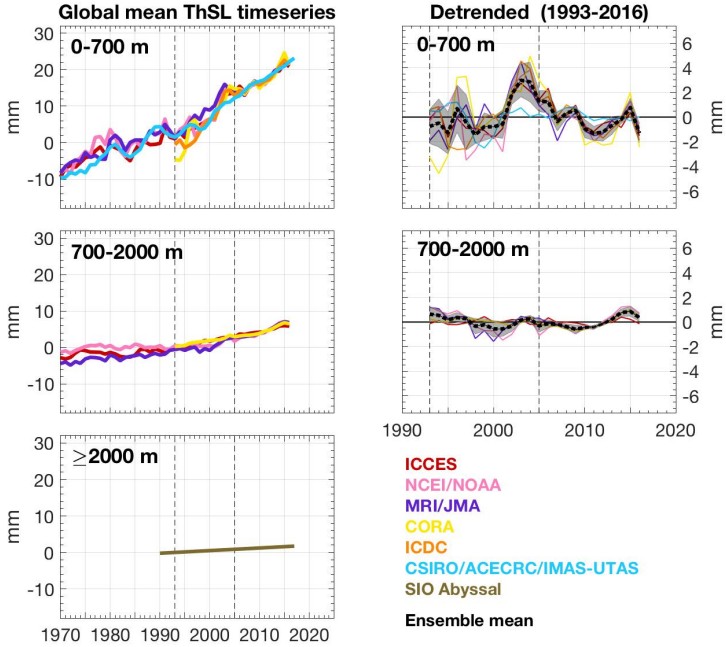


Figure 4. *Left- panels. Annual mean global mean thermosteric anomaly timseries since 1970,*
*from various research groups (colour) and for three depth-integrations: 0-700 m (top), 700-*
*2000 m (middle), and below 2000 m (bottom). Vertical dashed lines are plotted along 1993*
*and 2005. For comparison, all timeseries were offset arbitrarily. Right panels. Respective*
*linearly detrended timeseries for 1993-2016. Black bold dashed line is the ensemble mean*
*and gray shadow bar the ensemble spread (1-standard deviation). Units are mm.*

438

439

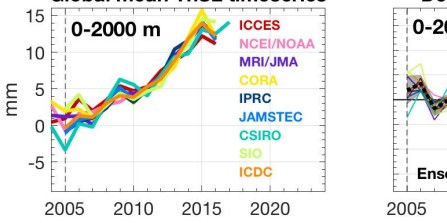
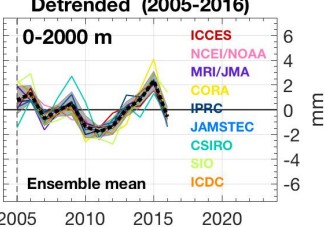

440

*Figure 5. Left- panel. Annual mean global mean thermosteric anomaly timseries since 2004,*
*from various research groups (colour) in the upper 2000 m. A vertical dashed line is plotted*
*along 2005. For comparison, all timeseries were offset arbitrarily. Right panel. Respective*
*linearly detrended timeseries for 2005-2016. Black bold dashed line is the ensemble mean*
*and gray shadow bar the ensemble spread (1-standard deviation). Units are mm.*

446

447



### 2.3.3. Ensemble mean thermosteric sea level

Given that the global mean thermosteric sea level anomaly estimates compiled for this study are not necessarily referenced to the same baseline climatology, they cannot be directly averaged together to create an ensemble mean. To circumvent this limitation, we created an ensemble mean in three steps, as explained below.

Firstly, we detrended the individual timeseries by removing a linear trend for 1993-2016 and averaged together to obtain an "ensemble mean variability timeseries". Secondly, we averaged together the corresponding linear trends of the individual estimates to obtain an "ensemble mean linear rate". Thirdly, we combined this "ensemble mean linear rate" with the "ensemble mean variability timeseries" to obtain the final ensemble mean timeseries. We applied the same steps for the Argo era (2005-2016).

To maximise the number of individual estimates used in the final full-depth ensemble mean timeseries, the three steps above were actually divided into depth-integrations and then summed. For the Argo era, we summed 0-2000 m (9 estimates) and ≥2000 m (1 estimate). For the altimetry era, we summed 0-700 m (6 estimates), 700-2000 (4 estimates) and ≥2000 m (1 estimate), although there is no statistical difference if the calculaltion was only based on the sum of 0-2000 m (4 estimates) and ≥2000 m (1 estimate). There is also no statistical difference between the full-depth ensemble mean timeseries created for the Altimeter and Argo eras during their overlapping years (since 2005).

Figure 6 shows the full-depth ensemble mean timeseries over 1993-2016 and 2005-2016. It reveals a global mean thermosteric sea level rise of about 30 mm over 1993-2016 (24 years) or about 18 mm over 2005-2016 (12 years) , with a record high in 2015. These thermosteric changes are equivalent to a linear rate of 1.32 +/- 0.4 mm/yr and 1.31 +/- 0.4 mm/yr respectively.





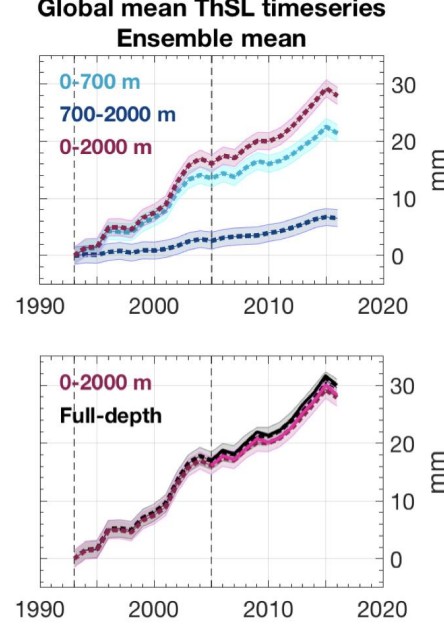

**Figure 6:** *Ensemble mean timeseries for global mean thermosteric anomaly, for three-depth integrations (top) and for 0-2000 m and full-depth (bottom). In the bottom panel, dashed lines are relative to 1993-2016 whereas solid lines are relative to 2005-2016. Errobars represent the ensemble spread (standard deviation). Units are mm.*

Figure 7 shows thermosteric sea level trends for each of the data sets used over the 1996-2016 (left panel) and 2005-2016 (right panel) time spans and different depth ranges (including full depth), as well as associated ensemble mean trends. The full depth ensemble mean trend amounts to 1.3 +/- 0.4 mm/yr over 2005-2016. It is similar to the 1993-2016 ensemble mean trend, suggesting negligible acceleration of the thermosteric component over the altimetry era.

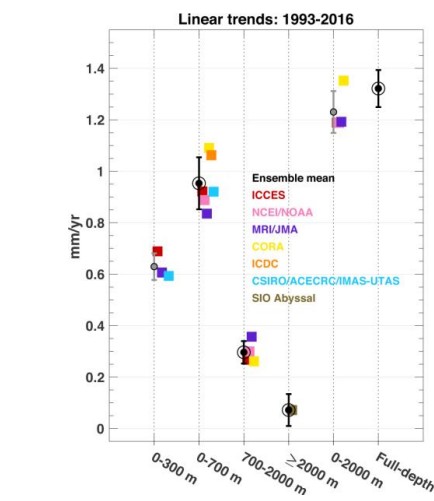
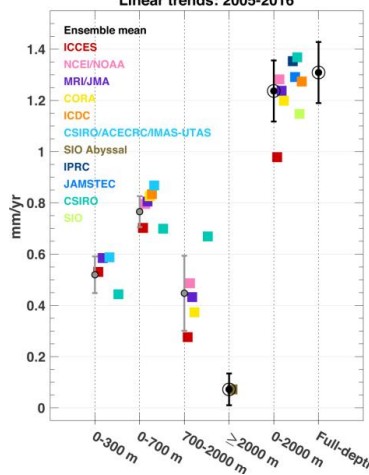



*Figure 7: Linear rates of global mean thermosteric sea level for depth-integrations (x-axis), for*
*invididual estimates and ensemble means, over 1993-2016 (left) and 2005-2016 (right). Ensemble*
*mean rates with a black circle were used in the estimation of the timeseries described in Section 2.3.4.*
*Errorbars are standard deviation due to spread of the estimates except for ≥2000 m. Units are mm/yr.*

**2.4 Glaciers**
Glaciers have strongly contributed to sea-level rise during the 20[th] century – around 40% -
and will continue to be an important part of the projected sea-level change during the 21[st]
century – around 30% (Kaser et al., 2006, Church et al., 2013, Gardner et al., 2013, Marzeion
et al., 2014, Zemp et al., 2015; Huss and Hock, 2015). Because glaciers are time-integrated
dynamic systems, a response lag of at least 10 years to a few hundred years is observed
between changes in climate forcing and glacier shape, mainly depending on glacier length and
slope (Johannesson et al., 1989, Bahr et al., 1998). Today, glaciers are globally (a notable
exception is the Karakoram/Kunlun Shan region, e.g. Brun et al., 2017) in a strong
disequilibrium with the current climate and are loosing mass, due essentially to the global
warming in the second half of the 20[th] century (Marzeion et al., 2018).
Global glacier mass changes are derived from in situ measurements of glacier mass changes
or glacier length changes. Remote sensing methods measure elevation changes over entire
glaciers based on differencing digital elevation models (DEMs) from satellite imagery
between two epochs (or at points from repeat altimetry), surface flow velocities for
determination of mass fluxes, and glacier mass changes from space-based gravimetry. Mass
balance modeling driven by climate observations is also used (Marzeion et al., 2017 reviews
of these different methods).
Glacier contribution to sea level is primarily the result of their surface mass balance and
dynamic adjustment, plus iceberg discharge and frontal ablation (below sea level) in the case
of marine-terminating glaciers. The sum of worldwide glacier mass balances (MBs) does not
correspond to the total glacier contribution to sea-level change for the following reasons:
- Glacier ice below sea level does not contribute to sea-level change, apart from a
small lowering when replacing ice with seawater of a higher density. Total volume of glacier
ice below sea level is estimated to be 10 – 60 mm sea-level equivalent (SLE, Huss and
Farinotti, 2012, Haeberli and Linsbauer, 2013, Huss and Hock, 2015).
- Incomplete transfer of melting ice from glaciers to the ocean: meltwater stored in
lakes or wetlands, meltwater intercepted by natural processes and human activities (e.g.



drainage to lakes and aquifers in endorheic basins, impoundment in reservoirs, agriculture use
of freshwater, Loriaux and Casassa, 2013, Käab et al., 2015).
Despite considerable progress in observing methods and spatial coverage (Marzeion et al.,
2017), estimating glacier contribution to sea-level change remains challenging due to the
following reasons:
- Number of regularly observed glaciers (in the field) remains very low (0.25% of the
200 000 glaciers of the world have at least one observation and only 37 glaciers have long-
term observations, Zemp et al. 2015).
- Uncertainty of the total glacier ice mass remains high (Figure 8, Grinsted et al., 2013,
Pfeffer et al., 2014, Farinotti et al., 2017, Frey et al. 2014).
- Uncertainties in glacier inventories and DEMs are not negligible. Sources of
uncertainties include debris-covered glaciers, disappearance of small glaciers, positional
uncertainties, wrongly mapped seasonal snow, rock glaciers, voids and artifacts in DEMs
(Paul et al., 2004, Bahr and Radić, 2012).
- Uncertainties of satellite retrieval algorithms from space-based gravimetry and
regional DEM differencing are still high, especially for global estimates (Gardner et al. 2013,
Marzeion et al., 2017, Chambers et al., 2017).
- Uncertainties of global glacier modeling (e.g. initial conditions, model assumptions
and simplifications, local climate conditions, Marzeion et al., 2012).
- Knowledge about some processes governing mass balance (e.g. wind redistribution
and metamorphism, sublimation, refreezing, basal melting) and dynamic processes (e.g. basal
hydrology, fracking, surging) remains limited (Farinotti et al., 2017).
An annual assessment of glacier contribution to sea-level change is difficult to perform (from
ground-based or space-based observations except space-based gravimetry), due to the sparse
and irregular observation of glaciers, and the difficulty of assessing accurately the annual
mass balance variability. Global annual averages are highly uncertain because of the sparse
coverage, but successive annual balances are uncorrelated and therefore averages over several
years are known with greater confidence.

### 593 *2.4.1 Glacier datasets*

The following datasets are considered, with a focus on the trends of annual mass changes:
1. Update of Gardner et al., 2013 (Reager et al., 2016), from satellite gravimetry and
altimetry, and glaciological records, called G16.



2. Update of Marzeion et al, 2012 (Marzeion et al., 2017), from global glacier
modeling and mass balance observations, called M17.
3. Update of Cogley (2009) (Marzeion et al., 2017), from geodetic and direct mass-
balance measurements, called C17.
4. Update of Leclercq et al., 2011 (Marzeion et al., 2017), from glacier length changes,
called L17.
5. Average of GRACE-based estimates of Marzeion et al. (2017), from spatial
gravimetry measurements, called M17-G.
In general it is not possible to align measurements of glacier mass balance with the calendar.
Most in-situ measurements are for glaciological years that extend between successive annual
minima of the glacier mass at the end of the summer melt season. Geodetic measurements
have start and end dates several years apart and distributed irregularly through the calendar
year; some are corrected to align with annual mass minima but most are not. Consequently
measurements discussed here for 1993-2016 (the altimetry era) and 2005-2016 (the GRACE
and Argo era) are offset by up to a few months from the nominal calendar years.
Peripheral glaciers around the Greenland and Antarctic ice sheets are not treated in detail in
this section (see sections 2.5 and 2.6 for mass-change estimates that combine the peripheral
glaciers with the Greenland Ice Sheet and Antarctic Ice Sheet respectively). This is primarily
because of the lack of observations (especially ground-based measurements) and also because
of the high spatial variability of mass balance in those regions, and the slightly different
climate (e.g. precipitation regime) and processes (e.g. refreezing). In the past, these regions
have often been neglected. However Radić and Hock (2010) estimated the total ice mass of
peripheral glaciers around Greenland and Antarctica as 191 +/- 70 mm SLE, with an actual
contribution to sea-level rise of around 0.23 +/- 0.04 mm/yr (Radić and Hock, 2011). Gardner
et al. (2013) found a contribution from Greenland and Antarctic peripheral glaciers equal to
0.12 +/- 0.05 mm/yr.
Note that some new or updated datasets for peripheral glaciers surrounding polar ice sheets
are under development and would hopefully be available in coming years in order to
incorporate Greenland and Antarctic peripheral glaciers in the estimates of global glacier
mass changes.

***2.4.2 Methods***
No globally complete observational dataset exists for glacier mass changes (except GRACE
estimates, see below). Any calculation of the global glacier contribution to sea-level change



has to rely on spatial interpolation or extrapolation or both, or to consider limited knowledge
of responses to climate change (due to the heterogeneous spatial distribution of glaciers
around the world). Consequently, most observational methods to derive glacier sea-level
contribution must extend local observations (in situ or satellite) to a larger region. Thanks to
the recent global glacier outline inventory (Randolph Glacier Inventory – RGI – first release
in 2012) as well as global climate observations, glacier modeling can now also be used to
estimate the contribution of glaciers to sea level (Marzeion et al., 2012, Huss and Hock, 2015,
Maussion et al., 2018, subm.). Still, those global modeling methods need to globalize local
observations and glacier processes which require fundamental assumptions and
simplifications. Only GRACE-based gravimetric estimates are global but they suffer from
large uncertainties in retrieval algorithms (signal leakage from hydrology, GIA correction)
and coarse spatial resolution, not resolving smaller glacierized mountain ranges or those
peripheral to the Greenland ice sheet.
DEM differencing method is not yet global, but regional, and can hopefully in the near future
be applied globally. This method needs also to convert elevation changes to mass changes
(using assumptions on snow and ice densities). In contrast, very detailed glacier surface mass
balance and glacier dynamic models are today far from being applicable globally, mainly due
to the lack of crucial observations (e.g., meteorological data, glacier surface velocity and
thickness) and of computational power for the more demanding theoretical models. However,
somewhat simplified approaches are currently developed to make best use of the steadily
increasing datasets. Modeling-based estimates suffer also from the large spread in estimates of
the actual global glacier ice mass (Figure 8). The mean value is 469 +/- 146 mm SLE, with
recent studies converging towards a range of values between 400 and 500 mm SLE global
glacier ice mass. But as mentioned above, a part of this ice mass will not contribute to sea
level.





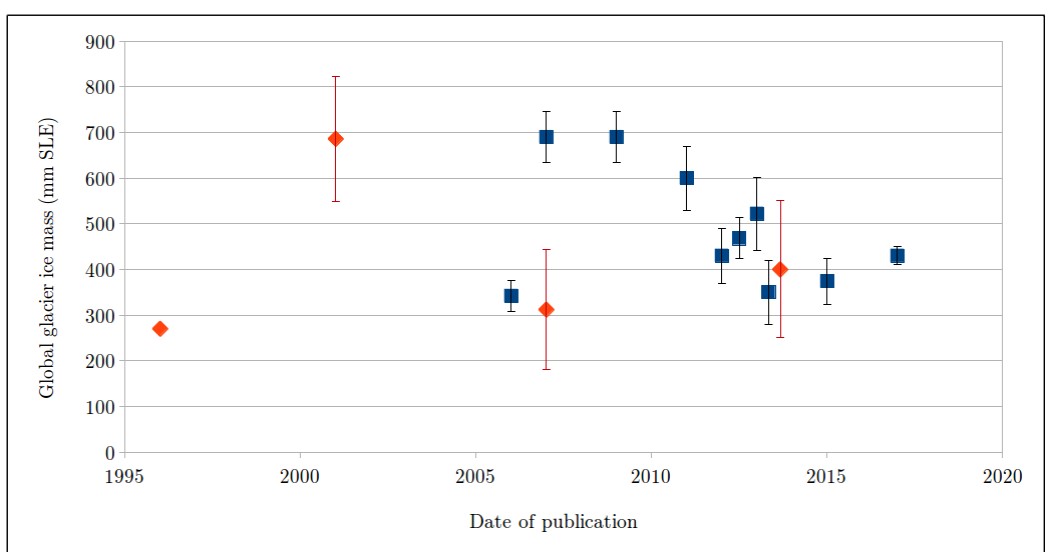

*Fig 8. Evolution of global glacier ice mass estimates (red marks are IPCC reports). Note that*
*Antarctica and Greenland peripheral glaciers are taken into account in this figure.*

**2.4.3 Results (trends)**
Table 3 presents most recent estimates of trends in global glacier mass balances.

|  | 1993 – 2016 mm/yr SLE | 2005 – 2016 mm/yr SLE |
|---|---|---|
| G16 |  | 0.70 +- 0.070[a] |
| M17 | 0.68 +- 0.032 | 0.80 +- 0.048 |
| C17 | 0.63 +- 0.070 | 0.75 +- 0.070[b] |
| L17 |  | 0.84 +- 0.640[c] |
| M17-G |  | 0.61 +- 0.070[d] |


*Table 3: All data are in mm/yr SLE. [a] The time period of G16 is 2002 – 2014. [b] The time*
*period of C17 is 2003 – 2009. [c] The time period of L17 is 2003 – 2009. [d] The time period of*
*M17-G is 2002/2005 – 2013/2015 because this value is an average of different estimates.*

The ensemble mean contribution of glaciers to sea-level rise for the time period 1993 – 2016
is 0.65 +/- 0.051 mm/yr SLE and 0.74 +/- 0.18 mm/yr for the time period 2005 – 2016



(uncertainties are averaged). Different studies refer to different time periods. However,
because of the probable low variability of global annual glacier changes, compared to other
components of the sea-level budget, averaging trends for slightly different time periods is
appropriate.
The main source of uncertainty is that the vast majority of glaciers are unmeasured, which
makes interpolation or extrapolation necessary, whether for in situ or satellite measurements,
and for glacier modeling. Other main contributions to uncertainty in the ensemble mean stem
from methodological differences, such as the downscaling of atmospheric forcing required for
glacier modeling, the separation of glacier mass change to other mass change in the spatial
gravimetry signal and the derivation of observational estimates of mass change from different
raw measurements (e.g. length and volume changes, mass balance measurements and geodetic
methods) all with their specific uncertainties.

## 683  2.5 Greenland

Ice sheets are the largest potential source of future sea level rise (SLR) and represent the
largest uncertainty in projections of future sea level. Almost all land ice (~99.5%) is locked in
the ice sheets, with a volume in sea level equivalent/SLE terms of 7.4 m for Greenland, and
58.3 m for Antarctica. It has been estimated that approximately 25% to 30% f total land ice
contribution to sea level rise over the last decade has been from the Greenland ice sheet (e.g.
Dieng et al., 2017, Box and Colgan, 2017).
There are three main methods that can be used to estimate the mass balance of the Greenland
ice sheet: (1) measurement of changes in elevation of the ice surface over time (dh/dt) either
from imagery or altimetry; (2) the mass budget or Input-Output Method (IOM) which
involves estimating the difference between the surface mass balance and ice discharge; and
(3) consideration of the redistribution of mass via gravity anomaly measurements which only
became viable with the launch of GRACE in 2002. Uncertainties due to, for example, the GIA
correction are small in Greenland compared to Antarctica: on the order of ±20 Gt/yr mass
equivalent (Khan et al., 2016). Prior to 2003, mass trends are reliant on IOM and altimetry.
Both techniques have limited sampling in time and/or space for parts of the satellite era
(1992-2002) and errors for this earlier period are, therefore, higher (van den Broeke et al.,
2016, Hurkmans et al., 2014).
The consistency between the three methods mentioned above was demonstrated for Greenland
by (Sasgen et al., 2012) for the period 2003-2009. Ice sheet wide estimates showed excellent



agreement although there was less consistency at a basin scale. We have, therefore, high
confidence and relatively low uncertainties in the rates for the Greenland ice sheet in the
satellite era.

**2.5.1 Datasets considered for the assessment**
This assessment of sea level budget contribution from the Greenland ice sheet considers the
following datasets:

| Reference | Time period | Method |
|---|---|---|
| Update from Barletta et al. (2013) | 2003-2016 | GRACE |
| Groh and Horwath (2016) | 2003-2015 | GRACE |
| Update from Luthcke et al. (2013) | 2003-2015 | GRACE |
| Update from Sasgen et al. (2012) | 2003-2016 | GRACE |
| Update from Schrama et al. (2014) | 2003-2016 | GRACE |
| Update from (van den Broeke et al., 2016) | 1993-2016 | Input/output Method (IOM) |
| Wiese et al. (2016) | 2003-2016 | GRACE |
| Update from Wouters et al. (2008) | 2003-2016 | GRACE |

*Table 4. The datasets received for consideration in this assessment*

**2.5.2. Methods and analyses**
All but one of these datasets are based on GRACE data and therefore provide annual time
series from ~2002 onwards. The one exception uses IOM (van den Broeke et al., 2016) to
give an annual mass time series for a longer time period (1993 onwards).
Notwithstanding this, each group has chosen their own approach to estimate mass balance
from GRACE observations. As the aim of this Global Sea Level Budget assessment is to
compile existing results (rather than undertake new analyses), we have not imposed a specific
methodology. Instead, we asked for the contributed datasets to reflect each group's 'best
estimate' of annual trends for Greenland using the method(s) they have published.
Greenland contains glaciers and ice caps around the margins of the main ice sheet, often
referred to as peripheral GIC (PGIC), which are a significant proportion of the total mass
imbalance (circa 15-20%) (Bolch et al., 2013). Some studies consider the mass balance of the
ice sheets and the PGIC separately but there has been, in general, no consistency in the



treatment of PGIC and many studies do not specify if they are included or excluded from the
total. The GRACE satellites have an approximate spatial resolution of 300 km and the large
number of studies that use GRACE, by default, include all land ice within the domain of
interest. For this reason, the results below for Greenland mass trends all include PGIC.
From these datasets, for each year from 1993 to 2015 (and 2016 where available), we have
calculated an average change in mass (calculated as the weighted mean based on the stated
error value for each year) and an error term. Prior to 2003, the results are based on just one
dataset (van den Broeke et al., 2016).

*2.5.3 Results*

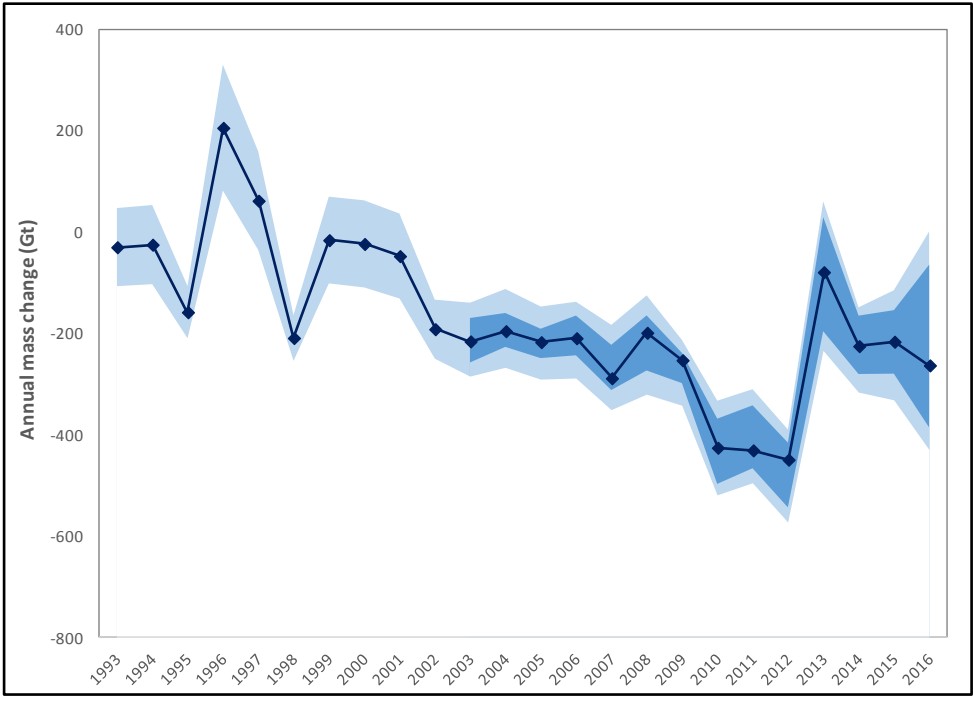


*Figure 9. Greenland annual mass change 1993 to 2016. The medium blue region shows the*
*range of estimates from the datasets listed in Table 1. The lighter blue region shows the range*
*of estimates when stated errors are included, to provide upper and lower bounds. The dark*
*blue line shows the weighted mean mass trend.*








| Year | Δ mass (Gt/yr) | Error (Gt.yr) | σ (Gt) |
|---|---|---|---|
| 1993 | -30 | 76 | |
| 1994 | -25 | 77 | |
| 1995 | -159 | 51 | |
| 1996 | 205 | 123 | |
| 1997 | 61 | 97 | |
| 1998 | -209 | 45 | |
| 1999 | -16 | 85 | |
| 2000 | -24 | 85 | |
| 2001 | -48 | 83 | |
| 2002 | -192 | 58 | |
| 2003 | -216 | 13 | 28 |
| 2004 | -196 | 12 | 24 |
| 2005 | -218 | 13 | 21 |
| 2006 | -210 | 12 | 29 |
| 2007 | -289 | 10 | 31 |
| 2008 | -199 | 11 | 39 |
| 2009 | -253 | 11 | 21 |
| 2010 | -426 | 9 | 42 |
| 2011 | -431 | 9 | 47 |
| 2012 | -450 | 10 | 41 |
| 2013 | -80 | 13 | 76 |
| 2014 | -225 | 13 | 38 |
| 2015 | -217 | 13 | 48 |
| 2016 | -263 | 23 | 123 |
| Average estimate 1993-2015 | -167 | 54 | |
| Average estimate 1993-2016 | -171 | 53 | |
| Average estimate 2005-2015 | -272 | 11 | |
| Average estimate 2005-2016 | -272 | 13 | |




*Table 5. Annual time series of Greenland mass change (GT/yr, negative values mean decreasing mass). Δ mass is calculated as the weighted mean based on the stated error value for each year. The error for each year is calculated as the mean of all stated 1-sigma errors divided by sqrt(N) where N is the number of datasets available for that year, assuming that the errors are uncorrelated. The standard deviation (σ) is also given to illustrate the level of agreement between datasets for each year when multiple datasets are available (2003 onwards).*

There is generally a good level of agreement between the datasets (Figure 9), and taken together they provide an average estimate of 171 Gt/yr of ice mass loss (or sea level budget contribution) from Greenland for the period 1993 to 2016, increasing to 272 Gt/yr for the period 2005 to 2016 (Table 5).

All the datasets illustrate the previously documented accelerating mass loss up to 2012 ((Rignot et al., 2011); (Velicogna, 2009)). In 2012, the ice sheet experienced exceptional surface melting reaching as far as the summit (Nghiem et al., 2012) and a record mass loss since, at least 1958, of over 400 Gt (van den Broeke et al., 2016) . The following years, however, show a reduced loss (not more than 270 Gt in any year). Inclusion of the years since 2012 in the 2005-2016 trend estimate reduces the overall rate of mass loss acceleration and its statistical significance. There is greater divergence in the GRACE time series for 2016. We associate this with the degradation of the satellites as they came towards the end of their mission. For 2005-2012, it might be inferred that there is a secular trend towards greater mass loss and from 2010-2012 the value is relatively constant. Inter-annual variability in mass balance of the ice sheet is driven, primarily, by the surface mass balance (i.e. atmospheric weather) and it is apparent that the magnitude of this year to year variability can be large: exceeding 360 Gt (or 1 mm sea level equivalent) between 2012 and 2013. Caution is required, therefore, in extrapolating trends from a short record such as this.

## 2.6 Antarctica

The annual turn over of mass of Antarctica is about 2,200 Gt/yr (over 6 mm/yr of SLE), 5 times larger than in Greenland (Wessem et al. 2017). In contrast with Greenland, ice and snow melt have a negligible influence on Antarctica's mass balance which is therefore completely controlled by the balance between snowfall accumulation in the drainage basins and ice discharge along the periphery. The continent is also 7 times larger than Greenland,



which makes satellite techniques absolutely essential to survey the continent. Interannual
variations in accumulation are large in Antarctica, showing decadal to multi-decadal
variability, so that many years of data are required to extract trends, and missions limited to
only a few years may produce misleading results (e.g. Rignot et al., 2011).
As in Greenland, the estimation of the mass balance has employed a variety of techniques,
including 1) the gravity method with GRACE since April 2002 until the end of the mission in
late 2016; 2) the input and output method (IOM) using a series of Landsat and Synthetic-
Aperture Radar (SAR) satellites for measuring ice motion along the periphery (Rignot et al.,
2011), ice thickness from airborne depth radar sounders such as Operation IceBridge
(Leuschen et al., 2014), and reconstructions of surface mass balance using regional
atmospheric climate models constrained by re-analysis data (RACMO, MAR and others); and
3) radar/laser altimetry method which mix various satellite altimeters and correct ice elevation
changes with density changes from firm models. The largest uncertainty in the GRACE
estimate in Antarctica is the GIA which is larger than in Greenland and a large fraction of the
observed signal. The IOM method compares two large numbers with large uncertainties to
estimate the mass balance as the difference. In order to detect an imbalance at the 10% level,
surface mass balance and ice discharge need to be estimated with a precision typically of 5 to
7%. The altimetry method is limited to areas of shallow slope, hence is difficult to use in the
Antarctic Peninsula and in the deep interior of the Antarctic continent due to unknown
variations of the penetration depth of the signal in snow/firn. The only method that expresses
the partitioning of the mass balance between surface processes and dynamic processes is the
IOM method (e.g. Rignot et al., 2011). The gravity method is an integrand method which does
not suffer from the limitations of SMB models but is limited in spatial resolution (e.g.
Velicogna et al., 2014). The altimetry method provides independent evidence of changes in
ice dynamics, e.g. by revealing rapid ice thinning along the ice streams and glaciers revealed
by ice motion maps, as opposed to large scale variations reflecting a variability in surface
mass balance (McMillan et al., 2014).
All these techniques have improved in quality over time and have accumulated a decade to
several decades of observations, so that we are now able to assess the mass balance of the
Antarctic continent using methods with reasonably low uncertainties, and multiple lines of
evidence as the methods are largely independent, which increases confidence in the results.
There is broad agreement in the mass loss from the Antarctic Peninsula and West Antarctica;
most residual uncertainties are associated with East Antarctica as the signal is relatively small



compared to the uncertainties, although most estimates tend to indicate a low contribution to
sea level (e.g. Shepherd et al., 2012).

**2.6.1 Datasets considered for the assessment**
This assessment consider the following datasets:

| Reference | Method | 2005-2015 SLE Trend (mm/yr) | 1993-2015 SLE Trend (mm/yr) |
|---|---|---|---|
| Update from Martín-Español et al. (2016) | Joint inversion GRACE/altimetry /GPS | 0.43±0.07 | - |
| Update from Fosberg et al. (2017) | Joint inversion GRACE/CryoSat | 0.31±0.02 | - |
| Update from Groh and Horwath (2016) | GRACE | 0.32±0.11 | - |
| Update from Luthcke et al. (2013) | GRACE | 0.36±0.06 | - |
| Update from Sasgen et al. (2013) | GRACE | 0.47±0.07 | - |
| Update from Velicogna et al. (2014) | GRACE | 0.33±0.08 | - |
| Update from Wiese et al. (2016) | GRACE | 0.39±0.02 | - |
| Update from Wouters et al. (2013) | GRACE | 0.41±0.05 | - |
| Update from Rignot et al. 2011 | Input/output method (IOM) | 0.46±0.05 | 0.25±0.1 |
| Update from Schrama et al. (2014); version 1 | GRACE ICE6G GIA model | 0.47±0.03 | |
| Update fromSchrama et al. (2014); version 2 | GRACE Updated GIA models | 0.33±0.03 | |

*Table 6. The datasets received for consideration in this assessment including trend for the*
*2005-2015 and 1993-2015 expressed in mm/yr SLE. Positive values mean positive*
*contribution to sea level (i.e. sea level rise)*



### 2.6.2 Methods and analyses

The datasets used in this assessment are Antarctica mass balance time series generated using different approaches. Two estimates are a joint inversion of GRACE/altimetry/GPS data Martín-Español et al. (2016), and GRACE and CryoSat data Fosberg et al. (2017). Two methods are mascon solutions obtained from the GRACE intersatellite range-rate measurements (Luthcke et al., 2013; Wiese et al., 2016), three estimates use the GRACE spherical harmonics solutions (Velicogna et al., 2014; Wiese et al., 2016; Wouters et al., 2013) and one gridded GRACE products (Sasgen et. al., 2013).

All GRACE time series were provided as monthly time series except for the one using the Martín-Español et al. (2016) method that were provided as annual estimates. In addition, different groups use different GIA corrections, therefore the spread of the trend solutions represents also the error associated to the GIA correction which, in Antarctica, is the largest source of uncertainty. Sasgen et al. (2013) used their own GIA solution (Sasgen et al., 2017), Martín-Español et al. (2016) as well, Luthcke et al., (2013), Velicogna et al. (2014) and Groh and Horwath (2016) used IJ05-R2 (Ivins et al., 2013), Wouter et al. (2013) used Whitehouse et al. (2012), and Wise et al. (2016) used A et al. (2013). In addition, Groh and Horwath (2016) did not include the peripheral glaciers and ice caps, while all other estimates do.

Table 6 shows the Antarctic sea level contribution to sea level during 2005-2015 from the different GRACE solutions, for the input and output method (IOM), and from the altimetry method. We a single dataset, OIB (Rignot et al. 2011) that provides trends for the period 1993-2015. For the period 2005-2015, we calculated the annual sea level contribution from Antarctica using GRACE, IOM and altimetry estimates (Table 7).

As we are interested in evaluating the long term trend and inter-annual variability of the Antarctic contribution to sea level, for each GRACE datasets available in monthly time series, we first removed the annual and sub-annual components of the signal by applying a 13-month averaging filter and we then used the smoothed time series to calculate to annual mass change. Figure 10 shows the annual sea level contribution from Antarctica calculated from the GRACE derived estimates and for the Input and Output Method (IOM). The GRACE mean annual estimates are calculated as the mean of the annual contributions from the different groups, and the associated error calculated as the sum of the spread of the annual estimates and the mean annual error.





### 2.6.3 Results














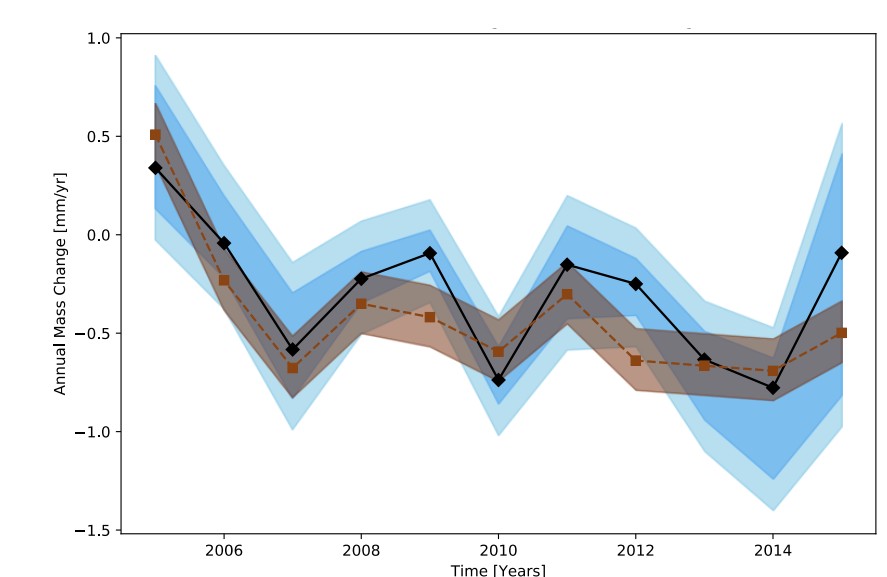

*Figure 10. Antarctic annual sea level contribution during 2005 to 2015. The black squares*
*are the mean annual sea level calculated using the GRACE datasets listed in Table 2.6a. The*
*darker blue band shows the range of estimates from the datasets. The light blue band account*
*for the error in the different GRACE estimates. The brown squares are the annual sea level*
*contribution calculated using the input output method (updated from Rignot et al. 2011), the*
*light brown band is the associated error.*

| Year | GRACE mm/yr SLE | IOM mm/yr SLE | Mean mm/yr SLE |
|---|---|---|---|
| 2005 | -0.34±0.47 | -0.51±0.16 | -0.42±0.31 |
| 2006 | 0.04±0.36 | 0.23±0.16 | 0.14±0.26 |
| 2007 | 0.58±0.42 | 0.68±0.16 | 0.63±0.29 |
| 2008 | 0.22±0.29 | 0.35±0.16 | 0.29±0.22 |
| 2009 | 0.09±0.26 | 0.42±0.16 | 0.26±0.21 |
| 2010 | 0.74±0.30 | 0.59±0.16 | 0.67±0.23 |





| | | | |
|---|---|---|---|
| 2011 | 0.15±0.39 | 0.30±0.16 | 0.23±0.27 |
| 2012 | 0.25±0.30 | 0.64±0.16 | 0.44±0.23 |
| 2013 | 0.63±0.38 | 0.67±0.16 | 0.65±0.27 |
| 2014 | 0.78±0.46 | 0.69±0.16 | 0.73±0.31 |
| 2015 | 0.09±0.77 | 0.50±0.16 | 0.29±0.46 |
| Average estimate 2005-2015 | 0.38±0.06 | 0.46±0.05 | 0.42±0.06 |

*Table 7. Annual sea level contribution from Antarctica during 2005-2015 from GRACE and*
*input and Output Method (IOM) calculated as described above and expressed in mm/yr SLE.*
*Also shown is the mean of the estimate from the two methods, associated errors are the mean*
*of the two estimate errors. Positive values mean positive contribution to sea level (i.e. sea*
*level rise)*

There is generally broad agreement between the GRACE datasets (Figure 10), as most of the
differences between GRACE estimates are caused by differences in GIA correction. We find
a reasonable agreement between GRACE and the IOM estimates although the IOM estimates
indicate higher losses. Taken together, these estimates yield an average of 0.42 mm/yr sea
level budget contribution from Antarctica for the period 2005 to 2015 (Table 7) and 0.25
mm/yr sea level for the time period 1993-2005, where the latter value is based on IOM only.
All the datasets illustrate the previously documented accelerating mass loss of Antarctica
((Rignot et al., 2011) ; (Velicogna, 2009)). In 2005-2010, the ice sheet experienced ice mass
loss driven by an increase in mass loss in the Amundsen Sea sector of West Antarctica
(Mouginot et al., 2014). The following years showed a reduced increase in mass loss, as
colder ocean conditions prevailed in the Amundsen Sea Embayment (ASE) sector of West
Antarctica in 2012-2013 which reduced the melting of the ice shelves in front of the glaciers
(Dutrieux et al., 2014). Divergence in the GRACE time series is observed after 2015 due to
the degradation of the satellites towards the end of the mission.
The large inter-annual variability in mass balance in 2005-2015, characteristic of Antarctica,
nearly masks out the trend in mass loss, which is more apparent in the longer time series than
in short time series. The longer record highlights the pronounced decadal variability in ice
sheet mass balance in Antarctica, demonstrating the need for multi decadal time series in
Antarctica, which have been obtained only by IOM and Altimetry. The inter-annual
variability in mass balance is driven almost entirely by surface mass balance processes. The





mass loss of Antarctica, about 200 Gt/yr in recent years, is only about 10% of its annual turn
over of mass (2,200 Gt/yr), in contrast with Greenland where the mass loss has been growing
rapidly to nearly 100% of the annual turn over of mass. This comparison illustrates the
challenge of detecting mass balance changes in Antarctica, but at the same time that satellite
techniques and their interpretation have made tremendous progress over the last 10 years,
producing realistic and consistent estimates of the mass using a number of independent
methods.

## 2.7 Terrestrial Water Storage


Human transformations of the Earth's surface have impacted the terrestrial water balance,
including continental patterns of river flow and water exchange between land, atmosphere and
ocean, ultimately affecting global sea level. For instance, massive impoundment of water in
man-made reservoirs has reduced the direct outflow of water to the sea through rivers, while
groundwater abstractions, wetland and lake storage losses, deforestation and other land use
changes have caused changes to the terrestrial water balance, including changing
evapotranspiration over land, leading to net changes in land-ocean exchanges (Chao et al.,
2008; Wada et al., 2012a,b; Konikow 2011; Church et al., 2013; Doll et al. ???). Overall, the
combined effects of direct anthropogenic processes have reduced land water storage,
increasing the rate of sea level rise (SLR) by 0.3-0.5 mm yr$^{-1}$ during recent decades (Church
et al., 2013; Gregory et al., 2013; Wada et al., 2016). Additionally, recent work has shown
that climate driven changes in water stores can perturb the rate of sea level change over
interannual to decadal time scales, making global land mass budget closure sensitive to
varying observational periods (Cazenave et al., 2014; Dieng et al., 2015; Reager et al., 2016;
Rietbroek et al., 2016). Here we discuss each of the major component contributions from
land, with a summary in Table 8, and estimate the net terrestrial water storage (TWS)
contribution to sea level.

### *2.7.1 Direct anthropogenic changes in terrestrial water storage*



#### *Water impoundment behind dams*


Wada et al. (2016) built on work by Chao et al. (2008) to combine multiple global reservoir
storage data sets in pursuit of a quality-controlled global reservoir database. The result is a list
of 48064 reservoirs that have a combined total capacity of 7968 km$^3$. The time history of



growth of the total global reservoir capacity reflects the history of the human activity in dam
building. Applying assumptions from Chao et al. (2008), Wada et al. (2016) estimated that
humans have impounded a total of 10,416 km$^3$ of water behind dams, accounting for a
cumulative 29 mm drop in global mean sea level. From 1950 to 2000 when global dam-
building activity was at its highest, impoundment contributed to the average rate of sea level
change at -0.51 mm/year. This was an important process in comparison to other natural and
anthropogenic sources of sea level change over the past century, but has now largely slowed
due to a global decrease in dam building activity.

*Global groundwater depletion*
Groundwater currently represents the largest secular trend component to the land water
storage budget. The rate of groundwater depletion (GWD) and its contribution to sea level has
been subject to debate (Gregory et al., 2013; Taylor et al., 2013). In the IPCC AR4 (Solomon
et al., 2007), the contribution of non-frozen terrestrial waters (including GWD) to sea-level
variation was not considered due to its perceived uncertainty (Wada, 2016). Observations
from GRACE opened a path to monitor total water storage changes including groundwater in
data scarce regions (Strassberg et al., 2007; Rodell et al. 2009; Tiwari et al. 2009; Jacob et al.,
2012; Shamsudduha et al., 2012; Voss et al., 2013). Some studies have also applied global
hydrological models in combination with the GRACE data (see Wada et al., 2016 for a
review).
Earlier estimates of GWD contribution to sea level range from 0.075 mm yr$^{-1}$ to 0.30 mm yr$^{-1}$
(Sahagian et al., 1994; Gornitz, 1995, 2001; Foster and Loucks, 2006). More recently, Wada
et al. (2012b), using hydrological modelling, estimated that the contribution of GWD to
global sea level increased from 0.035 (±0.009) to 0.57 (±0.09) mm yr$^{-1}$ during the 20$^{th}$ century
and projected that it would further increase to 0.82 (±0.13) mm yr$^{-1}$ by 2050. Döll et al. (2014)
used hydrological modeling, well observations, and GRACE satellite gravity anomalies to
estimate a 2000–2009 global GWD of 113 km$^3$ yr$^{-1}$ (0.314 mm yr$^{-1}$). This value represents the
impact of human groundwater withdrawals only and does not consider the effect of climate
variability on groundwater storage. A study by Konikow (2011) estimated global GWD to be
145 (±39) km$^3$ yr$^{-1}$ (0.41 ±0.1 mm yr$^{-1}$) during 1991-2008 based on measurements of changes
in groundwater storage from in situ observations, calibrated groundwater modelling, GRACE
satellite data and extrapolation to unobserved aquifers.
An assumption of most existing global estimates of GWD impacts on sea level change is that
nearly 100% of the GWD ends up in the ocean. However, groundwater pumping can also

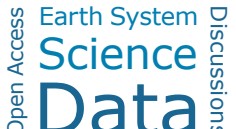



perturb regional climate due to land-atmosphere interactions (Lo and Famiglietti, 2013). A
recent study by Wada et al. (2016) used a coupled land-atmosphere model simulation to track
the fate of water pumped from underground and found it more likely that ~80% of the GWD
ends up in the ocean over the long-term, while 20% re-infiltrates and remains in land storage.
They estimated an updated contribution of GWD to global SLR ranging from 0.02 (±0.004)
mm yr$^{-1}$ in 1900 to 0.27 (±0.04) mm yr$^{-1}$ in 2000 (Figure 11). This indicates that previous
studies had likely overestimated the cumulative contribution of GWD to global SLR during
the 20th century and early 21st century by 5-10 mm.

*Land cover and land-use change*
Humans have altered a large part of the land surface, replacing 33% (Vitousek et al., 1997) or
even 41 % (Sterling et al., 2013) of natural vegetation by anthropogenic land cover such as
crop fields or pasture. Such land cover change can affect terrestrial hydrology by changing the
infiltration-to-runoff ratio, and can impact subsurface water dynamics by modifying recharge
and increasing groundwater storage (Scanlon et al., 2007). The combined effects of
anthropogenic land cover changes on land water storage can be quite complex.  Bosmans et
al. (2016), in a modelling study with PCR-GLOBWB, estimated that land cover change has
contributed to a discharge increase of 1058 km$^3$ yr$^{-1}$, on the same order of magnitude as the
effect of human water use. These recent model results suggest that land-use change is an
important topic for further investigation in the future. So far, this contribution remains highly
uncertain.

*Deforestation/afforestation*
At present, large losses in tropical forests and moderate gains in temperate-boreal forests
result in a net reduction of global forest cover (FAO, 2015; Keenan et al., 2015; MacDicken,
2015; Sloan and Sayer, 2015). Net deforestation releases carbon and water stored in both
biotic tissues and soil, which leads to sea level rise through three primary processes:
deforestation-induced runoff increases (Gornitz et al., 1997), carbon loss-related decay and
plant storage loss, and complex climate feedbacks (Butt et al., 2011; Chagnon and Bras, 2005;
Nobre et al., 2009; Shukla et al., 1990; Spracklen et al., 2012). Due to these three causes, and
if uncertainties from the land-atmospheric coupling are excluded, a summary by Wada et al.
(2016) suggests that the current net global deforestation leads to an upper-bound contribution
of ~0.035 mm/yr SLE.



*Wetland degradation*

Wetland degradation contributes to sea level primarily through (i) direct water drainage or removal from standing inundation, soil moisture, and plant storage, and (ii) water release from vegetation decay and peat combustion. Wada et al. (2016) consider a recent wetland loss rate of 0.565% $yr^{-1}$ since 1990 (Davidson, 2014) and a present global wetland area of 371 mha averaged from three databases: Matthews natural wetlands (Matthews and Fung, 1987), ISLSCP (Darras, 1999), and DISCover (Belward et al., 1999; Loveland and Belward, 1997). They assume a uniform 1-meter depth of water in wetlands (Milly et al., 2010), to estimate a contribution of recent global wetland drainage to sea level of 0.067 mm/yr. Wada et al. (2016) apply a wetland area and loss rate as used for assessing wetland water drainage, to determine the annual reduction of wetland carbon stock since 1990, if completely emitted, releases water equivalent to 0.003–0.007 mm $yr^{-1}$ SLE. Integrating the impacts of wetland drainage, oxidation and peat combustion, Wada et al. (2016) suggest that the recent global wetland degradation results in an upper bound of 0.074 mm/yr SLE.

*Lake storage changes*

Lakes store the greatest mass of liquid water on the terrestrial surface (Oki and Kanae, 2006), yet, because of their "dynamic" nature (Sheng et al., 2016; Wang et al., 2012), their overall contribution to sea level remains uncertain. In the past century, perhaps the greatest contributor in global lake storage was the Caspian Sea (Milly et al., 2010), where the water level exhibits substantial oscillations attributed to meteorological, geological, and anthropogenic factors (Ozyavas et al., 2010, Chen et al., 2017). Assuming the lake level variation kept pace with groundwater changes (Sahagian et al., 1994), the overall contribution of the Caspian Sea, including both surface and groundwater storage variations through 2014, has been about 0.03 mm $yr^{-1}$ SLE since 1900, 0.075 (±0.002) mm $yr^{-1}$ since 1995, or 0.109 (±0.004) mm $yr^{-1}$ since 2002. Additionally, between 1960 and 1990, the water storage in the Aral Sea Basin declined at a striking rate of 64 $km^3$ $yr^{-1}$, equivalent to 0.18 mm $yr^{-1}$ SLE (Sahagian, 2000; Sahagian et al., 1994; Vörösmarty and Sahagian, 2000) due mostly to upstream water diversion for irrigation (Perera, 1993), which was modeled by Pokhrel et al. (2012) to be ~500 $km^3$ during 1951–2000, equivalent to 0.03 mm $yr^{-1}$ SLE. Dramatic decline in the Aral Sea continued in the recent decade, with an annual rate of 6.043 (±0.082) $km^3$ $yr^{-1}$ measured from 2002 to 2014 (Schwatke et al., 2015). Assuming that groundwater drainage has kept pace with lake level reduction (Sahagian et al., 1994), the Aral Sea has contributed 0.0358 (±0.0003) mm $yr^{-1}$ to the recent sea level rise.








*Water cycle variability*
Natural changes in the interannual to decadal cycling of water can have a large effect on the
apparent rate of sea level change over decadal and shorter time periods (Milly et al., 2003;
Lettenmaier and Milly, 2009; Llovell et al., 2010). For instance, ENSO-driven modulations of
the global water cycle can be important in decadal-scale sea level budgets and can mask
underlying secular trends in sea level (Cazenave et al., 2014, Nerem et al., 2018).
Sea level variability due to climate-driven hydrology represents a super-imposed variability
on the secular rates of SLR and not a long-term offset to the GMSL. While this term can be
large and is important in the interpretation of the sea level record, it is arguably the most
difficult term in the land water budget to quantify.














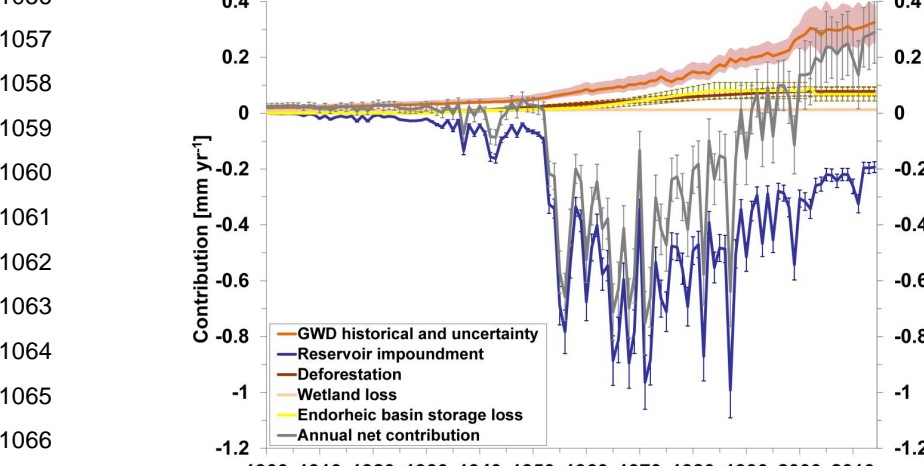

*Figure 11*: *Time series of the estimated annual contribution of terrestrial water storage*
*change to global sea-level over the period 1900-2014 (rates in mm yr$^{-1}$ SLE) (modified from*
*Wada et al., 2016).*

**2.7.2. Net terrestrial water storage**

*GRACE-based estimates*



Measurements of non-ice-sheet continental land mass from GRACE satellite gravity have
been presented in several recent studies (Jensen et al., 2013, Rietbroek et al., 2016; Reager et
al., 2016, Scanlon et al., 2018), and can be used to constrain a global land mass budget. Note
that these 'top-down' estimates contain both climate-driven and direct anthropogenic driven
effects, which makes them most useful in assessing the total impact of land water storage
changes and closing the budget of all contributing terms. GRACE observations, when
averaged over the whole land domain following Reager et al. (2016), indicate a total TWS
change (including glaciers) over the 2002-2014 study period of approximately +0.32 ± 0.13
mm year$^{-1}$ sea level equivalent (i.e., ocean gaining mass). Global mountain glaciers have been
estimated to lose mass at a rate of 0.65 +/- 0.09 mm yr$^{-1}$ (e.g. Gardner et al., 2013; Reager et
al., 2016) during that period, such that a mass balance indicates that global glacier-free land
gained water at a rate of -0.33 ± 0.16 mm yr$^{-1}$ SLE (i.e., ocean losing mass; Figure 12). A
roughly similar estimate was found from GRACE using glacier free river basins globally (-
0.21 ± 0.09 mm/yr) (Scanlon et al., 2018). Thus, the GRACE-based net TWS estimates
suggest a negative sea level contribution from land over the GRACE period (Table 8).
However, mass change estimate from GRACE incorporates uncertainty from all potential
error sources that arise in processing and post-processing of the data, including from the GIA
model, and from the geocenter and mean pole corrections.

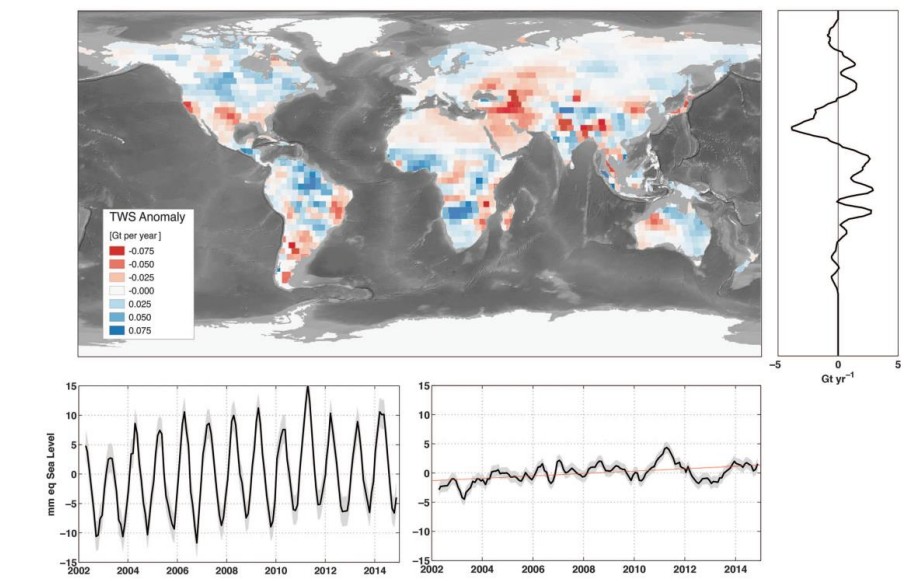

*Figure 12:  An example of trends in land water storage from GRACE observations, April*
*2002 to November 2014.  Glaciers and ice sheets are excluded.  Shown are the global map*



*(gigatons per year), zonal trends, and full time series of land water storage (in mm yr⁻¹ SLE).*
*Following methods details in Reager et al., (2016), GRACE shows a total gain in land water*
*storage during the 2002-2014 period, corresponding to a sea level trend of -0.33 ± 0.16 mm*
*yr⁻¹ SLE (modified from Reager et al., 2016). These trends include all human-driven and*
*climate-driven processes in Table 1, and can be used to close the land water budget over the*
*study period.*

*Estimates based on global hydrological models*
Global land water storage can also be estimated from global hydrological models (GHMs)
and global land surface models (LSMs). These compute water, or water and energy
balances, at the Earth surface, yielding time variations of water storage in response to
prescribed atmospheric data (temperature, humidity and wind) and the incident water and
energy fluxes from the atmosphere (precipitation and radiation). Meteorological forcing is
usually based on atmospheric model reanalyses. Model uncertainties result from several
factors. Recent work has underlined the large differences among different state-of-the-art
precipitation datasets (Beck et al., 2017) with large impacts on model results at seasonal
(Schellekens et al., 2017) and longer time scales (Felfelani et al., 2017). Another source of
uncertainty is the treatment of subsurface storage in soils and aquifers, as well as dynamic
changes in storage capacity due to representation of frozen soils and permafrost, the
complex effects of dynamic vegetation, atmospheric vapor pressure deficit estimation and
an insufficiently deep soil column.  A recent study comparing water storage trends from 5
LSMs and 2 GHMs to GRACE storage trends found that models estimated the opposite
trend in net land water storage to GRACE for the 2002 – 2014 period, attributed to soil
depth limitations in models (Scanlon et al., 2018). These combined error sources are
responsible for a range of storage trends across models of approximately 0.5 ± 0.2 mm/yr
SLE. Figure 13 shows model discrepancies at global and regional scales. In terms of global
land average, model differences can cause up to ~0.4 mm/yr SLE uncertainty.

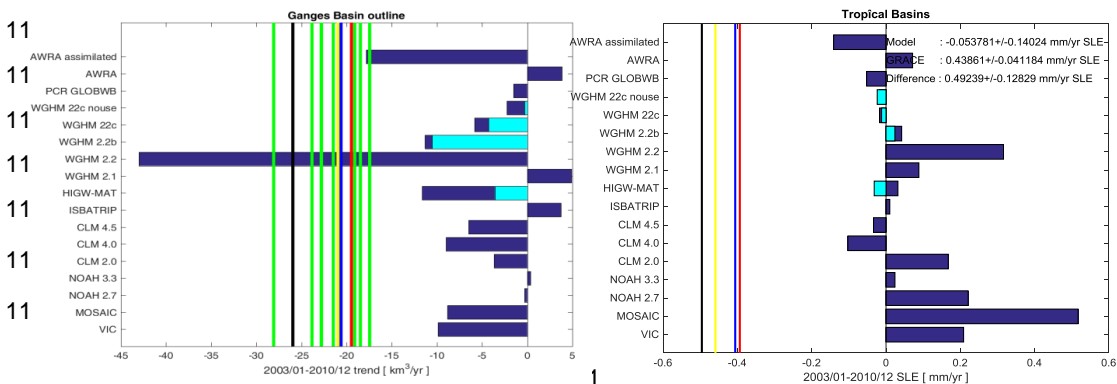




*Figure 13: Regional contribution to sea level rise (2003 – 2010) for the Ganges basin (left*
*pznel) and tropical basins (right panel), as estimated from 17 different models (vertical axes)*
*and 4 GRACE products (JPL, CSR and GSFC mascons – black, red, blue – and GRGS*
*products - yellow) over the GRACE time period. Tropical regions are the main discrepancies*
*between model and GRACE estimates, possibly due to inconsistent model soil storage*
*capacity. The variability among GRACE products (up to 25%) is much smaller than*
*variability among models (up to 300%).*


| **Estimate Terrestrial Water Storage contribution to sea level** | | **2002-2014/15 mm/yr SLE** *(Positive values mean sea level rise)* |
|---|---|---|
| **Human contributions by component** | | |
| Ground water depletion | *Wada et al. (2016)* | 0.30 (± 0.1) |
| Reservoir impoundment | *Wada et al. (2017)* | -0.24 (± 0.02) |
| Deforestation (after 2010) | *Wada et al. (2017)* | 0.035 |
| Wetland loss (after 1990) | *Wada et al. (2017)* | 0.074 |
| Endorheic basin storage loss | | |
|    Caspian Sea | *Wada et al. (2017)* | 0.109 (± 0.004) |
|    Aral Sea | *Wada et al. (2017)* | 0.036 (± 0.0003) |
| | | |
| Aggregated human intervention (sum of above) | *Scanlon et al. (2018)* | 0.15 to 0.24 |
| **Hydrological model-based estimates** | | |
| WGHM model *(natural viariability plus human intervention; P. Döll, personal communication)* | | 0.15 +/- 0.14 |
| ISBA-TRIP model *(natural variability only ; Decharme et al., 2016) + human intervention from Wada et al. (2016) (from Dieng et al., 2017)* | | 0.23 +/- 0.10 |
| **GRACE-based estimates of total land water storage (not including glaciers)** *(Reager et al., 2016; Rietbroek et al., 2016; Scanlon et al., 2018)* | | -0.20 to -0.33 (± 0.09 - 0.16) |


*Table 8. Estimates of TWS components due to human intervention and net TWS based on*
*hydrological models and GRACE*


**2.7.3 Synthesis**
Based on the different approaches to estimate the net land water storage contribution, we
estimate that corresponding sea level rate ranges from -0.33 to 0.23 mm/yr during the period



of 2002-2014/15 due to water storage changes (Table 8). According to GRACE, the net TWS

change (i.e. not including glaciers) over the period 2002-2014 shows a negative contribution

to sea level of -0.33 mm/yr and -0.21 mm/yr, Reager et al. (2016) and Scanlon et al. (2018)

respectively. Such a negative signal is not currently reproduced by hydrological models which

estimate slightly positive trends over the same period (see Table 8). It is to be noted however

that looking at trends only over periods of the order of a decade may not be appropriate due

the strong interannual variability of TWS at basin and global scales. For example, Figure 5

from Scanlon et al. (2018) (see also Figure S9 from their Supplementary Material), that

compares GRACE TWS and model estimates over large river basins over 2002-2014, clearly

show that the discrepancies between GRACE and models occur at the end of the record for

the majority of basins. This is particularly striking for the Amazon basin (the largest

contributor to TWS), for which GRACE and models agree reasonably well until 2011, and

then depart significantly, with GRACE TWS showing strongly positive trend since then,

unlike models. Such a divergence at the end of the record is also noticed for several other

large basins (see Scanlon et al., Figure S9 SM). No clear explanation can be provided yet,

even though one may questions the quality of the meteorological forcing used by hydrological

models for the recent years. But this calls for some caution when comparing GRACE and

models on the basis of trends only. Much more work is needed to understand differences

among models, and between models and GRACE. Of all components entering in the sea level

budget, the TWS contribution currently appears as the most uncertain one.

## 2.8 Glacial Isostatic Adjustment

The Earth's dynamic response to the waxing and waning of the late-Pleistocene ice sheets is

still causing isostatic disequilibrium in various regions of the world. The accompanying slow

process of GIA is responsible for regional and global fluctuations in relative and absolute sea

level, 3D crustal deformations and changes of the Earth's gravity field (for a review, see

Spada, 2017). To isolate the contribution of current climate change, geodetic observations

must be corrected for the effects of GIA (King *et al.* 2010). These are obtained by solving the

"Sea Level Equation'' (Farrell and Clark 1976, Mitrovica and Milne 2003). The sea level can

be expressed as $S=N-U$, where $S$ is the rate of change of sea-level relative to the solid Earth, $N$

is the geocentric rate of sea-level change, and $U$ is the vertical rate of displacement of the

solid Earth. The sea level equation accounts for solid Earth deformational, gravitational and

rotational effects on sea level, which are sensitive to the Earth's mechanical properties and to





the melting chronology of continental ice. Forward GIA modelling, based on the solution of
the sea level equation, provides predictions of unique spatial patterns (or *fingerprints,* see Plag
and Juettner, 2001) of relative and geocentric sea-level change (*e.g.*, Milne *et al.* 2009, Kopp
*et al.* 2015).  During the last decades, the two fundamental components of GIA modelling
have been progressively constrained from the observed history of relative sea level during the
Holocene (see *e.g.*, Lambeck and Chappell 2001, Peltier 2004). In the context of climate
change, the importance of GIA has been recognised in the mid 1980s, when the awareness of
global sea-level rise stimulated the evaluation of the isostatic contribution to tide gauge
observations (see Table 1 in Spada and Galassi 2012). Subsequently, GIA models have been
applied to the study of the pattern of sea level change from satellite altimetry (Tamisiea
2011), and since 2002 to the study of the gravity field variations from GRACE. Our primary
goal here is to analyse GIA model outputs that have been used to infer global mean sea level
change and ice sheet volume change from geodetic datasets during the altimetry era. These
outputs are the sea-level variations detected by satellite altimetry across oceanic regions ($n$),
the ocean mass change ($w$) and the ice sheets mass balance from GRACE. We also discuss the
GIA correction that needs to be applied to GRACE-based land water storage changes. The
GIA correction applied to tide gauge-based sea level observations at the coastlines is not
discussed here. Since GIA evolves on time scales of millennia (*e.g.*, Turcotte and Schubert,
2014), the rate of change of all the isostatic signals in can be considered constant on the time
scale of interest.

### 2.8.1 GIA correction to altimetry-based sea level

Unlike tide gauges, altimeters directly sample the sea surface in a geocentric reference frame.
Nevertheless, GIA contributes significantly to the rates of absolute sea-level change observed
over the "altimetry era'', which require a correction $N_{gia}$ that is obtained by solving the SLE
(*e.g.*, Spada 2017). As discussed in detail by Tamisiea (2011), $N_{gia}$ is sensitive to the assumed
rheological profile of the Earth and to the history of continental ice sheets. The variance of
$N_{gia}$ over the surface of the oceans is much reduced, being primarily determined by the change
of the Earth's gravity potential, apart from a spatially uniform shift. As discussed by Spada
and Galassi (2016), the GIA contribution $N_{gia}$ is strongly affected by variations in the
centrifugal potential associated with Earth rotation, whose fingerprint is dominated by a
spherical harmonic contribution of degree $l=2$ and order $m=\pm 1$. Since $N_{gia}$ has a smooth
spatial pattern. The global the GIA correction to altimetry data can be obtained by simply
subtracting its average $n = <N_{gia}>$ over the ocean sampled by the altimetry missions. The



computation of the GIA contribution $N_{gia}$ has been the subject of various investigations, based
on different GIA models. The estimate by Peltier (2001) of $n$ = -0.30 mm/yr, based on GIA
model ICE-4G (VM2), has been adopted in the majority of studies estimating GMSL rise
from altimetry. Since $n$ appears to be small compared to the global mean sea-level rise from
altimetry (~3 mm/yr) and to its uncertainty (~0.3 mm/yr, see sub section 2.2), a more precise
evaluation has not been of concern until recently. In Table 9, we summarize the values of $n$
according to works in the literature where various GIA model models and averaging methods
have been employed. Based on values in Table 9 for which a standard deviation is available,
the weighted average of $n$, assumed to represent the best estimate, is $n$ = (-0.29 ± 0.02) mm/yr
where the uncertainty corresponds to $2\sigma$.

***2.8.2 GIA correction to GRACE-based ocean mass***
GRACE observations of present-day gravity variations are sensitive to GIA, due to the sheer
amount of rock material that is transported by GIA throughout the mantle and the resulting
changes in surface topography, especially over the formerly glaciated areas. The continuous
change in the gravity field results in a nearly linear signal in GRACE observations. Since the
gravity field is determined by global mass redistribution, GIA models used to correct GRACE
data need to be global as well, especially when the region of interest is represented by all
ocean areas. To date, the only global ice reconstruction publicly available is provided by the
University of Toronto. Their latest product, named ICE-6G, has been published and
distributed in 2015 (Peltier *et al.* 2015); note that the ice history has been simultaneously
constrained with a specific Earth model, named VM5a. During the early period of the
GRACE mission, the available Toronto model was ICE-5G (VM2) (Peltier, 2004). However,
different groups have independently computed GIA model solutions based on the Toronto ice
history reconstruction, by using different implementations of GIA codes and somehow
different Earth models. The most widely used model is the one by Paulson *et al.* (2007), later
updated by A *et al.* (2013). Both use a deglaciation history based on ICE-5G, but differ for the
viscosity profile of the mantle: A *et al.* use a 3D compressible Earth with VM2 viscosity
profile and a PREM-based elastic structure used by Peltier (2004), whereas Paulson *et al.*
(2007) use an incompressible Earth with self-gravitation, and a Maxwell 1-D multi-layer
mantle. Over most of the oceans, the GIA signature is much smaller than over the continents.
However, once integrated over the global ocean, the signal $w$ due to GIA is about -1 mm/yr of
equivalent sea level change (Chambers *et al.* 2010), which is of the same order of magnitude
as the total ocean mass change induced by increased ice melt (Leuliette and Willis 2011). The



main uncertainty in the GIA contribution to ocean mass change estimates, apart from the
general uncertainty in ice history and earth mechanical properties, originates from the
importance of changes in the orientation of the Earth's rotation axis (Chambers *et al.* 2010,
Tamisiea 2011). Different choices in implementing the so-called "rotational feedback" lead to
significant changes in the resulting GIA contribution to GRACE estimates. The issue of
properly accounting from rotational effects has not been settled yet (Mitrovica *et al.* 2005,
Peltier and Luthcke 2009, Mitrovica and Wahr 2011, Martinec and Hagedoorn 2014). Table
1(c) summarises the values of the mass-rate GIA contribution *w* according to the literature,
where various models and averaging methods are employed. The weighed average of the
values in Table 9 for which an assessment of the standard deviation is available, is $w$ = (-1.44
± 0.36) mm/yr (the uncertainty is $2\sigma$), which we assume to represent the preferred estimate.

***2.8.3 GIA correction to GRACE-based terrestrial water storage***
As discussed in the previous section, the GIA correction to apply to GRACE over land is
significant, especially in regions formely covered by the ice caps (Canada and Scandinavia).
Over Canada, GIA models significantly differ. Figure 14 that shows difference between two
models of GIA correction to GRACE over land, the A et al. (2013) and Peltier et al. (2009)
models.


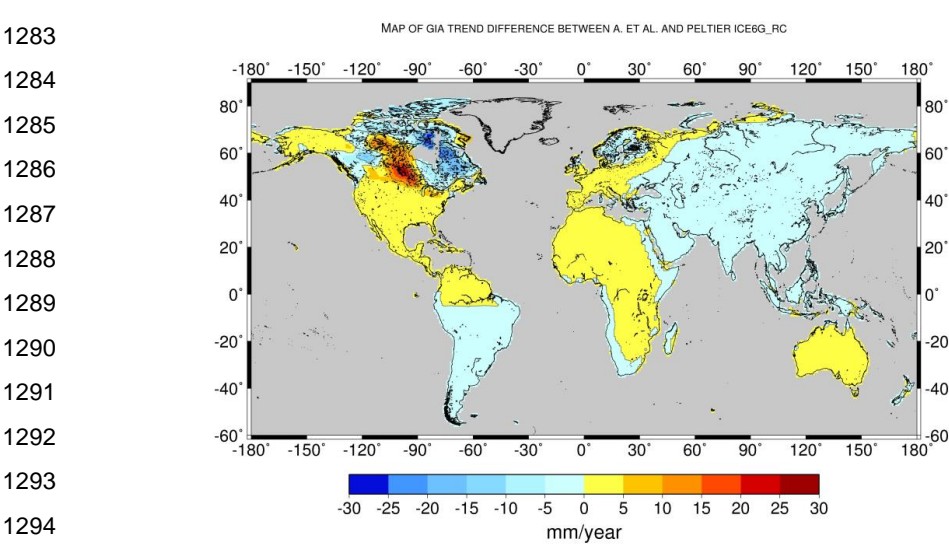

*Figure 14: Difference map between two models of GIA correction to GRACE over land: A et*
*al. (2013) minus Peltier et al. (2015) models.Unit in mm/yr SLE.*




When averaged over the whole land surface as done in some studies to estimate the combined
effect of land water storage and glacier melting from GRACE (e.g., Reager et al., 2015; see
section 2.7), the GIA correction ranges from ~ 0.5 to 0.7 mm/yr (in mm/yr SLE). Values for
different GIA models are given in Table 9.

### 2.8.4  GIA correction to GRACE-based ice sheet mass balance

The GRACE gravity field observations allow the determination of mass balances of ice sheets
and large glacier systems with anaccuracy similar or superior to the input-output-method or
satellite laser and radar altimetry (Shepherd *et al.* 2012). However, GRACE ice-mass balances
rely on successfully separating and removing the apparent mass change related to GIA. While
the GIA correction is small compared to the mass balance for Greenland ice sheet (ca. <
10%), its magnitude and uncertainty in Antarctica is of the order of the ice-mass balance itself
(*e.g.* Martín-Español *et al.* 2016). Particularly for todays glaciated areas, GIA remains poorly
resolved due to the sparsity of data constraints, leading to large uncertainties in the climate
history, the geometry and retreat chronology of the ice sheet, as well as the Earth structure.
The consequences are ambiguous GIA predictions, despite fitting the same observational data.
There are two principal approaches towards resolving GIA underneath the ice sheets.
Empirical estimates can be derived making use of the different sensitivities of satellite
observations to ice-mass changes and GIA (*e.g.* Riva *et al.* 2009, Wu *et al.* 2010).
Alternatively, GIA can be modelled numerically by forcing an Earth model with a fixed ice
retreat scenario (*e.g.* Peltier 2009, Whitehouse *et al.* 2012) or with output from a
thermodynamic ice sheet model (Gomez *et al.* 2013, Konrad *et al.* 2015). Values of GIA-
induced apparent mass change for Greenland and Antarctica as listed in the literature should
be applied with caution (Table 9) when applying tham to GRACE mass balances. Each of
these estimates may rely on a different GRACE post-processing strategy and may differ in the
approach used for solving the gravimetric inverse problem (*mascon* analysis, forward-
modelling, averaging kernels). Of particular concern is the modelling and filtering of the pole
tide correction caused by the rotational variations related to GIA, affecting coefficients of
harmonic degree $l=2$ and order $m=\pm1$. As mentioned above, agreement on the modelling of
the rotational feedback has not been reached within the GIA community. Furthermore, the
pole tide correction applied during the determination gravity-field solutions differs between
the GRACE processing centres and may not be consistent with the GIA correction listed. This
inconsistency may introduce a significant bias in the ice-mass balance estimates (*e.g.* Sasgen



*et al.* 2013, Supplement). Wahr *et al.* 2015 presented recommendations on how to treat the
pole tides in GRACE analysis; however, a systematic inter-comparison of the GIA predictions
in terms of their low-degree coefficients and their consistency with the GRACE processing
standards is required.

Table 9.  *Estimated contributions of GIA to the rate of absolute sea level change observed by*
*altimetry (a), and to the rate of mass change observed by GRACE over the global oceans (b)*
*the Greenland and Antarctic ice sheets (c), and the rate of mass change observed by GRACE*
*over land (d) during the altimetry era. The GIA corrections are expressed in mm/yr SLE*
*except over Greenland and Antarctica where values are given in Gt/yr (ice mass equivalent).*
*Most of the GIA contributions are expressed as a value ± one standard deviation; a few*
*others are given in terms of a plausible range, for some the uncertainties are not specified.*

(a) GIA correction to absolute sea level measured by altimetry

| Reference | GIA (mm/yr) | Notes |
|---|---|---|
| Peltier (2009) (Table 3) | -0.30 ± 0.02 -0.29 ± 0.03 -0.28 ± 0.02 | Average of 3 groups of 4 values obtained by variants of the analysis procedure, using ICE-5G(VM2), over a global ocean, in the range of latitudes 66°S to 66°N and 60°S to 60°N, respectively. |
| Tamisiea (2011) (Figure 2) | -0.15 to -0.45 -0.20 to -0.50 | Simple average over the oceans for a range of estimates obtained varying the Earth model parameters, over a global ocean and between latitudes 66°S and 66°N. |
| Huang (2013) (Table 3.6) | -0.26 ± 0.07 -0.27 ± 0.08 | Average from an ensemble of 14 GIA models over a global ocean and between latitude from 66°S to 66°N. |
| Spada (2017) (Table 1) | -0.32 ± 0.08 | Based on four runs of the Sea Level Equation solver SELEN (Spada and Stocchi, 2007) using model ICE-5G(VM2), with different assumptions in solving the SLE. |

(b) GIA contribution to GRACE mass-rate of change over the oceans



| Reference | GIA (mm/yr) | Notes |
|---|---|---|
| Peltier (2009) (Table 3) | -1.60 ± 0.30 | Average of values from 12 corrections for variants of the analysis procedure, using ICE-5G (VM2). |
| Chambers (2010) (Table 1) | -1.45 ± 0.35 | Average over the oceans for a range of estimates produced by varying the Earth models. |
| Tamisiea (2011) (Figures 3 and 4) | -0.5 to -1.9 -0.9 to −1.5 | Ocean average of a range of estimates varying the Earth model, and based on a restricted set, respectively. |
| Huang (2013) (Table 3.7) | -1.31 ± 0.40 -1.26 ± 0.43 | Average from an ensemble of 14 GIA models over a global ocean and between latitude from $66^{o}S$ to $66^{o}N$, respectively. |

(c) GIA contribution to GRACE mass-rate of the ice sheets

| Reference | **Greenland** GIA (Gt/yr) | Notes |
|---|---|---|
| Simpson et al. (2009)[r] | -3 ± 12[m] | Thermodynamic sheet / solid Earth model, 1D (uncoupled); constrained by geomorphology; inversion results in Sutterley *et al.* (2014). |
| Peltier (2009) (ICE-5G)[g] | -4[f] | Ice load reconstruction / solid Earth model, 1D (ICE-5G / similar to VM2); Greenland component of ICE-5G (13 Gt/yr) + Laurentide component of ICE-5G (-17 Gt/yr); inversion results in Khan *et al.* 2016, Discussion. |
| Khan et al. (2016) (GGG-1D)[r] | 15 ± 10[f] | Ice load reconstruction / solid Earth model, 1D (uncoupled); constrained with geomorphology & GPS; Greenland component (+32 Gt/yr) + Laurentide component of ICE-5G (-17 Gt/yr); inversion results in Khan *et al.* (2016), Discussion. |





| Reference | GIA (Gt/yr) | Notes |
|---|---|---|
| Fleming *et al.* (2004)[r] (Green1) | 3[f] | Ice load reconstruction / solid Earth model, 1D (uncoupled); constrained with geomorphology; Greenland component (+ 20 Gt/yr) + Laurentide component of ICE-5G (-17 Gt/yr); inversion in Sasgen *et al.* (2012, supplement). |
| Wu *et al.* (2010)[g] | -69 ± 19[m] | Joint inversion estimate based on GPS, satellite laser ranging, and very long baseline interferometry, and bottom pressure from ocean model output; inversion results in Sutterley *et al.* (2014). |

| **Antarctica** | | |
|---|---|---|
| Reference | GIA (Gt/yr) | Notes |
| Whitehouse *et al.* (2012) (W12a)[r] | 60[n] | Thermodynamic sheet / solid Earth model, 1D (uncoupled); constrained by geomorphology; inversion results in Shepherd *et al.* 2012, supplement (Fig. S8). |
| Ivins *et al.* (2013) (IJ05_R2)[r] | 40-65[n] | Ice load reconstruction / solid Earth model, 1D; constrained by geomorphology and GPS uplift rates; Ivins et al. 2013; inversion results in Shepherd *et al.* 2012, Supplement (Fig. S8). |
| Peltier (2009) (ICE-5G)[g] | 140-180[n] | Ice load reconstruction / solid Earth model ICE-5G(VM2); constrained by geomorphology; inversion results in Shepherd *et al.* 2012, supplement (Fig. S8). |
| Argus *et al.* (2014) (ICE-6G)[g] | 107[n] | Ice load reconstruction / solid Earth model ICE-6G(VM5a); constrained by geomorphology and GPS; theory recently corrected by Purcell *et al.* 2016; inversion results in Argus *et al.* (2014), conclusion 7.8. |
| Sasgen *et al.* (2017) (REGINA)[r] | 55 ± 22[f] | Joint inversion estimate based on GRACE, altimetry, GPS and viscoelastic response functions; lateral heterogeneous Earth model parameters; inversion results in Sasgen *et al.* (2017), Table 1. |



| Gunter *et al.* (2014) (G14)[r] | ca. 64 ± 40 [a] (multimodel uncert.) | Joint inversion estimate based on GRACE, altimetry, GPS and regional climate model output; conversion of uplift to mass using average rock density; inversion results in, Gunter *et al.* (2014) Table 1. |
| Martin-Español *et al.* (2016) (RATES)[r] | 55 ± 8 45 ± 7* | Joint inversion estimate based on GRACE, altimetry, GPS and regional climate model output; inversion results in Sasgen *et al.* (2017), * is improved for GIA of smaller spatial scales; inversion results in Martin-Español *et al.* (2016), Fig. 6. |

[r] regional model; [g] global model; [m] mascon inversion; [f] forward modelling inversion; [a]
averaging kernel inversion; [n] inversion method not specified.

(d) GIA correction to GRACE-based terrestrial water storage estimates


| GIA MODEL | Mean GIA correction over land excluding Greenland, Antacrtica, Iceland, Svalsbard, Hudson Bay and Black Sea (mm/yr sea level equivalent/SLE) |
| --- | --- |
| A. et al., 2013 | 0.63 |
| Peltier ICE5G | 0.68 |
| Peltier ICE6G_rc | 0.71 |
| ANU_ICE6G | 0.53 |

The GRACE-based ocean mass, Antarctica mass and terrestrial water storage changes are
much model dependent. As these GIA corrections cannot be assessed from independent
information, they represent a large source of uncertainties to the sea level budget components
based on GRACE.

**2.9  Ocean mass change from GRACE**



1360   Since 2002, GRACE satellite gravimetry has provided a revolutionary means for

1361 measuring global mass change and redistribution at monthly intervals with unprecedented

1362 accuracy, and offered the opportunity to directly estimate ocean mass change due to water

1363 exchange between the ocean and other components of the Earth (e.g., ice sheets, mountain

1364 glaciers, terrestrial water). GRACE time-variable gravity data have been successfully applied

1365 in a series of studies of ice mass balance of polar ice sheets (e.g., Velicogna and Wahr, 2006;

1366 Luthcke et al., 2006) and mountain glaciers (e.g., Tamisiea et al., 2005; Chen et al., 2007) and

1367 their contributions to global sea level change. GRACE data can also be used to directly study

1368 long-term oceanic mass change or non-steric sea level change (e.g., Willis et al., 2008;

1369 Leuliette et al., 2009; Cazenave et al., 2009), and provide a unique opportunity to study

1370 interannual or long-term terrestrial water storage (TWS) change and its potential impacts on

1371 sea level change (Richey et al., 2015; Reager et al., 2016).

1372   GRACE time-variable gravity data can be used to quantify ocean mass change from three

1373 different main approaches. One is through measuring ice mass balance of polar ice sheets and

1374 mountain glaciers and variations of TWS, and their contributions to the GMSL (e.g.,

1375 Velicogna and Wahr, 2005; Schrama et al., 2014). The second approach is to directly quantify

1376 ocean mass change using ocean basin mask (kernel) (e.g., Chambers et al., 2004; Llovel et al.,

1377 2010; Johnson and Chambers, 2013). In the ocean basin kernel approach, coastal ocean areas

1378 within certain distance (e.g., 300 or 500 km) from the coast are excluded, in order to minimize

1379 contaminations from mass change signal over the land (e.g., glacial mass loss and TWS

1380 change). The third approach solves mass changes on land and over ocean at the same time via

1381 forward modeling (e.g., Chen et al., 2013; Yi et al., 2015). The forward modeling is a global

1382 inversion to reconstruct the "true" mass change magnitudes over land and ocean with

1383 geographical constraint of locations of the mass change signals, and can help effectively

1384 reduce leakage between land and ocean (Chen et al., 2015).

1386   Estimates of ocean mass changes from GRACE are subject to a number of major error

1387 sources, which include leakage errors from the larger signals over ice sheets and land

1388 hydrology due to GRACE's low spatial resolution (of at least a few to several hundred km)

1389 and the need of coastal masking, spatial filtering of GRACE data to reduce spatial noise,

1390 errors and biases in geophysical model corrections (e.g., GIA, atmospheric mass) that need to

1391 be removed from GRACE observations first to isolate oceanic mass change and/or polar ice

1392 sheets and mountain glaciers mass balance, and residual measurement errors in GRACE

1393 gravity measurements, especially those associated with GRACE low-degree gravity changes.



In addition, how to deal with the absent degree-1 terms, i.e., geocenter motion in GRACE gravity fields is expected to affect on GRACE estimated oceanic mass rates and ice mass balances.

| Ocean mass trends (mm/yr) | Time Period | Mass |
|---|---|---|
| Chen et al. (2013) (A13 GIA) | 2005.01-2011.12 | $1.80 \pm 0.47$ |
| Johnson and Chambers (2013) (A13 GIA) | 2003.01-2012.12 | $1.80 \pm 0.15$ |
| Purkey et al. (2014) (A13 GIA) | 2003.01-2013.01 | $1.53 \pm 0.36$ |
| Dieng et al. (2015a) (Paulson07 GIA) | 2005.01-2012.12 | $1.87 \pm 0.11$ |
| Dieng et al. (2015b) (Paulson07 GIA) | 2005.01-2013.12 | $2.04 \pm 0.08$ |
| Yi et al. (2015) (A13 GIA) | 2005.01-2014.07 | $2.03 \pm 0.25$ |
| Rietbroek et al. (2016) | 2002.04-2014.06 | $1.08 \pm 0.30$ |
| Chambers et al. (2017) | $2005.0 - 2015.0$ | $2.11 \pm 0.36$ |

*Table 10. Recently published (since 2013) estimates of GRACE ocean rate with GIA corrections (Mass), Most of the listed studies use either the A13 (A et al., 2013) or Paulson07 (Paulson et al., 2007) GIA model.*

The uncertainty estimates of the listed studies (Table 10) are computed from different methods, with different considerations of error sources into the error budget, and represent different confidence levels.

With a different treatment of the GRACE land-ocean signal leakage effect through global forward modeling, Chen et al. (2013) estimates ocean mass rates using GRACE RL05 time-



variable gravity solutions over the period 2005-2011, and shows that the ocean mass change
contributes to $1.80 \pm 0.47$ mm/yr (over the same period), which is significantly larger than
previous estimates over about the same period. Yi et at. (2015) further confirms that correct
calibration of GRACE data and appropriate treatment of GRACE leakage bias are critical to
improve the accuracy of GRACE estimated ocean mass rates. Table 10 summarizes different
estimates of GRACE ocean mass rates.
As demonstrated in Chen et al. (2013), different treatments of just the degree-2 spherical
harmonics of GRACE gravity solution alone can lead to substantial differences in GRACE
estimated ocean mass rates (ranging from 1.71 to 2.17 mm/yr). Similar estimates from
GRACE gravity solutions from different data processing centers can also be different. In the
meantime, long-term degree-1 spherical harmonics variation, representing long-term
geocenter motion and neglected in some of the previous studies (due to the lack of accurate
observations) are also expected to have non-negligible effect on GRACE derived ocean mass
rates (Chen et al., 2013). Different methods for computing ocean mass change using GRACE
data may also lead to different estimates (Chen et al., 2013; Johnson and Chambers, 2013,
Jensen et al., 2013).
To help better understand the potential and uncertainty of GRACE satellite gravimetry in
quantification of the ocean mass rate, Table 11 provides a comparison of GRACE-estimated
ocean mass rates over the period January 2005 to December 2016 based on different GRACE
data products and different data processing methods, including the CSR, GFZ and JPL
GRACE RL05 spherical harmonic solutions (i.e., the so-called GSM solutions), and CSR,
JPL, and GSFC mascon solutions (the available GSFC mascons only cover the period up to
July 2016). The three GRACE GSM results (CSR, GFZ, and JPL) are updates from Johnson
and Chambers (2013), with $\Delta C_{20}$ terms replaced by satellite laser ranging results (Cheng and
Ries, 2012), geocenter motion from Swenson et al. (2008), GIA model from A et al. (2013),
an averaging kernel with a land mask that extends out 300 km, and no destriping or
smoothing, as described in Johnson and Chambers (2013). An update of GRACE ocean mass
rate from Chen et al. (2013) is also included for comparisons, which is based on the CSR
GSM solutions using forward modeling (a global inversion approach), with similar treatments
of $\Delta C_{20}$, geocenter motion, and GIA effects.
The JPL mascon ocean mass rate is computed from all mascon grids over the ocean, and
the GSFC mascon ocean mass rate is computed from all ocean mascons, with the
Mediterranean, Black and Red Seas excluded. A coastline resolution improvement (CRI)



filter is already applied in the JPL mascons to reduce leakage (Wiese et al., 2016), and in both the GSFC and JPL mascon solutions, the ocean and land are separately defined (Luthcke et al., 2013; Watkins et al., 2015). For the CSR mascon results, an averaging kernel with a land mask that extends out 200 km is applied to reduced leakage (Chen et al., 2017). Similar treatments or corrections of $\Delta C_{20}$, geocenter motion, and GIA effects are also applied in the three mascon solutions. When solving GRACE mascon solutions, the GRACE GAD fields (representing ocean bottom pressure changes, or combined atmospheric and oceanic mass changes) have been added back to the mascon solutions. To correctly quantify ocean mass change using GRACE mascon solutions, the means of the GAD fields over the oceans, which represents mean atmospheric mass changes over the ocean (as ocean mass is conserved in the GAD fields) need to be removed from GRACE mascon solutions. The removal of GAD average over the ocean in GRACE mascon solutions has very minor or negligible effect (of ~ 0.02 mm/yr) on ocean mass rate estimates, but is important for studying GMSL change at seasonal time scales.

Over the 12-year period (2005-2016), the three GRACE GSM solutions show pretty consistent estimates of ocean mass rate, in the range of 2.3 to 2.5 mm/yr. Greater differences are noticed for the mascon solutions. The GSFC mascons show the largest rate of 2.61mm/yr. The CSR and JPL mascon solutions show relatively smaller ocean mass rates of 1.76 and 2.02 mm/yr, respectively, over the studied period. Based on the same CSR GSM solutions, the forward modeling and basin kernel estimates agree reasonably well (2.52 vs. 2.44 mm/yr). In addition to the $\Delta C_{20}$ terms, geocenter motion, and GIA correction, the degree-2 non-zonal spherical harmonics $\Delta C_{21}$ and $\Delta S_{21}$ of the current GRACE RL05 solutions are affected by the definition of the reference mean pole in GRACE pole tide correction (Wahr et al., 2015). This mean pole correction, excluded in all estimates listed in Table 11 (for fair comparison), is estimated to contribute ~ - 0.11 mm/yr to GMSL. How to reduce errors from the different sources play a critical role in estimating ocean mass change from GRACE time-variable gravity data.

| Ocean mass change (in mm/yr equivalent sea level) | Linear trend 2005-2016 |
|---|---|
| GSM CSR Forward Modeling (update from Chen et al., 2013) | 2.52±0.17 |
| GSM CSR (update from Johnson and Chambers, 2013) | 2.44±0.15 |



| | |
|---|---|
| GSM GFZ (update from Johnson and Chambers, 2013) | 2.30±0.15 |
| GSM JPL (update from Johnson and Chambers, 2013) | 2.48±0.16 |
| Mascon CSR (200 km) | 1.76±0.16 |
| Mascon JPL | 2.02±0.16 |
| Mascon GSFC (update from Luthcke et al., 2013) | 2.61±0.16 |
| Ensemble mean | 2.3 ± 0.19 |

*Table 11. Ocean mass rates estimated from GRACE for the period January 2005 – December*
*2016 (the GSFC mascon solutions cover up to July 2016). The uncertainty is based on 2 times*
*the sigma of least squares fitting.*

GRACE satellite gravimetry has brought a completely new era for studying global ocean
mass change. Owing to the extended record of GRACE gravity measurements (now over 15
years), improved understanding of GRACE gravity data and methods for addressing GRACE
limitations (e.g., leakage and low-degree spherical harmonics), and improved knowledge of
background geophysical signals (e.g., GIA), GRACE-derived ocean mass rates from different
studies in recent years show clearly increased consistency (Table 11). Most of the results
agree well with independent observations from satellite altimeter and Argo floats, although
the uncertainty ranges are still large. The GRACE Follow-On (FO) mission is scheduled to be
launched in March 2018. The GRACE and GRACE-FO together are expected to provide at
least over two (or even three) decades of time-variable gravity measurements. Continuous
improvements of GRACE data quality (in future releases) and background geophysical
models are also expected, which will help improve the accuracy GRACE observed ocean
mass change.
For the sea level budget assessment over the GRACE period, we will use the ensemble mean.

## 3. Sea Level Budget results
In section 2, we have presented the different terms of the sea level budget equation, mostly
based on published estimates (and in some cases, from their updates). We now use them to
examine the closure of the sea level budget. For all terms, we only consider ensemble mean
values.

**3.1 Entire altimetry era (1993-Present)**

*3.1.1 Trend estimates over 1993-Present*



Because it is now clear that the GMSL and some components are accelerating (e.g., Nerem et al., 2018), we propose to characterize the long term variations of the time series by both a trend and an acceleration. We start looking at trends. Table 12 gathers the trends estimated in section 2. The end year is not always the same for all components (see section 2). Thus the word 'present' means either 2015 or 2016 depending on the component. As no trend estimate is available for the entire altimetry era for the terrestrial water storage contribution, we do not consider this component. The residual trend (GMSL minus sum of components trend) may then provide some constraint on the TWS contribution.

| Component | Trends (mm/yr) 1993-Present |
|---|---|
| 1.GMSL (TOPEX-A drift corrected) | 3.07 +/- 0.37 |
| 2. Thermosteric sea level (full depth) | 1.3 +/- 0.4 |
| 3. Glaciers | 0.65 +/- 0.15 |
| 4. Greenland | 0.48 +/- 0.10 |
| 5. Antarctica | 0.25 +/- 0.10 |
| 6. TWS | / |
| 7. Sum of components (without TWS→ 2.+3.+4.+5.) | 2.7 +/- 0.23 |
| 8. GMSL minus sum of components (without TWS) | 0.37 +/- 0.3 |

*Table 12: Trend estimates for individual components of the sea level budget, sum of components and GMSL minus sum of components over 1993-present. Uncertainties of the sum of components and residuals represent rooted mean squares of components errors, assuming that errors are independent.*

Resuts presented in Table 12 are discussed in section 4.

### 3.1.2 Acceleration

The GMSL acceleration estimated in section 2.2 using Ablain et al. (2017b)'s TOPEX-A drift correction amounts to 0.10 mm/yr$^2$ for the 1993-2017 time span. This value is in good agreement with Nerem et al. (2018) estimate (of 0.084 +/- 0.025 mm/yr$^2$) over nearly the same period, after removal of the interannual variability of the GMSL. In Nerem et al.



(2018), acceleration of individual components are also estimated as well as acceleration of the
sum of components. The latter agrees well with the GMSL acceleration. Here we do not
estimate the acceleration of the component ensemble means because time series are not
always available. We leave this for a future assessment.
**3.2 GRACE and Argo period (2005- Present)**
*3.2.1 Sea level budget using GRACE-based ocean mass*
If we consider the ensemble mean trends for the GMSL, thermosteric and ocean mass
compents given in sections 2.2, 2.3 and 2.9 over 2005-present, we find agreement (within
error bars) between the observed GMSL (3.5 +/- 0.2 mm/yr) and the sum of Argo-based
thermosteric plus GRACE-based ocean mass (3.6 +/- 0.4 mm/yr) (see Table 12). The residual
(GMSL minus sum of components) trend amounts to -0.1 mm/yr. Thus in terms of trends, the
sea level budget appears closed over this time span within quoted uncertainties.
*3.2.2 Trend estimates over 2005-Present from estimates of individual contributions*
Table 13 gathers trends of individual components of the sea level budget over 2005-present,
as well sum of components and residuals (GMSL minus sum of components) trend. As for the
longer period, ensemble mean values are considered for each component.

| Component | Trend (mm/yr) 2005-Present |
|---|---|
| 1. GMSL | 3.5 +/- 0.2 |
| 2. Thermosteric sea level (full depth) | 1.3 +/- 0.4 |
| 3. Glaciers | 0.74 +/- 0.1 |
| 4. Greenland | 0.76 +/- 0.1 |
| 5. Antarctica | 0.42 +/- 0.1 |
| 6. TWS from GRACE (mean of Reager et al. and Scanlon et al.) | -0.27 +/- 0.15 |
| 7. Sum of components (2.+3.+4.+5.+6.) | 2.95 +/- 0.21 |
| 8. Sum of components (thermosteric full depth + GRACE-based ocean mass) | 3.6 +/- 0.4 |



| | |
|---|---|
| 9. GMSL minus sum of components (including GRACE-based TWS→2.+3.+4.+5.+6.) | 0.55 +/- 0.3 |
| 10.GMSL minus sum of components (without GRACE-based TWS→ 2.+3.+4.+5.) | 0.28 +/- 0.2 |
| 11. GMSL minus sum of components (thermosteric full depth + GRACE-based ocean mass) | -0.1 +/- 0.3 |


*Table 13: Trend estimates for individual components of the sea level budget, sum of*
*components and GMSL minus sum of components over 2005-present.*
**3.2.3 Year-to-year budget over 2005-Present using GRACE-based ocean mass**
We now examine the year-to-year sea level and mass budgets. Table 14 provides annual mean
values for the ensemble mean GMSL, GRACE-based ocean mass and Argo-based
thermosteric component. The components are expressed as anomalies and their reference is
arbitrary. So to compare with the GMSL, a constant offset for all years was applied to the
thermosteric and ocean mass annual means. The reference year (where all values are set to
zero) is 2003.

| Year | Ensemble mean GMSL mm | Sum of components mm | GMSL minus sum of components mm |
|---|---|---|---|
| 2005 | 7.00 | 8.78 | -0.78 |
| 2006 | 10.25 | 10.78 | -0.53 |
| 2007 | 10.51 | 11.35 | -0.85 |
| 2008 | 15.33 | 15.07 | 0.25 |
| 2009 | 18.78 | 18.88 | -0.10 |
| 2010 | 20.64 | 20.53 | 0.11 |
| 2011 | 20.91 | 21.38 | -0.48 |



| 2012 | 31.10 | 29.33 | 1.77 |
| 2013 | 33.40 | 33.87 | -0.47 |
| 2014 | 36.65 | 36.22 | 0.43 |
| 2015 | 46.34 | 45.69 | 0.65 |


*Table 14. Annual mean values for the ensemble means GMSL and sum of components*
*(GRACE-based ocean mass and Argo-based thermosteric, full depth). Constant offset applied*
*to the sum of components. The reference year (where all values are set to zero) is 2003.*

Figure 15 shows the sea level budget over 2005-2015 in terms of annual bar chart using
values given in Table 14. It compares for years 2005 to 2016 the annual mean GMSL (blue
bars) and annual mean sum of thermosteric and GRACE-based ocean mass (red bars). Annual
residuals are also shown (green bars). These are either positive or negative depending on
years. The trend of these annual residuals is estimated to 0.135 mm/yr.
In Figure 16 is also shown the annual sea level budget over 2005-2015 but now using the
individual components for the mass terms. As we have no annual estimates for TWS, we
ignore it, so that the total mass includes only glaciers, Greenland and Antarctica. The annual
residuals thus include the TWS component in addition to the missing contributions (e.g., deep
ocean warming). For years 2006 to 2011, the residuals are negative, an indication of a
negative TWS to sea level as suggested by GRACE results (Reager et al., 2016, Scanlon et al.,
2018). But as of 2012, the residuals become positive and on average over 2005-2015, the
residual trend amounts +0.28 mm/yr, a value larger than when using GRACE ocean mass.
Finally, Figure 17 presents the mass budget. It compares annual GRACE-based ocean mass to
the sum of the mass components, without TWS as in Figure 16. The residual trends over
2005-2015 time span is 0.14 mm/yr. It may dominantly represent the TWS contribution. From
one year to another residuals can be either positive or negative, suggesting important
interannual variability in the TWS or even in the deep ocean.








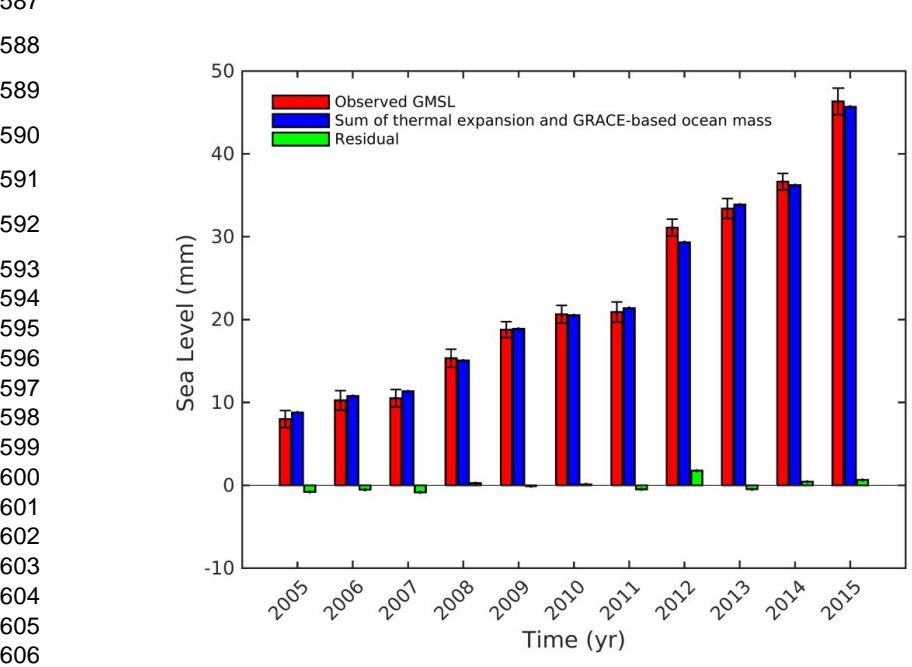

*Figure 15: Annual sea level (blue bars) and sum of thermal expansion (full depth) and GRACE ocean mass component (red bars). Black vertical bars are associated uncertainties. Annual residuals (green bars) are also shown.*

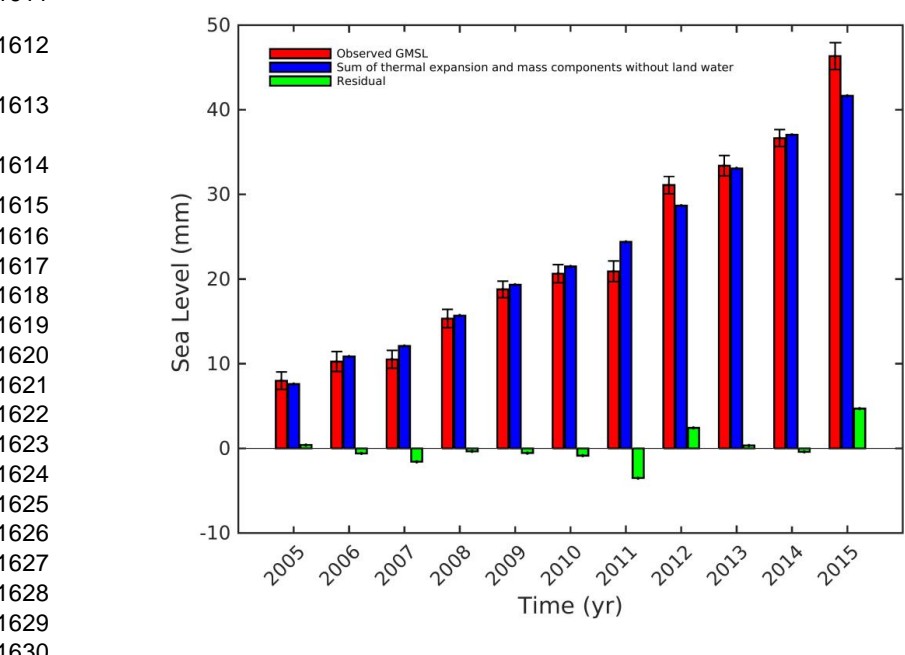



*Figure 16: Annual global mean sea level (blue bars) and sum components without TWS (full*
*depth thermal expansion+ glaciers + Greenland + Antarctica) (red bars). Black vertical bars*
*are associated uncertainties. Annual residuals (green bars) are also shown.*


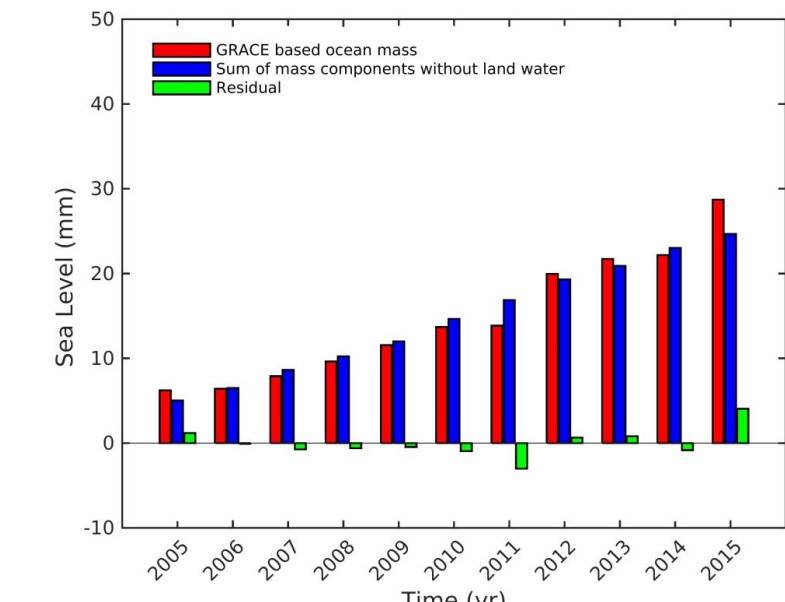

*Figure 17: Annual GRACE-based ocean mass (red bars) and sum components without TWS*
*(full depth thermal expansion+ glaciers + Greenland + Antarctica) (blue bars). Annual*
*residuals (green bars) are also shown.*

## 4. Discussion

The results presented in section 2 for the components of the sea level budget are based on
syntheses of the recently published literature. When needed, the time series have been
updated. In section 3, we considered ensemble means for each component to average out
random errors of individual estimates. We examined the closure/non closure of the sea level
budget using these ensemble mean values, for 2 periods: 1993-present and 2005-present
(Argo and GRACE period). Because of the lack of observation-based TWS estimate for the
1993-present time span, we compared the observed GMSL trend to the sum of components
excluding TWS. We found a positive residual trend of 0.37 +/- 0.3 mm/yr, supposed to



include the TWS contribution, plus other inperfectly known contributions (deep ocean
warming) and data errors.
For the 2005-present time span, we considered both GRACE-based ocean mass and sum of
individual mass components, allowing us to also look at the mass budget. For TWS, as
discussed in section 2.7, GRACE provides a negative trend contribution to sea level over the
last decade (i.e., increase on water storage on land) attributed to internal natural variability
(Reager et al., 2016), unlike hydrological models that lead to a small (possible not
significantly different from 0) positive contribution to sea level over the same period.
Assuming that GRACE observations are perfect, such discrepancies could be attributed to the
inability of models to correctly account for uncertainties in meteorological forcing and
inadequate modeling of soil storage capacity (see discussion in section 2.7). However, when
looking at the sea level budget over the GRACE time span and using the GRACE-based
TWS, we find a rather large positive residual trend (> 0.5 mm/yr) that needs to be explained.
Since GRACE-based ocean mass is supposed to represent all mass terms , one may want to
attribute this residual trend to an additional contribution of the deep ocean to the abyssal
contribution already taken into account here, but possibly underestimated because of
incomplete monitoring by current observing systems. If such a large positive contribution
from the deep ocean (meaning ocean warming) is real (which is unlikely, given the high
implied heat storage), this has to be confirmed by independent approaches e.g., using ocean
reanalyses, and eventually model-based and top-of-the-atmosphere estimates of the Earth
Energy Imbalance.
In addition to mean trends over the period, we also looked at the annual budget for all years
starting in 2005. For most components, annual mean values are provided during the Argo-
GRACE era, except for the terrestrial water storage component. However, the sea level
budget based on GRACE ocean mass (plus ocean thermal expansion; Figure 15) includes the
TWS contribution. As shown in Figure 15, yearly residuals are small, suggesting near closure
of the sea level budget. The residual trend amounts to 0.13 mm/yr. It could be interpreted as
an additional deep ocean contribution not accounted by the SIO estimate (see section 2.3).
However, when looking at Figure 15, we note that yearly residuals are either positive or
negative, an indication of interannual variability that can hardly be explained by a deep ocean
contribution.  The residual trend derived from the difference (GMSL minus sum of
components) (Table 13) amounts -0.1 +/- 0.3 mm/yr, suggesting a sea level budget closed
within 0.3 mm/yr over 2005-present, with no substantial deep ocean contribution.



Figure 17 compares GRACE ocean mass to the sum of mass components (excluding TWS,
for the reasons mentioned above). In principle, this mass budget may provide a constraint on
the TWS contribution. The corresponding residual trend amounts to 0.14 mm/yr over the
GRACE period, a value that agrees in sign with hydrological models estimates but disagrees
with GRACE-based TWS estimates. However, given the remaining data uncertainties, any
robust conclusion can hardly be reached. That being said, more work is needed to clarify the
sign discrepancy between GRACE-based and model-based TWS estimates.

## 1707   5. Concluding Remarks

As mentioned in the introduction, the global mean sea level budget has been the object of
numerous previous studies, including successive IPCC assessments of the published literature.
What is new in the effort presented here, is that it involves the international community
currently studying present-day sea level and its components. Moreover, it relies on a large
variety of datasets derived from different space-based and in situ observing systems. Near
closure of the sea level budget as reported here over the GRACE and Argo era suggests that
no large systematic errors affect these independent observing systems, including the satellite
altimetry system. Study of the sea level budget allows improved understanding of the
different processes causing sea level rise, such as ocean warming and land ice melt.  When
accuracy increases, it will offer an integrative view of the response of the Earth system to
natural & anthropogenic forcing and internal variability, and provide an independent
constraint on the current Earth Energy Imbalance. Validation of climate models against
observations is another important application of this kind of assessment (e.g., Slangen et al.,

2017).

However, important uncertainties still remain, that affect several terms of the budget; for
exemple the GIA correction applied to GRACE data over Antarctica or the net land water
storage contribution to sea level. The latter results from a variety of factors but is dominated
by ground water pumping and natural climate variability. Both terms are still uncertain and
accurately quantifying them remains a challenge.
Several ongoing international projects related to sea level should provide in the near future
improved estimates of the components of the sea level budget. This is the case, for exemple,
of the ice sheet mass balance inter-comparison exercise (IMBIE, 2nd assessment), a
community effort supported by NASA (National Aeronautics and Space Administration) and
ESA, dedicated to reconcile satellite measurements of ice sheet mass balance (Shepherd et al.,



2012). This is also the case for the ongoing ESA Sea level Budget Closure project (Horwath
et al., 2018) that uses a number of space-based Essential Climate Variables (ECVs)
reprocessed during the last 7 years in the context of the ESA Climate Change Initiative
project. The GRACE follow-on mission, scheduled for launch mid-2018, will lengthen the
current mass component time series, with hopefully increased precision and resolution.
Finally, the deep Argo project, still in an experimental phase, will provide important
information on the deep ocean heat content in the coming years. Availability of this new data
set will be open new insight on the total thermosteric component of the sea level budget,
allowing constraining other missing or poorly known contributions, from the evaluation of the
budget.
The sea level budget assessment discussed here essentially relies on trend estimates. But
annual budget estimates have been proposed for the first time over the GRACE-Argo era. It is
planned to provide updates of the global sea level budget every year, as done for more than a
decade for the global carbon budget (Le Queré et al., 2018). In the next assessments, updates
of all components will be considered, accounting for improved evaluation of the raw data,
improved processing and corrections, use of ocean reanalyses, etc. Need for additional
information where gaps exist should also be considered. As a closing remark, study of the sea
level budget in terms of time series, not just trends as done here, will be required.

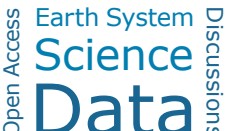

**List of authors and affiliations:**
Anny Cazenave (LEGOS, France), Benoit Meyssignac (LEGOS, France);
Michael Ablain (CLS, France), Jonathan Bamber (U. Bristol, UK), Valentina Barletta (DTU-
SPACE, Denmak), Brian Beckley (SGT Inc./NASA GSFC, USA), Jérôme Benveniste
(ESA/ESRIN, Italy), Etienne Berthier (LEGOS, France), Alejandro Blazquez (LEGOS,
France), Tim Boyer (NOAA, USA), Denise Caceres (Goethe U., Germany), Don Chambers
(U. South Florida, USA), Nicolas Champollion (U. Bremmen, Germany), Ben Chao (IES-AS,
Taiwan), Jianli Chen (U. Texas, USA), Lijing Cheng (IAP-CAS, China), John A. Church (U.
New South Wales, Australia), Stephen Chuter (U. Bristol, UK), J. Graham Cogley (Trent U.,
Canada), Soenke Dangendorf (U. Siegen, Germany), Damien Desbruyères (IFREMER,
France), Petra Döll (Goethe U., Germany), Catia Domingues (CSIRO, Australia), Ulrike Falk
(U. Bremen, Germany), James Famiglietti (JPL/Caltech, USA), Luciana Fenoglio-Marc (U.
Bonn, Germany), Rene Forsberg (DTU-SPACE, Denmark), Gaia Galassi (U. Urbino, Italy),
Alex Gardner (JPL/Caltech, USA), Andreas Groh (TU-Dresden, Germany), Benjamin
Hamlington (Old Dominion U., USA), Anna Hogg (U. Leeds, UK), Martin Horwath (TU-
Dresden, Germany), Vincent Humphrey (ETHZ, Switzerland), Laurent Husson (U. Grenoble,
France), Masayoshi Ishii (MRI-JMA, Japan), Adrian Jaeggi (U. Bern, Switzerland), Svetlana
Jevrejeva (NOC, UK), Gregory Johnson (NOAA/PMEL, USA), Jürgen Kusche (U. Bonn,
Germany), Kurt Lambeck (ANU, Australia), Felix Landerer (JPL/Caltech, USA), Paul
Leclercq (UIO, Norway), Benoit Legresy (CSIRO, Australia), Eric Leuliette (NOAA, USA),
William Llovel (LEGOS, France), Laurent Longuevergne (U. Rennes, France), Bryant D.
Loomis (NASA GSFC, USA), Scott B. Luthcke (NASA GSFC, USA), Marta Marcos (UIB,
Spain), Ben Marzeion (U. Bremen, Germany), Chris Merchant (U. Reading, UK), Mark
Merrifield (UCSD, USA), Glenn Milne (U. Ottawa, Canada), Gary Mitchum (U. South
Florida, USA), Yara Mohajerani (UCI, USA), Maeva Monier (Mercator-Ocean, France),
Steve Nerem (U. Colorado, USA), Hindumathi Palanisamy (LEGOS, France), Frank Paul
(UZH, Switzerland), Begoña Perez (Puertos del Estados, Spain), Christopher G. Piecuch
(WHOI, USA), Rui M. Ponte (AER inc., USA), Sarah G. Purkey (SIO/UCSD, USA), John T.
Reager (JPL/Caltech, USA), Roelof Rietbroek (U. Bonn, Germany), Eric Rignot (UCI and
JPL, USA), Riccardo Riva (TUDELFT, The Netherlands), Dean H. Roemmich (SIO/UCSD
USA), Louise Sandberg Sørensen (DTU-SPACE, Denmark), Ingo Sasgen (AWI, Germany),
E.J.O. Schrama (TUDELFT, The Netherlands), Sonia I. Seneviratne (ETHZ, Switzerland),
C.K. Shum (Ohio State U., USA), Giorgio Spada (U. Urbino, Italy), Detlef Stammer (U.
Hamburg, Germany), Roderic van de Wal (U. Utrecht, The Netherlands), Isabella Velicogna




(UCI and JPL, USA), Karina von Schuckmann (Mercator-Océan, France), Yoshihide Wada
(U. Utrecht, The Netherlands), Yiguo Wang (NERSC/BCCR, Norway), Christopher Watson
(U. Tasmania, Australia), David Wiese (JPL/Caltech, USA), Susan Wijffels (CSIRO,
Australia), Richard Westaway (U. Bristol, UK), Guy Woppelmann (U. La Rochelle, France),
Bert Wouters (U. Utrecht, The Netherlands)

**Acknowledgements and author contributions:**
This community assessment was initiated by A. Cazenave and B. Meyssignac as a
contribution to the Grand Challenge 'Regional Sea Level and Coastal Impacts' of the World
Climate Research Programme (WCRP). The results presented in this paper were prepared by
9 different teams dedicated to the various terms of the sea level budget (i.e., altimetry-based
sea level, tide gauges, thermal expansion, glaciers, Greenland, Antarctica, terrestrial water
storage, glacial isostatic adjustment, ocean mass from GRACE). Thanks to the team leaders
(in alphabetic order), M. Ablain, J. Bamber, N. Champollion, J. Chen, C. Domingues, S.
Jevrejeva, J.T. Reager, K. von Schuckmann, G. Spada, I. Velicogna and R. van de Wal, who
interacted with their team members, collected all needed information, provided a synthetized
assessment of the literature and when needed, updated the published results. The coordinators
A.C. and B.M. collected those materials and prepared a first draft of the manuscript, but all
authors contributed to its refinement and to the discussion of the results. Special thanks are
addressed to J. Benveniste, E. Berthier, G. Cogley, J. Church, G. Johnson (PMEL
Contribution Number 4776), B. Marzeion, F. Paul, R. Ponte, and E. Schrama for improving
the successive versions of the manuscript, and to H. Palanisamy for providing all figures
presented in section 3.
The views, opinions, and findings contained in this paper are those of the authors and should
not be construed as an official NOAA, U.S. Government or other institutions position, policy,
or decision.




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
