# Peer review of "Global Sea Level Budget 1993-Present"

_Earth System Science Data, 2018_

## Referee Comment (RC1) · Anonymous Referee #1 · 17 Jun 2018

Review of
**Global Sea Level Budget 1993 – Present** by
WCRP Global Sea Level Budget Group

This manuscript presents a review of the latest sea-level estimates, as well as of the single contributions (ocean thermal expansion, mass change from glaciers and ice sheets, land water, glacial isostatic adjustment). Bringing all the various data sets together, it assesses the sea-level budget over two periods with varying observational coverage of total mean sea level as well as sea-level contributions, namely the altimetry period (1993-present) and the GRACE/Argo period (2005-present). It is concluded that the sea-level budget is closed over the latter period when combining the contributions of thermal expansion (from Argo buoys) and ocean mass (from GRACE). However, larger discrepancies occur when the ocean mass contributions (glaciers, ice sheet, land water) are considered separately. The contribution of land water is identified as a major uncertainty over both periods considered.

The manuscript contains a wealth of information, an extensive review over the latest literature and estimates of sea level, it's contributions and the respective uncertainties over the periods of interest. Thus, it is very informative and provides a good overview over the state of the art. I have no major criticisms on the contents. However, as it is usually the case with a paper of so many authors, I found it at times hard to read with many slips of the pen, typos and inconsistencies (see specific comments). I propose the authors (or one or two of them) go carefully through the manuscript and improve the readability.

**General comments**
In general sea level is used without hyphen int the text, but not always. After my understanding, there is no hyphen if used as stand alone, but when it is used as compound adjective, e.g. sea-level budget, sea-level rise, sea-level change etc, there should be a hyphen. In any case, make sure you use it consistently.
Sometimes you give sea-level rates in mm/yr and sometimes in mm*yr$^{-1}$ – please choose one and stick to it.
Many abbreviations are used in this papers, sometimes without an explanation. As this study is quite interdisciplinary, not everybody is familiar with all the acronyms. I suggest you add a list at the end of the paper with all the acronyms and their explanation.
Also, make sure you use all you abbreviations consistently (GMSL, SLE, etc).

**Specific comments**

1. Introduction

Line 67: What do you mean with "constant evolution"?
Line 82-83: To me, it is not clear how contributions from marginal seas, shelf areas and polar regions can be inferred from the sea-level budget.

2. Methods and Data
2.2 Altimetry-based global mean sea level over 1993-present

Line 184: Replace ">" with "poleward of".
Line 190: "the ocean response to atmospheric dynamics": could you be more specific? Why does that need to be corrected for? Surely, we are interested in the sea level response to some of these processes.

Figure 1: there is still space to make the legend a little bigger.

    Line 204: are the time series centred with zero mean in 1993? It does not look like it from the figure?

    Line 205: subtracted *from* each data set.

Line 258: Removing the *(moving?) linear* trend... Is there a figure to show this? If not, I suggest you add a "not shown", as I was expecting a figure, particularly as you highlighted the differences between the various detrended GMSL curves at the end of the paragraph, but could not find it.

Figure 3: are the accelerations give in the legend from the publications or from fitting a linear trend to the curves shown in the figure?

**2.3 Steric sea level**

Line 332: "referred *to* as..."

Line 342: "But sum of …." there is something wrong with this sentence. Please re-write it.

Line 366: *insight* instead of *inside*?

Line 368-370: You are talking about *virtually global* and *quasi global coverage* here. What to you mean? How is *global* defined?

Table 2: the "*" that appears in the third (Period) column is not explained. Also, the caption should be a bit more informative.

Line 463: *calculation*

Figure 6, Line 508-510: I don't really understand what you mean with *relative to 1993-2016/2005-2016*?

Line 512: I guess it should be *1993* instead of *1996.*

**2.4 Glaciers**

Line 552-553: in the parentheses either change to *provides a review of these different methods* or remove *of.*

Line 570-571: what do you mean by *long-term observations?* Is there a minimum number of years, or so?

Line 581: for better readability, skip the "(" before *e.g.* and only have the reference (Marzeion) in parentheses.

Line 608: … and *are* distributed...

Figure 8: Also here, I think, the caption could be more informative in order to understand the figure. For example, state that the different estimates stem from different publications and are based on different methods.

**2.5 Greenland**

Line 687: ...*25-30% of...*

Line 695-697: Are there other uncertainties except for GIA that are large in Antarctica compared to Greenland? If not, I would suggest to skip the "for example".

Line 702: Remove the parentheses around the reference.

Line 704: *mass rates* instead of just *rates*?

Line 730: So, does that mean that the weights used to compute the mean mass change change every year? How does the final time series differ from the ordinary mean?

Figure 9, Line 740: Do you mean the weighted mean mass *change* instead of *trend* as shown in the figure? Also, shown in the figure are the values tabulated in Table 5 (in Gt/yr as opposed to Gt in the figure)? Do you have to show the values in the Table or could the data just be provided as a supplement (except for the last 4 rows of the Table).

Line 759: Too many parentheses around the references.

**2.6 Antarctica**

There is a recent publication from the IMBIE Team that could be mentioned in this section: https://www.nature.com/articles/s41586-018-0179-y
Line 775: *to* instead of *with*.
Table 6: In the caption, please mention that the Table refers to Antarctica.
Line 826: put the references in parentheses.
Line 826 and Table 6 (third row): the reference should read Forsberg, and "r" is missing.
Line 827: I think, at this point, it could be useful to very briefly explain what a mascon solution is.
Line 843: "We ?? a single..." - a verb is missing. And what is OIB?
Line 845: But in Table 7, only GRACE and IOM estimates are shown, no altimetry estimates.
Line 849: *the annual mass change* instead of *to annual mass change*?
Figure 10: Why don't you show the mass change as in Figure 9 for Greenland. It would be more consistent. Also, just as for Figure 9 and Table 5, is it necessary to show the values plotted in Figure 10 again in Table 6 (with a reversed sign)?
Line 894: Too many parentheses around the references.
Line 897: I'm not sure if you use the abbreviation ASE again in the text. If not, just remove it.

2.7 Terrestrial Water Storage
Line 923: The year is missing in the reference of Döll et al.
Line 990: What is PCR-GLOBWB?
Line 991: Over what period was this number computed. What would be the sea level equivalent?
Line 1014: I really think it would be useful to explain all these abbreviations in a list at the end of the text.
Line 1110: I'm not sure but I think the figure shows mm not mm/yr. At least that's what the label on the y-axis says.
Figure 13: What do the green lines in the left panel show?
Line 1147: *panel*
Line 1149: "Tropical regions *show*..." instead of *are*.
Line 1165: Put the references in parentheses.
Line 1172: show*s*

2.8 Glacial Isostatic Adjustment
Line 1208: Maybe you could add a "glacial" or so before ice sheet, such that it is clear that you are not talking about the present day ice sheets.
Line 1214: Remove "in"?
Line 1222-1223: "the variance of N_gia over the surface of the oceans is much reduced": what do you mean by that? Reduced compared to what?
Line 1227: "Since..." - this sentence does not make sense.
Line 1233-1234: Well, n is small compared to the global mean sea-level rise, but not compared to its uncertainty.
Line 1235: Table 9*a*.
Line 1238: How has n been weighted?
Line 1271: Table *9b*.
Line 1272: weigh*t*ed. How were the weights computed?
Line 1278: I would rather say *ice sheets* instead of *ice caps*.
Line 1281: Please spend a few more sentences on describing and interpreting Figure 14!
Line 1302: Table 9*d*. Also, since the GIA correction to TWS is discussed before the GIA correction to GRACE-based ice-sheet mass balance, consider swapping the according parts of Table 9.
Line 1306: Add a space between *an* and *accuracy*.
Line 1322: *them* instead of *tham*.
Line 1322: Table 9*c*.
Table 9: The format of Table 9 should be adapted to the format of the other tables.

2.9 Ocean mass change from GRACE

Line 1383: *constraint*

Line 1386-1393: This is a very long sentence. Consider splitting it in several sentences or us i), ii) and so on to list the major error sources.

Line 1395: Remove *on*.

Table 10, first row: please be more accurate with the column titles (also check all the other Tables) – the first column should be "source" or "publication" and the last one "ocean mass trend SLE (mm/yr)" or so.

Table 10, Line 1398: ocean *mass trend*

Line 1402-1404: I think this sentence should be moved to the end of the next paragraph (Line 1413). It seems a bit lost here.

Line 1409: Remove *to*.

Line 1431: What is $\Delta C20$?

Table 11: same comment as for Table 10 – be more accurate with the titles of the columns. E.g. first column should be "GRACE data product" or so, and second "linear trend (mm/yr)" or so.

Line 1481-1482: As far as I understood, it was launched in March 2018, wasn't it? (Instead of "is scheduled to be launched")

3. Sea Level Budget results

Line 1532: Don't you mean Table 13 instead of 12?

Section 3.2.2: A very short section. I think, even though you discuss it in the Discussion, in this section you should at least comment on the large discrepancy between Row 7 and 8 in Table 13, and how it relates to the large uncertainty in TWS.

Line 1549-1551: Be more accurate: the table provides annual mean values for the ensemble mean GMSL and the *sum of components* (GRACE-based ocean mass and Argo-based thermosteric component).

Line 1567: Is the residual trend statistically significant?

5. Concluding Remarks

Line 1728: for *example*

Line 1730: Remove the long term for NASA to a list of abbreviations at the end of the text.

---

## Author Comment (AC1) · 8 Jul 2018

Dear Editor We have taken into account almost all comments made by Reviewer 1 on our manuscript. I attach 2 files of the revised version: (1) with tracked changes, and (2) a clean version with untracked changes. I also attach the response to the Rev.1's comments. We thank the reviewer for his/her careful reading of the manuscript and his/her many minor comments that will significantly improve the presentation of the results. Yours Sincerely Anny Cazenave Corresponding author

Please also note the supplement to this comment:
https://www.earth-syst-sci-data-discuss.net/essd-2018-53/essd-2018-53-AC1-supplement.pdf

[Figure]

[Figure]

**Supplement:**

**Global Sea Level Budget 1993-Present**

WCRP Global Sea Level Budget Group*

*A full list of authors and their affiliations appears at the end of the paper

**Revised version**

**8 July 2018**

Corresponding author: Anny Cazenave, LEGOS, 18 Avenue Edouard Belin, 31401 Toulouse, Cedex 9, France;  anny.cazenave@legos.obs-mip.fr

**Abstract**

Global mean sea level is an integral of changes occurring in the climate system in response to unforced climate variability as well as natural and anthropogenic forcing factors. Its temporal evolution allows detecting changes (e.g., acceleration) in one or more components. Study of the sea level budget provides constraints on missing or poorly known contributions, such as the unsurveyed deep ocean or the still uncertain land water component. In the context of the World

Climate Research Programme Grand Challenge entitled "Regional Sea Level and Coastal

Impacts", an international effort involving the sea level community worldwide has been recently initiated with the objective of assessing the various data sets used to estimate components of the sea level budget during the altimetry era (1993 to present). These data sets are based on the combination of a broad range of space-based and in situ observations, model estimates and algorithms. Evaluating their quality, quantifying uncertainties and identifying sources of discrepancies between component estimates is extremely useful for various applications in climate research. This effort involves several tens of scientists from about fifty research teams/institutions worldwide (www.wcrp-climate.org/grand-challenges/gc-sea-level).

The results presented in this paper are a synthesis of the first assessment performed during

2017-2018. We present estimates of the altimetry-based global mean sea level (average rate of

3.1 +/- 0.3 mm/yr and acceleration of 0.1 mm/yr$^2$ over 1993-present), as well as of the different components of the sea level budget (2.9 Ocean mass change from GRACE Line 1383:

constraint Line 1386-1393: This is a very long sentence. Consider splitting it in several sentences or us i), ii) and so on to list the major error sources. Line 1395: Remove on. Table

10, first row: please be more accurate with the column titles (also check all the other Tables) –

the first column should be "source" or "publication" and the last one "ocean mass trend SLE

(mm/yr)" or so. Table 10, Line 1398: ocean mass trend Line 1402-1404: I think this sentence should be moved to the end of the next paragraph (Line 1413). It seems a bit lost here. Line

1409: Remove to. Line 1431: What is $\Delta C20$? Table 11: same comment as for Table 10 – be more accurate with the titles of the columns. E.g. first column should be "GRACE data product"

or so, and second "linear trend (mm/yr)" or so. Line 1481-1482: As far as I understood, it was launched in March 2018, wasn't it? (Instead of "is scheduled to be launched") 3. Sea Level

Budget results Line 1532: Don't you mean Table 13 instead of 12? Section 3.2.2: A very short section. I think, even though you discuss it in the Discussion, in this section you should at least comment on the large discrepancy between Row 7 and 8 in Table 13, and how it relates to the large uncertainty in TWS. Line 1549-1551: Be more accurate: the table provides annual mean values for the ensemble mean GMSL and the sum of components (GRACE-based ocean mass and Argo-based thermosteric component). Line 1567: Is the residual trend statistically
significant? 5. Concluding Remarks Line 1728: for example Line 1730: Remove the long term

[revised manuscript text omitted]

In Table 6, the negative trend estimate by Zwally et al. (2016) is not added. It is worth noting that including it would only slightly reduce the ensemble mean trend.

**2.6.2 Methods and analyses**

The datasets used in this assessment are Antarctica mass balance time series generated using different approaches. Two estimates are a joint inversion of GRACE/altimetry/GPS data (Martín-Español et al.,2016), and GRACE and CryoSat data (Forsberg et al.,2017). Two methods are mascon solutions obtained from the GRACE intersatellite range-rate measurements over equal-area spherical caps covering the Earth' surface (Luthcke et al., 2013; Wiese et al., 2016), three estimates use the GRACE spherical harmonics solutions (Velicogna et al., 2014; Wiese et al., 2016; Wouters et al., 2013) and one gridded GRACE products (Sasgen et. al., 2013).

All GRACE time series were provided as monthly time series except for the one using the Martín-Español et al. (2016) method that were provided as annual estimates. In addition, different groups use different GIA corrections, therefore the spread of the trend solutions represents also the error associated to the GIA correction which, in Antarctica, is the largest source of uncertainty. Sasgen et al. (2013) used their own GIA solution (Sasgen et al., 2017), Martín-Español et al. (2016) as well, Luthcke et et al., (2013), Velicogna et al. (2014) and Groh and Horwath (2016) used IJ05-R2 (Ivins et al., 2013), Wouter et al. (2013) used Whitehouse et al. (2012), and Wise et al. (2016) used A et al. (2013). In addition, Groh and Horwath (2016) did not include the peripheral glaciers and ice caps, while all other estimates do.

Table 6 shows the Antarctic contribution to sea level during 2005-2015 from the different GRACE solutions, and for the input and output method (IOM).. There is a single IOM-based datasetthat provides trends for the period 1993-2015 (update of Rignot et al., 2011). 
[revised manuscript text omitted]

according to works in the literature where various GIA model models and averaging methods have been employed. Based on values in Table 9a for which a standard deviation is available, the average of $n$ (weighted by the inverse of associated errors), assumed to represent the best estimate, is $n = (-0.29 \pm 0.02)$ mm/yr where the uncertainty corresponds to $2\sigma$.

### *2.8.2 GIA correction to GRACE-based ocean mass*

GRACE observations of present-day gravity variations are sensitive to GIA, due to the sheer amount of rock material that is transported by GIA throughout the mantle and the resulting changes in surface topography, especially over the formerly glaciated areas. The continuous change in the gravity field results in a nearly linear signal in GRACE observations. Since the gravity field is determined by global mass redistribution, GIA models used to correct GRACE

data need to be global as well, especially when the region of interest is represented by all ocean areas. To date, the only global ice reconstruction publicly available is provided by the

University of Toronto. Their latest product, named ICE-6G, has been published and distributed in 2015 (Peltier et al., 2015); note that the ice history has been simultaneously constrained with a specific Earth model, named VM5a. During the early period of the GRACE mission, the available Toronto model was ICE-5G (VM2) (Peltier, 2004). However, different groups have independently computed GIA model solutions based on the Toronto ice history reconstruction, by using different implementations of GIA codes and somehow different Earth models. The most widely used model is the one by Paulson et al. (2007), later updated by A et al. (2013).

Both studies use a deglaciation history based on ICE-5G, but differ for the viscosity profile of the mantle: A et al. use a 3D compressible Earth with VM2 viscosity profile and a PREM-based elastic structure used by Peltier (2004), whereas Paulson et al. (2007) use an incompressible

Earth with self-gravitation, and a Maxwell 1-D multi-layer mantle. Over most of the oceans, the GIA signature is much smaller than over the continents. However, once integrated over the global ocean, the signal $w$ due to GIA is about -1 mm/yr of equivalent sea level change (Chambers et al., 2010), which is of the same order of magnitude as the total ocean mass change induced by increased ice melt (Leuliette and Willis, 2011). The main uncertainty in the GIA

contribution to ocean mass change estimates, apart from the general uncertainty in ice history and Earth mechanical properties, originates from the importance of changes in the orientation of the Earth's rotation axis (Chambers et al., 2010, Tamisiea, 2011). Different choices in implementing the so-called "rotational feedback" lead to significant changes in the resulting

GIA contribution to GRACE estimates. The issue of properly accounting from rotational effects has not been settled yet (Mitrovica et al., 2005, Peltier and Luthcke, 2009, Mitrovica and Wahr,

2011, Martinec and Hagedoorn, 2014). Table 9bsummarises the values of the mass-rate GIA

contribution $w$ according to the literature, where various models and averaging methods are employed. The weighed average of the values in Table 9b for which an assessment of the standard deviation is available, is $w$ = (-1.44 ± 0.36) mm/yr (the uncertainty is $2\sigma$), which we assume to represent the preferred estimate.

*2.8.3 GIA correction to GRACE-based terrestrial water storage*

As discussed in the previous section, the GIA correction to apply to GRACE over land is significant, especially in regions formely covered by the ice sheets (Canada and Scandinavia).

Over Canada, GIA models significantly differ. This is illustrated in Figure 13 that shows difference between two models of GIA correction to GRACE over land, the A et al. (2013) and

Peltier et al. (2009) models. We see that over the majority of the land areas, differences are small, except over north Canada, in particular around the Hudson Bay, where differences larger than +/- 20 mm/yr SLE are noticed. This may affect GRACE-based TWS estimates over

Canadian river basins.

MAP OF GIA TREND DIFFERENCE BETWEEN A. ET AL. AND PELTIER ICE6G_RC

[revised manuscript text omitted]

(c) GIA contribution to GRACE-based terrestrial water storage change

| *Reference* | *GIA correction (mm/yr SLE)* |
| --- | --- |
| | *(without Greenland, Antacrtica, Iceland,* |
| | *Svalsbard, Hudson Bay and Black Sea* |
| | |
| A et al. (2013) | 0.63 |
| Peltier ICE5G | 0.68 |
| Peltier ICE6G_rc | 0.71 |
| ANU_ICE6G | 0.53 |

(d) GIA contribution to GRACE mass-rate of the ice sheets

| *Reference* | ***Greenland*** *GIA (Gt/yr)* | *Notes* |
| --- | --- | --- |
| Simpson et al. (2009)[r] | -3 ± 12[m] | Thermodynamic sheet / solid Earth model, 1D (uncoupled); constrained by geomorphology; inversion results in Sutterley *et al.* (2014). |
| Peltier (2009) (ICE-5G)[g] | -4[f] | Ice load reconstruction / solid Earth model, 1D (ICE-5G / similar to VM2); Greenland component of ICE-5G (13 Gt/yr) + Laurentide component of ICE-5G (-17 Gt/yr); inversion results in Khan *et al.* 2016, Discussion. |
| Khan *et al.* (2016) (GGG-1D)[r] | 15 ± 10[f] | Ice load reconstruction / solid Earth model, 1D (uncoupled); constrained with geomorphology & GPS; Greenland component (+32 Gt/yr) + Laurentide component of ICE-5G (-17 Gt/yr); inversion results in Khan *et al.* (2016), Discussion. |
| Fleming *et al.* (2004)[r] (Green1) | 3[f] | Ice load reconstruction / solid Earth model, 1D (uncoupled); constrained with geomorphology; Greenland component (+ 20 Gt/yr) + Laurentide component of ICE- |

5G (-17 Gt/yr); inversion in Sasgen *et al.* (2012, supplement).

| Reference | | | Notes |
|---|---|---|---|
| Wu *et al.* (2010)[g] | -69 ± 19[m] | | Joint inversion estimate based on GPS, satellite laser ranging, and very long baseline interferometry, and bottom pressure from ocean model output; inversion results in Sutterley *et al.* (2014). |
| *Reference* | ***Antarctica GIA (Gt/yr)*** | *Notes* | |
| Whitehouse *et al.* (2012) (W12a)[r] | 60[n] | | Thermodynamic sheet / solid Earth model, 1D (uncoupled); constrained by geomorphology; inversion results in Shepherd *et al.* 2012, supplement (Fig. S8). |
| Ivins *et al.* (2013) (IJ05_R2)[r] | 40-65[n] | | Ice load reconstruction / solid Earth model, 1D; constrained by geomorphology and GPS uplift rates; Ivins et al. 2013; inversion results in Shepherd *et al.* 2012, Supplement (Fig. S8). |
| Peltier (2009) (ICE-5G)[g] | 140-180[n] | | Ice load reconstruction / solid Earth model ICE-5G(VM2); constrained by geomorphology; inversion results in Shepherd *et al.* 2012, supplement (Fig. S8). |
| Argus *et al.* (2014) (ICE-6G)[g] | 107[n] | | Ice load reconstruction / solid Earth model ICE-6G(VM5a); constrained by geomorphology and GPS; theory recently corrected by Purcell *et al.* 2016; inversion results in Argus *et al.* (2014), conclusion 7.8. |
| Sasgen *et al.* (2017) (REGINA)[r] | 55 ± 22[f] | | Joint inversion estimate based on GRACE, altimetry, GPS and viscoelastic response functions; lateral heterogeneous Earth model parameters; inversion results in Sasgen *et al.* (2017), Table 1. |

[revised manuscript text omitted]

With a different treatment of the GRACE land-ocean signal leakage effect through global forward modeling, Chen et al. (2013) estimated ocean mass rates using GRACE RL05 time-variable gravity solutions over the period 2005-2011, and showed that the ocean mass change contributes to 1.80 ± 0.47 
[revised manuscript text omitted]

We are grateful to the anonymous reviewer for his/her throrough comments that helped to improve the manuscript.

---

## Editor Comment (EC1) · G. M. R. Manzella (Editor) · 18 Jul 2018

Comment on

Essd-2018-53     Submitted on 13 Apr 2018
Global Sea Level Budget 1993-Present
WCRP Global Sea Level Budget Group

The paper is presenting a very good overview of works on sea level rise.

It could be complete if the authors introduce a small discussion on the little glacial era (see the lines 78 – 82 of the introduction) and also the so called Munk enigma.

A key study is presented in a paper by Walter Munk published in 2002[1]. Its importance is lying on the careful analysis made on causes and effects, as in schematised in Figure 1.

It is a seminal study since is obliging researcher to give attention to all factors influencing sea level.

Studies were demonstrating that after the little ice age early in the 19th century, sea level rose at 18 cm/century (cm/cy - the historic rate) with no measurable acceleration until the mid-20th century, when thermal expansion associated with greenhouse warming became significant, contributing an additional 3 cm by the year 2000. Greenhouse-related sea level rise has accelerated to the present rate of 6 cm/cy, making the historic + greenhouse rate 24 cm/cy (Figure 4.2.1A). The relative heat content between the sea surface and 3000 m depth (Figure 2B) and the global temperature changes[2] (Figure 4.2.1C) are providing the base for the calculation of a steric sea level rise by 12 cm/cy, as represented by the line AB.

[Figure]

[1] Munk Walter (2002). Twentieth century sea level: an enigma. Proceedings of the National Academy of Sciences of the United States of America, vol. 99 no. 10, 6550–6555, doi: 10.1073/pnas.092704599
[2] Levitus, S., Antonov, J. I., Wang, J., Delworth, T. L., Dixon, K. W. & Broccoli, A. J. (2001) Science 292, 267–270.

Figure 1. Cartoon from Munk 2002, showing the contribution to sea level trends.

Munk addressed the attention of researchers to the IPCC 2001[3] rate of sea level rise that was composed by the sea level in 2000 referred to 1900 (21 cm), the greenhouse contribution (3 cm) and the 'best estimate' of eustatic contribution (6 cm):

$$\zeta - \zeta_{greenhose} - \zeta_{eustatic} = 21 - 3 - 6 = 12 \text{ cm}$$

rise unaccounted for (the Munk sea level rise enigma)!

It can be published after the revision suggested by the referee and additional text related to the above comments.
* * *
[3] Church, J. A., Gregory, J. M., Huybrechts, P., Kuhn, M., Lambeck, K., Nhuan, M. T., Qin, D. & Woodworth, P. L. (2001) in *Climate Change 2001: The Scientific Basis* (Cambridge Univ. Press, Cambridge, U.K.), pp. 639−694.

---

## Author Comment (AC2) · 22 Jul 2018

Reviewer 2's comment: The paper is presenting a very good overview of works on sea level rise. It could be complete if the authors introduce a small discussion on the little glacial era (see the lines 78 – 82 of the introduction) and also the so called Munk enigma. A key study is presented in a paper by Walter Munk published in 2002[1]. Its importance is lying on the careful analysis made on causes and effects, as in schematised in Figure 1. [1] It is a seminal study since is obliging researcher to give attention to all factors influencing sea level. Studies were demonstrating that after the little ice age early in the 19th century, sea level rose at 18 cm/century (cm/cy - the historic rate) with no measurable acceleration until the mid-20th century, when thermal expansion associated with greenhouse warming became significant, contributing an additional 3

cm by the year 2000. Greenhouse-related sea level rise has accelerated to the present rate of 6 cm/cy, making the historic + greenhouse rate 24 cm/cy (Figure 4.2.1A). The relative heat content between the sea surface and 3000 m depth (Figure 2B) and the global temperature changes2 (Figure 4.2.1C) are providing the base for the calculation of a steric sea level rise by 12 cm/cy, as represented by the line AB. Munk addressed the attention of researchers to the IPCC 20013 rate of sea level rise that was composed by the sea level in 2000 referred to 1900 (21 cm), the greenhouse contribution (3 cm) and the 'best estimate' of eustatic contribution (6 cm): 3 $\zeta - \zeta$greenhose $- \zeta$eustatic = 21 − 3 − 6 = 12 cm rise unaccounted for (the Munk sea level rise enigma)! It can be published after the revision suggested by the referee and additional text related to the above comments. Response: We thank Reviewer 2 for his comment about the Munk's enigma. Munk's paper was very popular when it was published. However at that time (16 years ago), limited information was available about the components of the sea level budget. Following Munk's original paper, Mitrovica et al. 2006) proposed an improved theory of rotational stability of the Earth, effectively removed the constraints proposed by Munk (2002). This theory allows a polar ice sheet contribution to 20th century sea-level rise of as much as ∼1.1 mm/yr, with about 0.8 mm/yr beginning in the 20th century. In addition, since then, recent re-estimates based on tide gauges of the 20th century sea level rise suggest that the 20th century rate could have been lower than assumed in Munk's paper. Moreover, new studies by Gregory et al. (2013) and Slangen et al. (2017) based on observations and models, show that the 20th century sea level budget can be approximately explained within uncertainties. We extract below a few sentences from these two articles. Extract from Gregory et al., 2013: "Considering the twentieth century as a whole, Munk (2002) described GMSLR as an "enigma": it began too early, it had too linear a trend, and it was too large. The first two problems relate to an expectation, based on a general understanding of the processes concerned, that in a warmer climate the rates of thermal expansion and of glacier mass loss will tend to increase. Therefore, we might suppose that these climate-related contributions to GMSLR increased during the twentieth century. However, the trend of GMSLR during

recent decades was actually not very much larger than during the twentieth century as a whole. For instance, Church and White (2011) find 1.9 +/- 0.4 mm/yr for 1961–2009 and 1.7 +/- 0.2 mm/yr for 1900–2009. The third problem, of GMSLR being too large, is shown by the model-derived estimates of contributions reported by Church et al. (2001), which could explain only 50% of twentieth century GMSLR. To balance the budget and explain the form of the time series requires time-dependent information about the contributions to GMSLR throughout the century. We are enabled to make progress by new work summarized in this paper (Gregory et al., 2013) regarding the contributions from thermal expansion, glaciers, the Greenland ice sheet, groundwater depletion, and reservoir impoundment."

Extract from Slangen et al., 2017: "In recent decades, one of the major questions in sea-level research has been the closure of the 20th century (global mean) sea-level budget (Munk 2002; Gregory et al. 2013a). There are two parts to this question: firstly, the sum of the observations of the individual contributions tend to underestimate the total observed change, raising the issue of different types of uncertainties in the observations. Secondly, the sum of model-based sea-level contributions tend to underestimate the total observed change for the 20th century, probably due to a combination of uncertainties/imperfections in both models and observations. The observations of total sea-level change and the individual contributions are spatially and temporally sparse, difficult to quality-control, and biased to the Northern Hemisphere (and therefore perhaps not representative of GMSL, Thompson et al. 2016), until satellite data started to become available in the early 1990's. The model-based contributions on the other hand, may not fully account for all climate variability, such as multi-decadal variations in the ocean or the delayed response of glaciers and ice sheets to externally-driven climate change. A couple of years ago, a large community effort (Gregory et al. 2013a) explored a wide range of observational estimates of all contributions to sea-level change, and managed to close the observational budget to within uncertainties. In this paper we will focus on the second part of the problem, namely at reconciling the summed model estimates with total observed GMSL change. This is an important

topic, as a better understanding of and ability to model past sea-level change increases confidence in the models' ability to project future sea-level changes."

Response to Reviewer 2 (continued): In the revised version, we added a few lines about the 20th century sea level budget. Lines 113 to 118 of the revised manuscript: "For example, Munk (2002) found that the 20th century sea-level rise could not be closed with the data available at that time and showed that if the missing contribution were due to polar ice melt, this would be in conflict with external astronomical constraints. The enigma has been resolved in two ways. Firstly, an improved theory of rotational stability of the Earth effectively removed the constraints proposed by Munk (2002), and allows a polar ice sheet contribution to 20th century sea-level rise of as much as ∼1.1 mm/yr, with about 0.8 mm/yr beginning in the 20th century. In addition, more recent studies by Gregory et al. (2013) and Slangen et al. (2017), combining observations with model estimates, showed that it was possible to effectively close the 20th century sea level budget within uncertainties." As a result of these two areas of progress, we consider that Walter Munk's 'enigma' is now effectively solved.

Please also note the supplement to this comment:
https://www.earth-syst-sci-data-discuss.net/essd-2018-53/essd-2018-53-AC2-supplement.pdf

---

## Author Comment (AC3) · 23 Jul 2018

Response to Reviewer 2 (update)

Reviewer 2's comment: The paper is presenting a very good overview of works on sea level rise. It could be complete if the authors introduce a small discussion on the little glacial era (see the lines 78 – 82 of the introduction) and also the so called Munk enigma. A key study is presented in a paper by Walter Munk published in 2002[1]. Its importance is lying on the careful analysis made on causes and effects, as in schematised in Figure 1. 1 It is a seminal study since is obliging researcher to give attention to all factors influencing sea level. Studies were demonstrating that after the little ice age early in the 19th century, sea level rose at 18 cm/century (cm/cy - the historic rate)

with no measurable acceleration until the mid-20th century, when thermal expansion associated with greenhouse warming became significant, contributing an additional 3 cm by the year 2000. Greenhouse-related sea level rise has accelerated to the present rate of 6 cm/cy, making the historic + greenhouse rate 24 cm/cy (Figure 4.2.1A). The relative heat content between the sea surface and 3000 m depth (Figure 2B) and the global temperature changes2 (Figure 4.2.1C) are providing the base for the calculation of a steric sea level rise by 12 cm/cy, as represented by the line AB. Munk addressed the attention of researchers to the IPCC 20013 rate of sea level rise that was composed by the sea level in 2000 referred to 1900 (21 cm), the greenhouse contribution (3 cm) and the 'best estimate' of eustatic contribution (6 cm): 3 $\zeta - \zeta$greenhose $- \zeta$eustatic = 21 − 3 − 6 = 12 cm rise unaccounted for (the Munk sea level rise enigma)! It can be published after the revision suggested by the referee and additional text related to the above comments. Response: We thank Reviewer 2 for his comment about the Munk's enigma. Munk's paper was very popular when it was published. However at that time (16 years ago), limited information was available about the components of the sea level budget. Following Munk's original paper, Mitrovica et al. (2006) proposed an improved theory of rotational stability of the Earth, effectively removed the constraints proposed by Munk (2002). This theory allows a polar ice sheet contribution to 20th century sea-level rise of as much as ∼1.1 mm/yr, with about 0.8 mm/yr beginning in the 20th century. In addition, since then, recent re-estimates based on tide gauges of the 20th century sea level rise suggest that the 20th century rate could have been lower than assumed in Munk's paper. Moreover, new studies by Gregory et al. (2013) and Slangen et al. (2017) based on observations and models, show that the 20th century sea level budget can be approximately explained within uncertainties. We extract below a few sentences from these two articles. Extract from Gregory et al., 2013: "Considering the twentieth century as a whole, Munk (2002) described GMSLR as an "enigma": it began too early, it had too linear a trend, and it was too large. The first two problems relate to an expectation, based on a general understanding of the processes concerned, that in a warmer climate the rates of thermal expansion and of glacier mass loss will tend to increase. Therefore, we might suppose that these climate-related contributions to GMSLR increased during the twentieth century. However, the trend of GMSLR during recent decades was actually not very much larger than during the twentieth century as a whole. For instance, Church and White (2011) find 1.9 +/- 0.4 mm/yr for 1961–2009 and 1.7 +/- 0.2 mm/yr for 1900–2009. The third problem, of GMSLR being too large, is shown by the model-derived estimates of contributions reported by Church et al. (2001), which could explain only 50% of twentieth century GMSLR. To balance the budget and explain the form of the time series requires time-dependent information about the contributions to GMSLR throughout the century. We are enabled to make progress by new work summarized in this paper (Gregory et al., 2013) regarding the contributions from thermal expansion, glaciers, the Greenland ice sheet, groundwater depletion, and reservoir impoundment."

Extract from Slangen et al., 2017: "In recent decades, one of the major questions in sea-level research has been the closure of the 20th century (global mean) sea-level budget (Munk 2002; Gregory et al. 2013a). There are two parts to this question: firstly, the sum of the observations of the individual contributions tend to underestimate the total observed change, raising the issue of different types of uncertainties in the observations. Secondly, the sum of model-based sea-level contributions tend to underestimate the total observed change for the 20th century, probably due to a combination of uncertainties/imperfections in both models and observations. The observations of total sea-level change and the individual contributions are spatially and temporally sparse, difficult to quality-control, and biased to the Northern Hemisphere (and therefore perhaps not representative of GMSL, Thompson et al. 2016), until satellite data started to become available in the early 1990's. The model-based contributions on the other hand, may not fully account for all climate variability, such as multi-decadal variations in the ocean or the delayed response of glaciers and ice sheets to externally-driven climate change. A couple of years ago, a large community effort (Gregory et al. 2013a) explored a wide range of observational estimates of all contributions to sea-level change, and managed to close the observational budget to within uncertainties.

In this paper we will focus on the second part of the problem, namely at reconciling the summed model estimates with total observed GMSL change. This is an important topic, as a better understanding of and ability to model past sea-level change increases confidence in the models' ability to project future sea-level changes."

Response to Reviewer 2 (continued): In the revised version, we added a few lines about the 20th century sea level budget. Lines 113 to 122 of the revised manuscript: "For example, Munk (2002) found that the 20th century sea-level rise could not be closed with the data available at that time and showed that if the missing contribution were due to polar ice melt, this would be in conflict with external astronomical constraints. The enigma has been resolved in two ways. Firstly, an improved theory of rotational stability of the Earth (Mitrovica et al., 2006) effectively removed the constraints proposed by Munk (2002), and allows a polar ice sheet contribution to 20th century sea-level rise of as much as ∼1.1 mm/yr, with about 0.8 mm/yr beginning in the 20th century. In addition, more recent studies by Gregory et al. (2013) and Slangen et al. (2017), combining observations with model estimates, showed that it was possible to effectively close the 20th century sea level budget within uncertainties." As a result of these two areas of progress, we consider that Walter Munk's 'enigma' is now effectively solved.

References: 1. Gregory, J. M., N. J. White, J. A. Church, M. F. P. Bierkens, J. E. Box, M. R. van den Broeke, J. G. Cogley, X. Fettweis, E. Hanna, P. Huybrechts, L. F. Konikow, P. W. Leclercq, B. Marzeion, J. Oerlemans, M. E. Tamisiea, Y. Wada, L. M.Wake, R. S. W. van de Wal, Twentieth-Century Global-Mean Sea Level Rise: Is the Whole Greater than the Sum of the Parts? J. Climate, 26, 4476–4499, doi:10.1175/JCLI-D-12-00319.1, 2013. 2. Mitrovica J.X., Wahr J., Matsuyama I., Paulson A., and Tamisiea M.E., Reanalysis of ancient eclipse, astronomic and geodetic data: A possible route to resolving the enigma of global sea-level rise. Earth and Planetary Science Letters, 243, 3-4, 390-399, doi: 10.1016/j.epsl.2005.12.029, 2006. 3. Slangen, A.B.A., Meyssignac, B., Agosta, C., Champollion, N., Church, J.A., Fettweis, X., Ligtenberg, S.R.M., Marzeion, B., Melet, A., Palmer, M.D., Richter, K., Roberts, C.D., Spada, G.,

[Figure]

Evaluating model simulations of 20th century sea-level rise. Part 1: global mean sea-level change. J. Clim. 30, 21, 8539–8563. https://dx.doi.org/10.1175/jcli-d-17-0110.1, 2017.

Please also note the supplement to this comment:
https://www.earth-syst-sci-data-discuss.net/essd-2018-53/essd-2018-53-AC3-supplement.pdf

[Figure]

**Supplement:**

**Global Sea Level Budget 1993-Present**

WCRP Global Sea Level Budget Group*

*A full list of authors and their affiliations appears at the end of the paper

**Revised version**
**23 July 2018**

Corresponding author: Anny Cazenave, LEGOS, 18 Avenue Edouard Belin, 31401 Toulouse, Cedex 9, France;  anny.cazenave@legos.obs-mip.fr

**Abstract**

Global mean sea level is an integral of changes occurring in the climate system in response to unforced climate variability as well as natural and anthropogenic forcing factors. Its temporal evolution allows detecting changes (e.g., acceleration) in one or more components. Study of the sea level budget provides constraints on missing or poorly known contributions, such as the unsurveyed deep ocean or the still uncertain land water component. In the context of the

World Climate Research Programme Grand Challenge entitled "Regional Sea Level and

Coastal Impacts", an international effort involving the sea level community worldwide has been recently initiated with the objective of assessing the various data sets used to estimate components of the sea level budget during the altimetry era (1993 to present). These data sets are based on the combination of a broad range of space-based and in situ observations, model estimates and algorithms. Evaluating their quality, quantifying uncertainties and identifying sources of discrepancies between component estimates is extremely useful for various applications in climate research. This effort involves several tens of scientists from about fifty research teams/institutions worldwide (www.wcrp-climate.org/grand-challenges/gc-sea- level). The results presented in this paper are a synthesis of the first assessment performed during 2017-2018. We present estimates of the altimetry-based global mean sea level (average rate of 3.1 +/- 0.3 mm/yr and acceleration of 0.1 mm/yr$^2$ over 1993-present), as well as of the different components of the sea level budget (http://doi.org/10.17882/54854). We further examine closure of the sea level budget, comparing the observed global mean sea level with the sum of components. Ocean thermal expansion, glaciers, Greenland and Antarctica contribute by 42%, 21%, 15% and 8% to the global mean sea level over the 1993-present. We also study the sea level budget over 2005-present, using GRACE-based ocean mass estimates instead of sum of individual mass components. Our results demonstrate that the global mean sea level can be closed to within 0.3 mm/yr (one sigma). Substantial uncertainty remains for the land water storage component, as shown in examining individual mass contributions to sea level.

**1. Introduction**

Global warming has already several visible consequences, in particular increase of the Earth's mean surface temperature and ocean heat content (Rhein et al., 2013, Stocker et al., 2013), melting of sea ice, loss of mass of glaciers (Gardner et al., 2013), and ice mass loss from the Greenland and Antarctica ice sheets (Rignot et al., 2011, Shepherd et al., 2012). On average over the last 50 years, about 93% of heat excess accumulated in the climate system because of greenhouse gas emissions has been stored in the ocean, and the remaining 7% has been warming the atmosphere and continents, and melting sea and land ice (von Schuckmann et al., 2016). Because of ocean warming and land ice mass loss, sea level rises. Since the end of the last deglaciation about 3000 years ago, sea level remained nearly constant (e.g., Lambeck et al., 2010, Kemp et al., 2011, Kopp et al. 2014). However, direct observations from in situ tide gauges available since the mid-to-late 19[th] century show that the 20[th] century global mean sea level has started to rise again at a rate of 1.2 mm/yr to 1.9 mm/yr (Church and White, 2011, Jevrejeva et al., 2014a, Hay et al., 2015, Dangendorf et al., 2017). Since the early 1990s sea level rise is measured by high-precision altimeter satellites and the rate has increased to ~3 mm/yr on average (Legeais et al., 2018, Nerem et al., 2018).

Accurate assessment of present-day global mean sea level variations and its components (ocean thermal expansion, ice sheet mass loss, glaciers mass change, changes in land water storage, etc.) is important for many reasons. The global mean sea level is an integral of changes occurring in the Earth's climate system in response to unforced climate variability as well as natural and anthropogenic forcing factors e.g., net contribution of ocean warming, land ice mass loss, and changes in water storage in continental river basins. Temporal changes of the components are directly reflected in the global mean sea level curve. If accurate enough, study of the sea level budget provides constraints on missing or poorly known contributions, e.g., the deep ocean or polar regions under sampled by current observing systems, or still uncertain changes in water storage on land due to human activities (e.g. ground water depletion in aquifers). Global mean sea level corrected for ocean mass change in principle allows one to independently estimate temporal changes in total ocean heat content, from which the Earth's energy imbalance can be deduced (von Schuckmann et al., 2016). The sea level and/or ocean mass budget approach can also be used to constrain models of Glacial Isostatic Adjustment (GIA). The GIA phenomenon has significant impact on the interpretation of GRACE-based space gravimetry data over the oceans (for ocean mass change) and over Antarctica (for ice sheet mass balance). However, there is still incomplete consensus on best estimates, a result of uncertainties in deglaciation models and mantle viscosity structure. Finally, observed changes of the global mean sea level and its components are fundamental for validating climate models used for projections.

In the context of the Grand Challenge entitled "Regional Sea Level and Coastal Impacts" of the World Climate Research Programme (WCRP), an international effort involving the sea level community worldwide has been recently initiated with the objective of assessing the sea level budget during the altimetry era (1993 to present). To estimate the different components of the sea level budget, different data sets are used. These are based on the combination of a broad range of space-based and in situ observations. Evaluating their quality, quantifying their uncertainties, and identifying the sources of discrepancies between component estimates, including the altimetry-based sea level time series, are extremely useful for various applications in climate research.

Several previous studies have addressed the sea level budget over different time spans and using different data sets. For example, Munk (2002) found that the 20[th] century sea-level rise could not be closed with the data available at that time and showed that if the missing contribution were due to polar ice melt, this would be in conflict with external astronomical constraints. The enigma has been resolved in two ways. Firstly, an improved theory of rotational stability of the Earth (Mitrovica et al., 2006) effectively removed the constraints proposed by Munk (2002), and allows a polar ice sheet contribution to 20[th] century sea-level rise of as much as ~1.1 mm/yr, with about 0.8 mm/yr beginning in the 20[th] century. In addition, more recent studies by Gregory et al. (2013) and Slangen et al. (2017), combining observations with model estimates, showed that it was possible to effectively close the 20[th] century sea level budget within uncertainties. For the altimetry era, many studies have investigated closure of the sea level budget (
[revised manuscript text omitted]

In Table 6, the negative trend estimate by Zwally et al. (2016) is not added. It is worth noting that including it would only slightly reduce the ensemble mean trend.

**2.6.2 Methods and analyses**

The datasets used in this assessment are Antarctica mass balance time series generated using different approaches. Two estimates are a joint inversion of GRACE/altimetry/GPS data (Martín-Español et al.,2016), and GRACE and CryoSat data (Forsberg et al.,2017). Two methods are mascon solutions obtained from the GRACE intersatellite range-rate measurements over equal-area spherical caps covering the Earth' surface (Luthcke et al.,

2013; Wiese et al., 2016), three estimates use the GRACE spherical harmonics solutions (Velicogna et al., 2014; Wiese et al., 2016; Wouters et al., 2013) and one gridded GRACE

products (Sasgen et. al., 2013).

All GRACE time series were provided as monthly time series except for the one using the

Martín-Español et al. (2016) method that were provided as annual estimates. In addition, different groups use different GIA corrections, therefore the spread of the trend solutions represents also the error associated to the GIA correction which, in Antarctica, is the largest source of uncertainty. Sasgen et al. (2013) used their own GIA solution (Sasgen et al., 2017),

Martín-Español et al. (2016) as well, Luthcke et al., (2013), Velicogna et al. (2014) and Groh and Horwath (2016) used IJ05-R2 (Ivins et al., 2013), Wouter et al. (2013) used Whitehouse et al. (2012), and Wise et al. (2016) used A et al. (2013). In addition, Groh and Horwath (2016) did not include the peripheral glaciers and ice caps, while all other estimates do.

Table 6 shows the Antarctic contribution to sea level during 2005-2015 from the different

GRACE solutions, and for the input and output method (IOM).. There is a single IOM-based dataset that provides trends for the period 1993-2015 (update of Rignot et al., 2011). 
[revised manuscript text omitted]
. However, it is important to notice that $n$ is of comparable magnitude as the GMSL trend uncertainty, currently estimated to ~0.3 mm/yr (see sub section 2.2). In Table 9a, we summarize the values of $n$ according to works in the literature where various GIA model models and averaging methods have been employed. Based on values in Table 9a for which a standard deviation is available, the average of $n$ (weighted by the inverse of associated errors), assumed to represent the best estimate, is $n = (-0.29 \pm 0.02)$ mm/yr where the uncertainty corresponds to $2\sigma$.

**2.8.2 GIA correction to GRACE-based ocean mass**

GRACE observations of present-day gravity variations are sensitive to GIA, due to the sheer amount of rock material that is transported by GIA throughout the mantle and the resulting changes in surface topography, especially over the formerly glaciated areas. The continuous change in the gravity field results in a nearly linear signal in GRACE observations. Since the gravity field is determined by global mass redistribution, GIA models used to correct GRACE data need to be global as well, especially when the region of interest is represented by all ocean areas. To date, the only global ice reconstruction publicly available is provided by the University of Toronto. Their latest product, named ICE-6G, has been published and distributed in 2015 (Peltier et al., 2015); note that the ice history has been simultaneously constrained with a specific Earth model, named VM5a. During the early period of the GRACE mission, the available Toronto model was ICE-5G (VM2) (Peltier, 2004). However, different groups have independently computed GIA model solutions based on the Toronto ice history reconstruction, by using different implementations of GIA codes and somehow different Earth models. The most widely used model is the one by Paulson et al. (2007), later updated by A et al. (2013). Both studies use a deglaciation history based on ICE-5G, but differ for the viscosity profile of the mantle: A et al. use a 3D compressible Earth with VM2 viscosity profile and a PREM-based elastic structure used by Peltier (2004), whereas Paulson et al. (2007) use an incompressible Earth with self-gravitation, and a Maxwell 1-D multi-layer mantle. Over most of the oceans, the GIA signature is much smaller than over the continents. However, once integrated over the global ocean, the signal *w* due to GIA is about -1 mm/yr of equivalent sea level change (Chambers et al., 2010), which is of the same order of magnitude as the total ocean mass change induced by increased ice melt (Leuliette and Willis, 2011). The main uncertainty in the GIA contribution to ocean mass change estimates, apart from the general uncertainty in ice history and Earth mechanical properties, originates from the importance of changes in the orientation of the Earth's rotation axis (Chambers et al., 2010, Tamisiea, 2011). Different choices in implementing the so-called "rotational feedback" lead to significant changes in the resulting GIA contribution to GRACE estimates. The issue of properly accounting from rotational effects has not been settled yet (Mitrovica et al., 2005, Peltier and Luthcke, 2009, Mitrovica and Wahr, 2011, Martinec and Hagedoorn, 2014). Table 9b summarizes the values of the mass-rate GIA contribution *w* according to the literature, where various models and averaging methods are employed. The weighed average of the values in Table 9b for which an assessment of the standard deviation is available, is $w = (-1.44 \pm 0.36)$ mm/yr (the uncertainty is $2\sigma$), which we assume to represent the preferred estimate.

**2.8.3 GIA correction to GRACE-based terrestrial water storage**

As discussed in the previous section, the GIA correction to apply to GRACE over land is significant, especially in regions formerly covered by the ice sheets (Canada and Scandinavia). Over Canada, GIA models significantly differ. This is illustrated in Figure 13 that shows difference between two models of GIA correction to GRACE over land, the A et al. (2013) and Peltier et al. (2009) models. We see that over the majority of the land areas, differences are small, except over north Canada, in particular around the Hudson Bay, where differences larger than +/- 20 mm/yr SLE are noticed. This may affect GRACE-based TWS estimates over Canadian river basins.

[Figure]

*Figure 13: Difference map between two models of GIA correction to GRACE over land:  A et al. (2013) versus Peltier et al. (2015), the units in mm/yr SLE.*

[revised manuscript text omitted]

(c) GIA contribution to GRACE-based terrestrial water storage change

| Reference | GIA correction (mm/yr SLE) (without Greenland, Antarctica, Iceland, Svalbard, Hudson Bay and Black Sea |
|---|---|
| A et al. (2013) | 0.63 |
| Peltier ICE5G | 0.68 |
| Peltier ICE6G_rc | 0.71 |

ANU_ICE6G                     0.53

(d) GIA contribution to GRACE mass-rate of the ice sheets

| Reference | *Greenland*
*GIA*
*(Gt/yr)* | *Notes* |
|---|---|---|
| Simpson et al. (2009)[r] | -3 ± 12[m] | Thermodynamic sheet / solid Earth model, 1D (uncoupled); constrained by geomorphology; inversion results in Sutterley *et al.* (2014). |
| Peltier (2009) (ICE-5G)[g] | -4[f] | Ice load reconstruction / solid Earth model, 1D (ICE-5G / similar to VM2); Greenland component of ICE-5G (13 Gt/yr) + Laurentide component of ICE-5G (-17 Gt/yr); inversion results in Khan *et al.* 2016, Discussion. |
| Khan *et al.* (2016) (GGG-1D)[r] | 15 ± 10[f] | Ice load reconstruction / solid Earth model, 1D (uncoupled); constrained with geomorphology & GPS; Greenland component (+32 Gt/yr) + Laurentide component of ICE-5G (-17 Gt/yr); inversion results in Khan *et al.* (2016), Discussion. |
| Fleming *et al.* (2004)[r] (Green1) | 3[f] | Ice load reconstruction / solid Earth model, 1D (uncoupled); constrained with geomorphology; Greenland component (+ 20 Gt/yr) + Laurentide component of ICE-5G (-17 Gt/yr); inversion in Sasgen *et al.* (2012, supplement). |
| Wu *et al.* (2010)[g] | -69 ± 19[m] | Joint inversion estimate based on GPS, satellite laser ranging, and very long baseline interferometry, and bottom pressure from ocean model output; inversion results in Sutterley *et al.* (2014). |
| Reference | *Antarctica*
*GIA*
*(Gt/yr)* | *Notes* |

| | | |
|---|---|---|
| Whitehouse *et al.* (2012) (W12a)[r] | 60[n] | Thermodynamic sheet / solid Earth model, 1D (uncoupled); constrained by geomorphology; inversion results in Shepherd *et al.* 2012, supplement (Fig. S8). |
| Ivins *et al.* (2013) (IJ05_R2)[r] | 40-65[n] | Ice load reconstruction / solid Earth model, 1D; constrained by geomorphology and GPS uplift rates; Ivins et al. 2013; inversion results in Shepherd *et al.* 2012, supplement (Fig. S8). |
| Peltier (2009) (ICE-5G)[g] | 140-180[n] | Ice load reconstruction / solid Earth model ICE-5G(VM2); constrained by geomorphology; inversion results in Shepherd *et al.* 2012, supplement (Fig. S8). |
| Argus *et al.* (2014) (ICE-6G)[g] | 107[n] | Ice load reconstruction / solid Earth model ICE-6G(VM5a); constrained by geomorphology and GPS; theory recently corrected by Purcell *et al.* 2016; inversion results in Argus *et al.* (2014), conclusion 7.8. |
| Sasgen *et al.* (2017) (REGINA)[r] | $55 \pm 22$[f] | Joint inversion estimate based on GRACE, altimetry, GPS and viscoelastic response functions; lateral heterogeneous Earth model parameters; inversion results in Sasgen *et al.* (2017), Table 1. |
| Gunter *et al.* (2014) (G14)[r] | ca. $64 \pm 40$[a] (multimodel uncert.) | Joint inversion estimate based on GRACE, altimetry, GPS and regional climate model output; conversion of uplift to mass using average rock density; inversion results in, Gunter *et al.* (2014) Table 1. |
| Martin-Español *et al.* (2016) (RATES)[r] | $55 \pm 8$ $45 \pm 7$* | Joint inversion estimate based on GRACE, altimetry, GPS and regional climate model output; inversion results in Sasgen *et al.* (2017), * is improved for GIA of smaller spatial scales; inversion results in Martin-Español *et al.* (2016), Fig. 6. |

[r] regional model; [g] global model; [m] mascon inversion; [f] forward modeling inversion; [a]
averaging kernel inversion; [n] inversion method not specified.

[revised manuscript text omitted]

We are grateful to the anonymous reviewer for his/her thorough comments that helped to improve the manuscript.